# Long term MAX-DOAS measurements of NO₂, HCHO and aerosols and evaluation of corresponding satellite data products over Mohali in the Indo-Gangetic plain

Vinod Kumar[1], Steffen Beirle[1], Steffen Dörner[1], Abhishek Kumar Mishra[2], Sebastian Donner[1], Yang Wang[1], Vinayak Sinha[2] and Thomas Wagner[1]

[1]Max Planck Institute for Chemistry, Mainz
[2]Department of Earth and Environmental Sciences, Indian Institute of Science Education and Research Mohali

*Correspondence to*: Vinod Kumar (vinod.kumar@mpic.de)

**Abstract.** We present comprehensive long term ground-based MAX-DOAS measurements of aerosols, nitrogen dioxide (NO₂) and formaldehyde (HCHO) from Mohali (30.667 °N, 76.739 °E, 310m above mean sea level), located in the densely populated Indo-Gangetic Plain (IGP) of India. We investigate the temporal variation of tropospheric columns, surface volume mixing ratio (VMR) and vertical profiles of aerosols, NO₂ and HCHO and identify factors driving their ambient levels and distributions for the period from January 2013 to June 2017. We observed mean aerosol optical depth (AOD) at 360 nm, tropospheric NO₂ vertical column density (VCD) and tropospheric HCHO VCD for the measurement period to be $0.63 \pm 0.51$, $(6.7 \pm 4.1) \times 10^{15}$ molecules cm$^{-2}$ and $(12.1 \pm 7.5) \times 10^{15}$ molecules cm$^{-2}$, respectively. Concerning the tropospheric NO₂ VCDs, Mohali was found to be less polluted than urban and suburban locations of China and western countries, but comparable HCHO VCDs were observed. For the more than four years of measurements during which the region around the measurement location underwent significant urban development, we did not observe obvious annual trends in AOD, NO₂ and HCHO. High tropospheric NO₂ VCDs were observed in periods with enhanced biomass and biofuel combustion (e.g. agricultural residue burning and domestic burning for heating). Highest tropospheric HCHO VCDs were observed in agricultural residue burning periods with favourable meteorological conditions for photochemical formation, which in previous studies have shown an implication on high ambient ozone also over the IGP. Highest AOD is observed in the monsoon season, indicating possible hygroscopic growth of the aerosol particles. Most of the NO₂ is located close to the surface, whereas significant HCHO is present at higher altitudes up to 600 meters during summer indicating active photochemistry at high altitudes. The vertical distribution of aerosol, NO₂, and HCHO follow the change in ERA5 BLH between summer and winter. However, deep convection during monsoon transports the pollutants at high altitudes similar to summer despite a shallow ERA5 BLH. Strong gradients in the vertical profiles of HCHO are observed during the months when primary anthropogenic sources dominate the formaldehyde production. High-resolution MODIS AOD measurements correlate well but were systematically higher than MAX-DOAS AODs. The ground-based MAX-DOAS measurements were used to evaluate three NO₂ data products and two HCHO data products of the ozone monitoring instrument (OMI) for the first time over India and the IGP. NO₂ VCDs from OMI correlate reasonably with MAX-DOAS VCDs, but are lower by ~30-50% due to the difference in vertical sensitivities

and the rather large OMI footprint. OMI HCHO VCDs exceed the MAX-DOAS VCDs by up to 30%. We show that there is significant scope for improvement in the a priori vertical profiles of trace gases, which are used in OMI retrievals. The difference in vertical representativeness was found to be crucial for the observed biases in $NO_2$ and HCHO surface VMR intercomparisons. Using the ratio of $NO_2$ and HCHO VCDs measured from MAX-DOAS, we have found that the peak daytime ozone production regime is sensitive to both $NO_x$ and VOCs in winter but strongly sensitive to $NO_x$ in other seasons.

## 1 Introduction

Air pollution is a serious issue in south Asia with Indo-Gangetic plain (IGP) being one of the hotspots of both present and future forecasts (Giles, 2005). For example, over the IGP, the ambient air quality standard of several criteria air pollutants (e.g. ozone, $PM_{10}$ and $PM_{2.5}$) are violated for more than 60% of the days in a year (Pawar et al., 2015;Kumar et al., 2016). $NO_x$ (sum of NO and $NO_2$) and volatile organic compounds (VOCs) are the precursors of ozone and secondary organic aerosols. Formaldehyde, the most abundant carbonyl compound in the atmosphere, is the primary source of $HO_2$ radicals in the troposphere, which in the presence of $NO_x$, can ramp up ozone production (Wolfe et al., 2016;Fortems-Cheiney et al., 2012). While $NO_x$ is majorly an anthropogenic primary pollutant, formaldehyde has both biogenic and anthropogenic sources and majorly formed during the atmospheric oxidation of methane and VOCs (e.g. alkenes), a process that is related to the production of ozone in the troposphere. An increase in both $NO_2$ and HCHO over India with an average annual rate of 2.2% and 1.5% per year respectively was documented using more than 15 years of dataset from multiple satellite instruments (Mahajan et al., 2015). Significant spatial and seasonal variabilities in $NO_2$ and HCHO were shown in these studies, but their maximum tropospheric columns were observed over the IGP.

Differential optical absorption spectroscopy (DOAS) (Platt, 1994), a technique based on the Beer Lambert's law, has found its versatile application in the last two decades for remote sensing of tropospheric pollutants including aerosol, $NO_2$ and HCHO from both ground-based and space-borne platforms. Ground-based Multi-Axis (MAX-) DOAS instruments provide continuous measurements of trace gases and aerosol including their vertical profile by observing scattered sunlight at different, mostly slant, elevation angles (Hönninger et al., 2004;Wagner et al., 2011;Wagner et al., 2004;Sinreich et al., 2005). One of the significant advantages of this technique is that, from one spectrum, many atmospheric constituents (e.g. aerosol, $NO_2$, HCHO, BrO, glyoxal, HONO, oxygen dimer ($O_4$), $SO_2$ and water vapour) can be quantified. Another advantage of MAX-DOAS technique is that it does not require a radiometric calibration and can be operated autonomously even in very remote locations. Due to its simple design, vast applicability for detection of multiple atmospheric constituents, low power demand, minimal maintenance, possible automation and remote access, MAX-DOAS instruments have been extensively employed both for long term monitoring (Ma et al., 2013;Chan et al., 2019;Wang et al., 2017a;Wang et al., 2017b) and extensive fields campaigns (Li et al., 2013;Heckel et al., 2005;Schreier et al., 2020;Halla et al., 2011) over the last decade. These measurements have been used for characterization of pollution and its source attribution (Wang et al., 2014), emission strength (Shaiganfar et al., 2017;Shaiganfar et al., 2011), chemistry and transport (MacDonald et al., 2012) and validation of satellite observations (Wang

et al., 2017a;Drosoglou et al., 2017;Mendolia et al., 2013). Over India, ground-based measurements of trace gases are limited

primarily to *in situ* measurements (e.g. Gaur et al. (2014), Sinha et al. (2014) and Kumar et al. (2016)), whereas MAX-DOAS measurement of trace gases (e.g. $NO_2$, HCHO) and aerosol have rarely been reported. The few studies are limited only to four days of mobile measurement around Delhi (Shaiganfar et al., 2011) for estimation of $NO_x$ emission from Delhi and satellite validation and more recently at a suburban site Pantnagar (29.03° N, 79.47° E), (Hoque et al., 2018) and a rural site Barkachha (25.06°N, 82.59°E ) in the Indo Gangetic plain (Biswas et al., 2019). Though *in situ* techniques provide crucial continuous

measurements of targeted atmospheric pollutants (e.g. $NO_x$, $O_3$, aerosol and VOCs), logistical constraints in their setup and maintenance limit their spatial and temporal coverage. Unless specifically designed inlets are used to alternate between different altitudes or mounted on aircraft/balloon, these measurements also lack the information about vertical profiles.

The DOAS principle has also been applied on the backscattered signal measured in the UV and Visible wavelengths by several sun-synchronous satellite instruments to provide almost daily global coverage of spatial distribution of aerosol (Torres et al.,

2007;Levy et al., 2013), $NO_2$ (Boersma et al., 2011), formaldehyde (González Abad et al., 2015;Zara et al., 2018) and several other trace gases (Gonzalez Abad et al., 2019) for more than two decades. Satellite observations of $NO_2$ and HCHO have been extensively used for a variety of applications ranging from (but not limited to) validating chemistry transport models in various atmospheric environments, assessment of bottom-up emission inventories, assessing seasonal and long term trends, constraining emissions strength $NO_x$ sources, lifetime of $NO_x$ (Beirle et al., 2011;Huijnen et al., 2010;Ma et al., 2013;Chan et

al., 2019),  VOC emissions trends, characterization and their source contribution (Kaiser et al., 2018;Zhu et al., 2017;Fu et al., 2007). Satellite observations over India have been employed to study long term trends and spatial distribution of $NO_2$ and HCHO, trends in $NO_x$ emissions (Ghude et al., 2008;Ghude et al., 2013;Mahajan et al., 2015;Hilboll et al., 2013), investigation of important processes contributing to HCHO formation and constraining the VOC emissions (Chaliyakunnel et al., 2019;Surl et al., 2018).

Despite their attractive spatial coverage, satellite observations have their inherent uncertainties arising due to retrieval algorithm, presence of clouds, the underlying assumption for calculations of a priori profiles, airmass factors and background corrections. For example, satellite measurements of trace gases located close to the surface usually underestimate the actual values around megacities due to the so-called aerosol shielding and gradient smoothing effect (Ma et al., 2013). Ground-based remote sensing techniques, e.g. MAX-DOAS has proved instrumental for validation of satellite measurements (Wang et al.,

2017a;Jin et al., 2016;Ma et al., 2013;Schreier et al., 2020;Mendolia et al., 2013;Irie et al., 2008;Brinksma et al., 2008). In addition to the validation, MAX-DOAS measurements complement the satellite observation by providing information about the diurnal and vertical profiles of trace gases and aerosol. Additionally, the MAX-DOAS observations also have the potential to bridge the gap between the scales of *in situ* and satellite observations as the prior are more sensitive to concentration close to their inlet, whereas the latter are representative of a larger area up to few hundred sq. Kilometres.

Over India, application of ground-based remote sensing techniques for validation of atmospheric chemistry and composition observations is limited majorly for aerosol measurements, except for  the one by Shaiganfar et al. (2011) using four days of mobile measurements. Over polluted regions, OMI was found to underestimate the $NO_2$ VCDs while the inverse was observed

for clean regions. Sun photometers have been used in the past to validate MODIS AOD measurements (Tripathi et al., 2005;Mhawish et al., 2019). Considering the spatial and temporal variation of emission sources pan India, there is an urgent

need for validation of satellite observations of trace gases with ground-based remote sensing measurements. Even though several *in situ* measurements of $NO_2$ have been reported over India, the fundamental difference in the retrieved information for satellite and *in situ* measurements (VCD and surface concentration) also precludes a direct intercomparison. The two stationary MAX-DOAS measurements so far over India focussed primarily on surface volume mixing ratios (VMRs) and have not reported the VCDs of trace gases, and hence lack intercomparison with the satellite observation. To the best of our

knowledge, so far there have not been any measurements probing the vertical distribution of $NO_2$ and formaldehyde over India, which limits the understanding of vertical transport of pollutants at various temporal scales (diurnal or seasonal). Moreover, the retrieved profile results close to the surface can also be compared to *in situ* measurements.

In this paper, we present more than four years (January 2013 – June 2017) of MAX-DOAS measurements of AOD, $NO_2$ and HCHO vertical column densities, vertical profiles and surface concentration (extinction) from Mohali in the north-west IGP

and investigate the factors driving these parameters. We perform a detailed comparison of several $NO_2$, HCHO and AOD data products of OMI and MAIAC AOD data product of MODIS with MAX-DOAS measurements and discuss the discrepancies. The ratio of $NO_2$ and formaldehyde was employed to investigate whether $NO_x$ or VOC drives ozone production in different seasons. The volume mixing ratio of $NO_2$ and HCHO close to the surface was evaluated with *in situ* measurements to understand the spatial representativeness of both the measurements.

**2 Experimental and Data analyses**

**2.1 Site Description**

Here, we report the measurements and satellite observations from a suburban site Mohali, located in the north-west Indo Gangetic plain. Fig. 1 shows the location of Mohali, major cities and terrain map around north India. We also show the spatial distribution of mean $NO_2$ tropospheric vertical column densities probed by TROPOMI satellite (van Geffen et al.,

2017;Veefkind et al., 2012) for the period December 2017-October 2018 in a 25km × 25km box around Mohali. The Himalayan mountain range starts at ~35km in the North. The *in situ* and ground-based remote sensing measurements were performed at IISER Mohali atmospheric chemistry facility (30.667 °N, 76.739 °E, 310m amsl). A detailed description of the site including the seasonal characteristic of meteorological parameters, characteristic wind sectors and major emission can be found elsewhere (Kumar et al., 2016;Pawar et al., 2015;Sinha et al., 2014). In the wind sector spanning from north to east, the urban

city Chandigarh is located, while the west and south comprise mostly of agricultural lands and small cities/towns. Two major power plants are located within 50 km of the measurement site. At ~45km in the north-west (~342°), is the 1260 MW Guru Gobind Singh super thermal power plant (PP1), Rupnagar, which was operational with 90% of its capacity until 2014. Since February 2014, a 1400 MW power plant (PP2) is functional in Rajpura which is ~18 km southwest (~230°) of the measurement site.

For the climate of India, the year can be divided into the following four seasons: winter (November to February), summer (March-June), monsoon (July-August) and post-monsoon (September-October). Large scale crop residue burning events occur in the late summer and late post-monsoon, which strongly perturb the atmospheric chemistry and composition (Kumar et al., 2018;Sarkar et al., 2013). In order to account for these perturbations, Kumar et al. (2016) have recommended further classification of summer and post-monsoon months in clean and polluted periods. The primary fetch region of the airmasses

is north-west throughout the year except monsoon season when wind direction is primarily south-east. Fig. F1 shows the wind rose plots indicating the wind speed and wind direction frequencies around Mohali in the four major seasons over the measurement period. Over the years 2012-2017 rapid urbanization has happened in Mohali and nearby regions (e.g. the commissioning of new international airport terminal and highways, extended construction activities for residential (e.g. Aero city, ECO city) and institutional (Knowledge city, Medicity) purpose). According to the census of 2011, the district of Mohali

holds the top rank in urban population growth among all districts in the state of Punjab at a rate of 90.2% for the period between years 2001-2011 (Tripathi and Mahey, 2017).

## 2.2 MAX-DOAS measurement setup and spectral analysis

A Multi Axis Differential Optical Absorption Spectroscopy (MAX-DOAS) instrument (Hoffmann Messtechnik GmbH) was installed at ca. 20 m above ground level with an azimuth viewing direction of 10º anticlockwise from the north. The instrument

primarily consists of a Czerny Turner spectrometer (ocean optics USB 2000+), an optical assembly consisting of a quartz lens that collects the scattered sunlight, quartz optical fibre that transmits the light to the spectrometer, and electronics in a sealed metal box. The box is mounted on a stepper motor which can be programmed to set the elevation viewing angle of the instrument. The spectral resolution of the spectrometer was ~0.7 nm in the spectral range of 318 nm – 465 nm with the field of view (FOV) of 0.7°. In order to avoid that light outside the telescope's FOV will be scattered onto the fibre, a black tube

(ca. 6 cm long) was mounted in front of the lens. The scattered sunlight spectra were recorded for elevation viewing angles 1º, 2º, 4º, 6º, 8º, 10º, 15º, 30º and 90º at a total integration time (number of scans × acquisition time for one scan) of 60 seconds each. Since the complete MAX-DOAS instrument was mounted outside, it was important to adjust the detector temperature so that the following two conditions are met:

     1.   The detector temperature is lower than the ambient temperature.

2.   The difference between the ambient temperature and detector temperature is not more than 20 ºC. This ensures that the workload on the Peltier cooler is manageable, and the detector temperature is stable.

Hence depending on the seasons, the detector temperature was adjusted. Fig F2 shows the nominal detector temperatures ($T_{set}$) and actual detector temperature ($T_{cold}$) for the various periods during the measurement period. The dark current and offset spectra were recorded every night, and while performing the spectral analysis, these were subtracted from the measured spectra

recorded at similar detector temperature. Additional offset was corrected from the measured spectrum accounting for the mean intensity recorded by the dark pixels (pixel no. 1-6 among the 2048 pixels) of the spectrometer. Wavelength to pixel calibrations were performed in QDOAS software (http://uv-vis.aeronomie.be/software/QDOAS/ : last access

03.09.2020)(Danckaert et al., 2012) every time the detector temperature was changed, by matching the structures in a measured spectrum in the zenith direction at around noontime with those in a highly resolved solar spectrum. Horizon scans were performed every day at around 12:00 local time (L.T.). Fig F3 shows a typical variation of measured intensity and its derivatives at three different wavelengths over an elevation angle range from -3° to 3°. A steep increase was observed in the measured intensity, which was centred between 0º and 0.3º for various wavelengths chosen for analysing the horizon scan. During the period of measurement, the horizon in the viewing direction was determined by a residential building with a height of about 40m at a distance of 3 km. The viewing angle of this visible horizon would be about 0.38° in good agreement with the results from the elevation scan. Thus we did not need to perform any correction for the true horizon in further analyses (Donner et al., 2020). We also see from Fig. F3 that the field of view (FOV) of the instrument is rather large (> 0.7°), and typically the RMS of the spectral analysis for the measurements at 1° elevation is substantially larger than those for the higher elevation angles. Hence, we excluded the measurements at 1° elevation angle from further analyses.

The measured spectra of the scattered sunlight were analysed for $NO_2$, HCHO and the oxygen dimer ($O_4$) using the QDOAS software. Table1 lists the wavelength intervals, included cross-sections and other relevant details pertaining to the different retrievals, and Fig 2 shows example DOAS fits and residuals for these retrievals. The typical values (peak of the frequency distribution) of the root mean square (RMS) of the DOAS fit residuals are around $5\times10^{-4}$, $7\times10^{-4}$, $6\times10^{-4}$, $6\times10^{-4}$, for $O_4$, $NO_2$ (UV), $NO_2$ (VIS) and HCHO, respectively. In order to retain analyses results corresponding to good quality fits, we have excluded the $O_4$, $NO_2$ and HCHO dSCDs corresponding to a RMS greater than $2\times10^{-3}$ and solar zenith angles higher than 85° (Wang et al., 2019). The RMS threshold removes 1.1%, 1.4%, 0.7% and 1.3% of the $O_4$, $NO_2$ (UV), $NO_2$ (VIS) and HCHO dSCDs, respectively, of all the measured dSCDs at solar zenith angles less than 85°.

The spectral analysis is performed with respect to a Fraunhofer reference spectrum (FRS) measured in the zenith direction of each complete elevation angle measurement sequence in order to account for the Fraunhofer lines and the stratospheric contribution of the absorbers (Hönninger et al., 2004). For analysing the off-axis spectra measured at time 't', we calculate the FRS at the time of the measurement by interpolating the zenith spectra measured before and after the complete measurement sequence. Thus, the primary retrieved quantity from MAX-DOAS spectral analysis is the so-called differential slant column density ($dSCD$). $dSCD$ of a trace gas (absorber) at an elevation angle α can be regarded as the difference between the absorber concentration integrated along the photon path at elevation angle α ($SCD_\alpha$) and zenith direction ($SCD_{90}$).

$$dSCD_\alpha = SCD_\alpha - SCD_{90} \qquad\qquad 1$$

dSCD is related to the tropospheric vertical column density (VCD) through differential airmass factors (dAMF or $AMF_\alpha - AMF_{90}$):

$$VCD = \frac{dSCD}{dAMF} \qquad\qquad 2$$

Unless specifically mentioned, we will refer VCD to tropospheric VCDs in the paper hereafter. The airmass factors are related to the light path of the photons reaching the telescope of the instrument and depend on several parameters, e.g. elevation angle,

solar zenith angle, relative solar azimuth angle with respect to the instrument, surface reflectance, aerosol and trace gas vertical profile and aerosol optical properties. In section 2.4, we provide the details of the calculation of airmass factors and subsequent profile inversion to retrieve the VCDs and vertical profiles.

DOAS retrievals of $NO_2$ can be performed both in the UV and visible wavelength windows, but have their respective advantages and limitations. $NO_2$ has a stronger absorption in the visible as compared to UV. Hence, the DOAS fit in the visible results in smaller fit uncertainties as compared to that in the UV (Fig. A1). This is an important aspect, especially for instruments with rather low quantum efficiencies like the detector of the MAX-DOAS instrument used in this study. However, the profile inversion methods (please refer to section 2.4) to retrieve $NO_2$ vertical profiles and VCDs also require information about aerosol extinction profiles at the same wavelength as that used for $NO_2$ retrieval. The aerosol extinction profiles retrieval for DOAS relies on the $O_4$ measurements, which has a rather weaker absorption at the visible wavelengths (in the spectral range of our instrument). Hence an alternative approach is to use an Ångström exponent to scale the aerosol extinction profiles derived at UV wavelengths to visible wavelengths. The aerosol extinction profiles calculated for the visible wavelength are subsequently used as an input parameter for the $NO_2$ profile inversion in the visible window. In Appendix A, we compare the performance and internal consistency of the $NO_2$ profiles and VCDs retrieved in the UV and visible by the profile inversion algorithm and by the geometric approximation under various sky conditions. Briefly, we found very good agreement between the $NO_2$ VCDs retrieved in the UV and visible under clear sky conditions with low aerosol load (slope =0.95 and r=0.9). Even in the clear sky case with high aerosol load and cloudy sky conditions, a reasonable agreement for $NO_2$ VCDs between retrieval in UV and Visible was observed (slope =0.75 and 0.78 for high aerosol and cloudy cases respectively, r=0.82 for both). The $NO_2$ dSCDs from the retrieval in UV wavelength window were found in good agreement with those retrieved in visible, but systematically lower (r=0.9, slope=0.95 and a negative offset of $1.1 \times 10^{14}$ molecules cm$^{-2}$).

**2.3 Cloud classification**

Clouds have a strong impact on MAX-DOAS measurements and subsequent profile inversion as they alter the light path and intensity (Wagner et al., 2014;Wagner et al., 2011). Clouds are generally not included within the radiative transfer models for profile inversion. The cloud classification scheme is based on the measured radiances at 360 nm, colour index (ratio of measured radiances at 330 and 390 nm) and the measured $O_4$ airmass factors ($O_4$ SCD/ $O_4$ VCD) (for details see Wagner et al. (2016)). Besides the absolute values of these quantities, also their temporal variation and their elevation dependencies are considered. The threshold for these quantities (the spread of $O_4$, the normalised CI, the spread of the CI and the temporal variation of the CI) are parametrized as polynomials of the SZA as provided in Wagner et al. (2016). We can classify the sky conditions into the following seven categories: 1. Clear sky with low aerosol load, 2. Clear sky with high aerosol load (AOD > 0.85 at 330nm), 3. Broken clouds, 4. Cloud holes, 5. Continuous clouds, 6. Fog, and 7. Optically thick clouds (Wagner et al., 2014;Wagner et al., 2016). In the first step, using the colour index (CI), its variation across zenith spectrum for adjacent elevation sequence and its variation within an elevation sequence, the primary cloud classification is performed to retrieve information about primary conditions (types 1-5). Identification of fog and thick clouds is performed in the second step using

O$_4$ AMF and measured radiance at 360 nm. Fog is identified if there is very little variation in O$_4$ dSCD for different elevation angles within the measurement sequence. The details pertaining to the calculation of thresholds for radiance for identification of thick clouds are provided in Appendix B. While the classification of aerosols might be slightly affected by the specific properties of the local aerosol, the cloud classification is robust to the variability of aerosol properties. However, this is not critical here, because the main aim – the cloud classification – is hardly affected by these specific aerosol properties.

Fig. 3 shows the percentage of the mean monthly sky condition for the complete measurement period. The most prominent sky condition in all the seasons is "clear sky with high aerosol load", comprising ~48 % of the total. March and April are marked by the maximum occurrence of clear sky conditions with low aerosol load at ~18%. Continuous clouds and optically thick clouds are most abundant in July-August which is marked by the Monsoon season. Please note that due to the widespread crop residue burning and suppressed meteorological conditions, October and November months witness severe smog events in the north-west Indo Gangetic plain leading to very poor air quality and low visibility. The poor air quality conditions extend until December due to emission from domestic burning for heating and similar meteorological conditions. The prevalent high aerosol load conditions are also marked by the cloud classification algorithm as seen by occurrences greater than 55% in October, November and December. Fog is observed in December and January when winter is at its peak.

## 2.4 Profile inversion to retrieve vertical profiles and vertical column densities from slant column densities

In order to account for the complex dependence of airmass factors on viewing geometry and measurement conditions, radiative transfer models (e.g. McARTIM (Deutschmann et al., 2011)) are employed. dAMFs are calculated for various combinations nodes of viewing geometry and profiles (of trace gases and aerosol) which are stored offline as multi-dimensional lookup tables (LUTs) for various wavelengths (e.g. separate LUTs for 343nm, 360nm and 430nm). Profile inversion techniques use these LUTs to determine the scenarios which best match the measured dSCDs. From these scenarios, the aerosol and trace gas profiles, VCDs and AOD are derived.

We have used MAPA (Mainz Profile algorithm)(Beirle et al., 2019) version 0.98 for this purpose. The vertical profiles of trace gas concentrations (or aerosol extinction) can be parametrised using three profile parameters namely column parameters (c) (VCD for trace gases and AOD for aerosol), height parameter (h) and shape parameter (s) (Beirle et al., 2019;Wagner et al., 2011). In the first step, the aerosol profiles are retrieved using the measured O$_4$ dSCDs. A Monte Carlo approach is utilized to identify the best ensemble of the forward model parameters (h, s and c) which fit the measured O$_4$ dSCDs for the sequence of elevation angles. Generally, a scaling factor (0.8 in most of the cases) is applied to the measured O$_4$ dSCD before they are used in the profile retrieval (see Wagner et al. (2019) and references therein). The reason for this scaling factor is still not understood, and in section 3.3, we also investigate the effect of the different scaling factors on the intercomparison of the retrieved AOD with satellite observation from MODIS. In the second step, the aerosol profiles retrieved from the O$_4$ inversion are used as input to retrieve the similar model parameters (h, s and c) for the trace gases (e.g. NO$_2$ and HCHO). To assess the quality of the retrievals, MAPA also provides "valid", "warning" or "error" flags for each measurement sequence, which are calculated based on pre-defined thresholds for various fit parameters (Beirle et al., 2019). We have used the lookup tables calculated at

360 nm for the inversion of $O_4$ and $NO_2$ in the UV, 343 nm for HCHO and 430 nm for $NO_2$ in the visible window. For the

HCHO profile inversion, we observed unrealistic h and s at high solar zenith angles (SZA> 60°), which are probably related

to spectral interferences with the ozone absorption within the DOAS analysis. Therefore, we only consider HCHO profile

results for measurements with SZA less than 60°. For the retrieval of $NO_2$ in the visible wavelength window and HCHO, the

aerosol extinction profiles retrieved at 360 nm were scaled to those at 430 nm and 343 nm using an Ångström exponent of

1.54. This value was derived as the mean of the Ångström exponent (AE) between 470-550 nm measured by MODIS for the

measurement period, where we do not observe a strong intra-annual variation (Fig. F4). We have calculated the Ångström

exponent ($AE$) using the measured AOD at 470 nm and 550 nm according to:

$$AE = -\frac{\log\left(AOD_{470}/AOD_{550}\right)}{\log\left(470/550\right)} \qquad\qquad 3$$

We also investigated the effect of the choice of Ångström exponent on the profile inversion for a smaller subset of our data

spanning 15 days. We found that AE values of 1.25 and 1.75 (minimum 5[th] percentile and maximum 95[th] percentile in Fig. F4)

resulted in the same number of valid retrievals and the difference in the mean $NO_2$ VCD was less than 0.1%. The surface $NO_2$

concentrations were slightly higher (4%) for AE value of 1.25 and were 3% lower for AE value of 1.75 as compared to those

for an AE value of 1.54.

For the profile inversion using MAPA, we compare the number of valid retrievals and retrievals flagged as warning and error

in Table 2. We note that the sky condition associated with thick clouds and fog are mostly flagged as errors in the profile

inversion. For the analyses results shown in the paper hereafter, we only retained the DOAS measurements corresponding to

sky condition without thick clouds and fog.

## 2.5 Satellite data

### 2.5.1 OMI

The ozone monitoring instrument (OMI) aboard the AURA satellite crosses the equator at 13:42 local solar time in the

ascending orbit. OMI has an effective ground pixel size of 13×24 km$^2$ at nadir view, which broadens up to 13×150 km$^2$ at the

swath edges (Levelt et al., 2006;Schenkeveld et al., 2017). The intensity of backscattered solar radiation from the Earth's

atmosphere is measured by two spectrometers in three bands in the spectral ranges of 260-311nm, 307-383 nm and 349-503

nm. With an across-track swath width of ~2600km, OMI provides daily coverage of the earth in 14 orbits. Several data products

of aerosol optical depth (AOD), $NO_2$ and HCHO retrieved from OMI measurements are available, which we briefly describe

in Appendix C and compare against MAX-DOAS observation in the subsequent sections. From the level 2 data corresponding

to individual orbits every day, we have retained the data pertaining to the centre of the ground pixel within 0.25⁰ × 0.25⁰ of the

measurement site. For sensitivity analysis, we have also separately selected the collocated pixels, i.e. OMI pixels whose corner

points contained the exact location of the measurement location. Ground pixels with effective cloud fraction > 0.3 or
measurements affected by the so-called row anomaly and problems in the DOAS retrieval were filtered out for the analysis. In
order to minimize the effect of the diurnal variation, we have only chosen the MAX-DOAS measurements between 07:00 UTC
and 09:00 UTC (between 12:30 and 14:30 local time) for comparison with OMI observations. We have used the OMAERUV
data product for AOD; DOMINO V2, OMNO2 v3.0 and the QA4ECV data products for $NO_2$ and the OMHCHO and QA4ECV
data products for HCHO for the evaluation with the corresponding MAX-DOAS retrieved quantities. The retrieval and quality
control details pertaining to these data products are provided in appendix C.

### 2.5.2 MODIS

Satellite measurements of the aerosol optical depth (AOD) were also obtained from the MODIS instruments on-board the
TERRA and AQUA satellites having equator overpasses at local solar times 10:30 and 13:30 respectively. We have used the
MAIAC data product available at $1 \times 1$ km$^2$ spatial resolution (Lyapustin et al., 2018). Since MAIAC is a combined product
of MODIS TERRA and AQUA, we have chosen the daily means of the DOAS measurements between 9:30 and 11:30 and
between 12:30 and 14:30 local time for the intercomparison.

### 2.6 Ancillary measurements

*In situ* measurements of $NO_x$ (NO and $NO_2$) were performed using a model 42i trace level analyser (Thermo Fischer Scientific)
based on the chemiluminescence technique. Ambient $NO_2$ is first converted into NO using a heated molybdenum converter,
which further reacts with excess ozone generated inside the instrument to produce a chemiluminescence signal proportional to
the available NO. Checks for zero drifts were performed every week, and 5 point calibrations were performed every month.
Meteorological parameters (e.g. ambient temperature, rainfall, wind speed, wind direction and relative humidity) were
measured using collocated met sensors (Met One Instruments Inc.). The details pertaining to the measurements principle and
calibration protocol can be found elsewhere (Sinha et al., 2014;Kumar et al., 2016).

*In situ* measurements of formaldehyde were performed at Mohali using a high sensitivity proton transfer reaction mass
spectrometer (PTR-MS) (Lindinger et al., 1998). Inside the PTR-MS, HCHO (proton affinity=170.4 kcal mol $^{-1}$) is chemically
ionised by hydronium ions ($H_3O^+$), because of its higher proton affinity than that of water vapour (proton affinity=165.2 kcal
mol$^{-1}$), prior to its detection using a quadrupole mass analyser. Using a PTR-MS, HCHO is detected at a protonated mass to
charge ratio (m/z) 31. The measured signal (counts s$^{-1}$) is converted to VMRs using the m/z dependent sensitivity factors,
which are usually determined using calibration experiments. Due to the degradation of the detector, the sensitivity of PTR-MS
might also change with time, which is evaluated using routine calibration performed using a gas standard of known VMR
(Chandra and Sinha, 2016;Sinha et al., 2014). However, such a calibration for HCHO could not be performed due to the
unavailability of a calibration standard. Hence, we have calculated theoretical sensitivity factors for HCHO, similar to the
method incorporated by Kumar et al. (2018). The major sources of uncertainty in the HCHO measurements, when a theoretical
sensitivity factor is used, are from 1) uncertainty in the proton transfer reaction rate constant of the reaction between HCHO

and $H_3O^+$ (~ 15%) and 2) ratio of transmission efficiencies of $HCHOH^+$ and $H_3O^+$ (~25%) (Zhao and Zhang, 2004;de Gouw and Warneke, 2007). Systematic uncertainties due to degradation of detector would increase with time. Ambient humidity is also known to interfere with the formaldehyde measurements with a PTR-MS, and we have performed an absolute humidity-based correction, according to Cui et al. (2016). HCHO VMRs increase by ~30% on average after application of the absolute humidity-based correction.

## 3 Results and discussion

### 3.1 Seasonal and annual trends of AOD, NO₂, HCHO

Fig. 4 shows the time series of monthly mean AOD, $NO_2$ VCD and HCHO VCD for the complete measurement period from January 2013 until June 2017. The vertical error bars show the monthly variability as the interquartile range. The gaps in the time series are due to instrument malfunction (primarily due to stepper motor and connection to measurement computer). The mean AOD for the complete measurement period was $0.63 \pm 0.51$, with the monthly means varying between 0.30 in March 2013 and 1.25 in August 2014. We quantify the seasonal variability of the measured AOD to be 187% which is calculated as the difference between the maximum and minimum of the 30 days running mean divided by the mean over the measurement period. For the same months, very small inter-annual variability in the monthly means (<0.2) was observed for all the months except April-July. We observe the maximum AOD (0.8-1.2) during the monsoon months (June - August) at Mohali, which is most probably caused by the hygroscopic growth of the aerosol particles (Altaratz et al., 2013). Previous studies comparing various MODIS AOD data products and AERONET measurements over the Indo-Gangetic plain have also found the maximum AOD in the monsoon months, but also significant differences among the different data products (Mhawish et al., 2019;Tripathi et al., 2005). The retrieval of the aerosol size distribution from AERONET measurements has shown that during monsoon, the coarse mode fraction increases by more than 50% of the annual average over the Indo-Gangetic plain (Tripathi et al., 2005). In order to further confirm that the high AOD during the monsoon is not an artefact caused by the persistent cloud cover over the IGP, we have investigated the seasonality of the AOD under cloud-free conditions measured at different AERONET sites in the IGP nearest to Mohali (New Delhi ~250 km south and Lahore: ~250 km west). Both at Lahore and New Delhi, high AOD values are observed in the monsoon months (June-August) (Fig. F5). Relatively high monthly mean AOD (0.6-0.9) values are also observed in May and October, which are characterized by crop residue fire emissions. We have also compared the MODIS AOD (converted to 440 nm using the AE derived from equation 3) to the AERONET AOD at these two stations and found very good agreement in the daily measured values with Pearson correlation coefficients (r) > 0.84 (Fig. F5) and an overall bias < 10% for both sites.

The mean $NO_2$ VCD for the complete measurement period was $(6.7 \pm 4.1) \times 10^{15}$ molecules $cm^{-2}$ with a variability in the monthly means between $4.7 \times 10^{15}$ molecules $cm^{-2}$ in July 2015 and $8.9 \times 10^{15}$ molecules $cm^{-2}$ in November 2013. The observed $NO_2$ VCDs are comparable to those observed in long term measurements in rural and suburban environments (Kramer et al., 2008;Drosoglou et al., 2017) and satellite observations in the Indian metropolitan city Mumbai (Hilboll et al., 2013). These

are much smaller than those observed in the urban areas worldwide (Mendolia et al., 2013;Drosoglou et al., 2017) or rural, suburban and urban locations of China (Ma et al., 2013;Wang et al., 2017a;Chan et al., 2019;Jin et al., 2016;Vlemmix et al., 2015), where the mean monthly levels are generally higher than $1 \times 10^{16}$ molecules cm$^{-2}$. The observed seasonal variability of 145% in the 30 days running means can be explained by the seasonality of emissions and the changing lifetimes.

For $NO_2$, we see that the monsoon months (July and August) are cleanest followed by early pre-monsoon months (March-April). The primary emission sources active throughout the year in and around the region are vehicular emissions, garbage burning and biomass burning for cooking and construction activities. In the suburban and rural regions around the measurement site, biomass (e.g. wood, domestic and agricultural residue) and biofuel (coal) burning serve as the primary source of heating. Increased emissions from biomass burning for domestic heating in winter when the atmospheric lifetime of $NO_2$ is maximum, marks the highest observed $NO_2$ VCDs in the year. Also, during the crop residue burning active periods of summer (May-June) and post monsoon (October-November), we observe enhanced $NO_2$ VCDs. Notwithstanding the strong urbanization trends, a significant annual trend is not observed in either of the three parameters (AOD, $NO_2$ and HCHO) measured by the MAX-DOAS, indicating the dominance of non-organized emission sources, e.g. domestic heating, garbage burning and crop residue burning for the ~ 4.5 years measurement period. This is further ascertained by the fact that we do not see any noticeable weekday-weekend dependence in either of $NO_2$, HCHO and AOD (Fig. F6).

The mean HCHO VCD for the complete measurement period was $(12.1 \pm 7.5) \times 10^{15}$ molecules cm$^{-2}$, with a strong seasonality with monthly means ranging from $5.3 \times 10^{15}$ molecules cm$^{-2}$ in March 2017 and $17.7 \times 10^{15}$ molecules cm$^{-2}$ in October 2015. The seasonal variability was found to be strongest at 284% for HCHO among the three parameters measured by the MAX-DOAS. The mean and monthly variability is comparable to those observed in the urban areas of China (Vlemmix et al., 2015;Wang et al., 2017a). In contrast to these long term measurements of formaldehyde in China, the minimum VCDs are not observed in the winter, but rather observed in March. Globally photochemical production from biogenic and anthropogenic hydrocarbons dominates the formaldehyde sources while having minor fraction being directly emitted from biomass burning and vegetation (Fortems-Cheiney et al., 2012). At complex suburban environments, e.g. Mohali, different sources can dominate the formaldehyde production in different periods of the year. At an urban site Kolkata in the IGP, the contribution of primary sources to ambient formaldehyde was observed to be 71% and 32% during summer and winter, respectively (Dutta et al., 2010).

At Mohali two distinct formaldehyde enhancement periods are observed; first in May-June and second in October, both of which are the periods when crop residue burning is practised around the region. Several identified and unidentified chemical compounds are formed in the atmosphere due to crop residue fires, which readily react with OH radicals and form second and higher generation oxidation products, and potentially form formaldehyde as a by-product (Kumar et al., 2018;Sarkar et al., 2013). The pre-monsoon period of May and June provides favourable conditions (e.g. long daytime hours, uninterrupted solar radiation and high temperature) for the photochemical production of secondary pollutants from the precursors emitted from wheat residue fires (Kumar et al., 2016;Sinha et al., 2014). This is reflected in very high formaldehyde VCDs observed in May and June in the range of $16.9 \times 10^{15} - 17.4 \times 10^{15}$ molecules cm$^{-2}$. Similar levels of HCHO VCDs are also observed in October,

when paddy residue fires are active in the region. Also, the photochemical production of ozone is attributed to the emission from crop residue fires by Kumar et al. (2016). The observed maximum $NO_2$ VCDs in the winter months indicates very high primary emissions, which could eventually lead to formaldehyde production from its co-emitted precursors. However, low ambient temperatures and lack of ample sunlight hours result in not so high HCHO VCDs as compared to May, June and October.

The agricultural lands in the northwest fetch region of Mohali practice agroforestry, where poplar (most common) and eucalyptus trees are planted in the periphery of fields (Sinha et al., 2014;Pathak et al., 2014;Mishra et al., 2020). Over the Indo-Gangetic plain, biogenic sources contribute to ~40% of the total VOC emission flux annually (Chaliyakunnel et al., 2019). In the early post-monsoon season, soil moisture availability and daytime temperatures between 30-35 °C provide a favourable condition for isoprene emissions (Guenther et al., 1991). Generally, a strong variability is observed in the number of rainfall events and the total rainfall over the IGP between different years (Fukushima et al., 2019), which also affects the soil moisture availability and in turn the biogenic emissions from plants. For the period discussed in this study with available MAX-DOAS HCHO measurements, the years 2014 and 2015 were quite different with respect to the monsoon rainfall. During the monsoon months, 2014 witnessed 18 rainy days (total rainfall = 378 mm), while 2015 witnessed 32 rainy days (total rainfall = 435 mm) in Mohali. Following the number of rainfall events, the early post monsoon months (Aug-Sep) of 2015 witness higher HCHO VCDs, which can be attributed to the photo-oxidation of stronger biogenic emissions of isoprene (Mishra and Sinha, 2020). Poplar trees in the Indian subcontinent show little emissions in the months from December to March (Singh et al., 2007) due to loss of leaves. Minimal biogenic and anthropogenic emissions of formaldehyde and its precursors in March resulted in the minimum VCDs during the year. From these observations, we conclude that anthropogenic emissions (primarily due to biomass burning) and their oxidation dominate the formaldehyde seasonality in most of the year except early post monsoon where biogenic emissions have a major contribution to the measured formaldehyde.

### 3.2 Diurnal variation and vertical profiles of AOD, $NO_2$ and HCHO

Fig. 5 shows the diurnal variation of the mean vertical profiles of aerosol extinction, $NO_2$ concentration and HCHO concentration for different months of the year retrieved using MAX-DOAS measurements at Mohali. The corresponding diurnal variations of the VMR of $NO_2$ and HCHO and aerosol extinction close to the surface are shown as figures F7-F9.

The vertical profile of aerosol extinction is expected to be primarily driven by the boundary layer height (BLH) and to some extent, the photochemistry, which eventually drives secondary aerosol formation (Wang et al., 2019). At Mohali, the diurnal evolution of the aerosol extinction profile heights reaches its maximum during afternoon hours. In Fig. 6A, we show the typical diurnal evolution of BLH from the ERA5 reanalysis data for the four major seasons. We observe a growth of the BLH from morning until noon with a maximum at 14:00 L.T. and a subsequent decline. The maximum BLH up to 3 km is observed in summer. Shallow daytime BLH up to 1.2 km are observed in the monsoon period due to overcast sky condition, stronger wind and high surface moisture, and in winter due to low surface temperature and low surface heat flux (Sathyanadh et al., 2017). We observe that the aerosol is trapped in the bottom layers (within 400m) in winter, whereas during the afternoon hours in

summer, monsoon and early post-monsoon months, aerosol extinction up to 0.2 km$^{-1}$ is observed even at around 1.5 km altitude. Though the ERA5 BLH is shallow in monsoon, yet we observe similar aerosol profiles during that period as those during summer, which indicates that at Mohali, the vertical distribution of aerosol does not follow ERA5 BLH transition from summer to monsoon. Over India, the monsoon months are characterised by strong convective activity which can bring the surface air aloft to several km despite a shallow ERA5 BLH (Lawrence and Lelieveld, 2010). The convection is rather strong

in the Himalayan foothill region (which also includes Mohali) and pumps the surface pollutants even into the UTLS (upper troposphere/lower stratosphere) (Fadnavis et al., 2015). The evidence of pollutant transport associated with deep convection is crucial for PAN formation in the UTLS, which is observed by the modelling studies over the IGP and Himalayan region. Long-lived non-methane VOCs (e.g. ethane) can be transported to the UTLS where both convective transported NO$_x$ from the surface and exchanged from stratosphere serve as fuel for PAN formation. High aerosol extinction (>1.5 km$^{-1}$) is observed in

the surface layer in the winter months, with the maximum in December. In the winter months, biomass burning contributes to primary aerosol, the formation of secondary aerosol and particle growth as a result of coating on existing aerosol particles. Moreover, the high ambient relative humidity in winter (Kumar et al., 2016) further contributes to the growth of the existing aerosol particles, which further increases the aerosol extinction and in the extreme cases can lead to intense fog.

In order to quantitatively describe the mixing altitude, we define a characteristic profile height H$_{75}$, as the height below which

75% of the trace gas column (or AOD) is located (Vlemmix et al., 2015). The profile parameters "h" used in the MAPA inversion algorithm represents the height below which the concentration (or extinction) of the trace gas (or aerosol) remains constant. For elevated profiles (for details, please refer to Wagner et al. (2011)), h refers to the height above the trace gas (or aerosol) layer where the concentration(or extinction) becomes zero. Using "h" and shape parameter "s", we calculate a profile height (H$_{75}$) by employing equations 2-5 of Beirle et al. (2019).

We show the diurnal evolution of characteristic profile heights (H$_{75}$) in Fig. F10 for the four major seasons. Fig. 6B shows the mean afternoon time characteristic profile heights (H$_{75}$) for aerosol, NO$_2$ and HCHO for different months, together with the mean ERA5 BLH. Due to their short atmospheric lifetime (< 6 hours) during daytime, H$_{75}$ for NO$_2$ and HCHO are lower than those for aerosol. H$_{75}$ for the measured species are observed to be smaller than the typical boundary layer heights. In the monsoon season, we observe H$_{75}$ comparable to those in summer, even though the boundary layer height is shallow and

comparable to that in winter. Trace gases and aerosol from the surface are lofted up due to deep convection in the monsoon leading to high H$_{75}$. This indicates that the vertical mixing of aerosol during the monsoon is not driven by the parameters used to calculate the ERA5 BLH, but rather follows the trend of ambient daytime temperature, which does not show such large difference between summer and monsoon (e.g. Fig. S2 of Kumar et al. (2016)). The profile heights for aerosols and HCHO in summer months are similar to those observed in Beijing (Vlemmix et al., 2015), but we observe a much stronger seasonal

dependence. H$_{75}$ up to 1.1 km for aerosols are observed for the summer (except March) and monsoon months, while in winter H$_{75}$ is usually less than 500 meters. In all months, we observe the minimum H$_{75}$ among the three measured species for NO$_2$ (~200 m to 450 m), due to its short lifetime and production close to emission sources near the surface. H$_{75}$ for NO$_2$ is generally much smaller than for Beijing (urban) and Xingtai (suburban) in China (Wang et al., 2019;Vlemmix et al., 2015).

In all the months and hours of the day, the major fraction of the $NO_2$ column is located in the bottom-most layer extending from surface until 200 meters. Until 11:00 local time (L.T.), more than 60% of the $NO_2$ column is located in the bottom-most layer in all the months. In winter months (Nov-Feb), the same is true for all hours of the day, but in the morning time (until 11:00 L.T.) the fraction in the bottom layer is even >80%. During the late afternoon of the summertime, monsoon and early post monsoon, when BLH grow deeper due to heating of the surface; we observe a considerable fraction (~20-30%) of $NO_2$ in the layer extending from 200 m-400 m. There is a very small fraction (<5%) of $NO_2$ column in the layers above 600 m.

We observe the maximum $NO_2$ columns in the morning hours between 08:00-11:00 local time, subsequently decreasing during the day. The major factors driving the $NO_2$ columns are emissions and lifetime with respect to OH radicals. For the surface concentration, boundary layer dynamics also play an important role. Emissions from traffic and biomass burning for heating and cooking peak in the morning and evening hours. The lifetime of $NO_2$ is minimum in the afternoon when OH radial concentration peaks, which explains the decrease during the day. The amplitude of the diurnal profiles of the $NO_2$ surface VMR (Fig. F7) is maximum during winter months when there is little diurnal variability in the BLH(dilution effect) and $NO_2$ lifetime (sink effect) between morning and late afternoon hours and the shape is driven primarily by the emissions (source). The amplitude is minimum during monsoon when biomass burning is ceased (Mishra and Sinha, 2020). In section 3.7, we will discuss the diurnal variation of the $NO_2$ surface VMR retrieved from MAX-DOAS and *in situ* measurements for different months.

For formaldehyde, we observe a comparable distribution among the 0-200m and 200-400m layers for all seasons except winter. In winter months, the bottom-most layer contains up to 50% of the total column, which is smaller as compared to aerosols and $NO_2$. During the late afternoon hours of summer, monsoon and the post monsoon period, we also observe a larger fraction of the HCHO column in the 200-400 m layer as compared to the 0-200 m layer. In the presence of high $NO_x$ close to the surface, OH concentration is depleted, which might result in slowing down of the formaldehyde production from precursor VOCs. The higher characteristic profile heights of HCHO as compared to those of $NO_2$ can be attributed to the secondary photochemical formation from primary precursor emissions. A considerable fraction of primary emissions is transported to intermediate layers (similar to $NO_2$) during summer, monsoon and early post-monsoon, where secondary products (e.g. formaldehyde) are formed due to active photochemistry. The months in which primary anthropogenic emissions of formaldehyde and its precursors are stronger (e.g. months except for March, April, July, August and September), the gradients of vertical profiles of HCHO are stronger in the layers from the surface to 1 km altitude. For the surface VMR, we observe maxima in the morning hours in all seasons except winter and late post monsoon (Fig. F8). Even though formaldehyde is formed photochemically, which should increase during the day, the VMR close to the surface is reduced due to the vertical mixing in the afternoon hours. We observe the highest daytime HCHO surface VMR in winter months since the major fraction of HCHO is trapped in the bottom-most layer.

## 3.3 Intercomparison and temporal trends of aerosol optical depth

Fig. 7A shows the time series of the aerosol optical depth (AOD) at 360 nm retrieved from MAX-DOAS measurements of $O_4$ (scaled by 0.8) and the AOD at 360 nm calculated from MODIS MAIAC data product over Mohali. The corresponding scatter plot is shown in Fig. 7B. Similar plots for OMAERUV AOD product at 354 nm are shown in Figures 7C and 7D. The solid line represents the monthly mean values, while the individual dots in the background represent the daily measurements. The vertical error bars represent the standard error of the mean ($\sigma/\sqrt{N}$), where σ represents the standard deviation and $N$ represent the number of daily measurements for the month.

The mean MAX-DOAS AOD at 360 nm for the measurement period if averaged around the MODIS overpass time (between 9:30 and 11:30 and between 12:30 and 14:30 L.T.) was 0.59 ± 0.39 as compared to the MODIS AOD of 0.81 ± 0.53. The correlation coefficients (r) for the linear regressions of daily and monthly mean data between the MAX-DOAS and MODIS AOD are found to be 0.78 and 0.85, respectively. For the monthly mean, we also performed the orthogonal distance regression (ODR) weighted by $\sigma^{-2}$ (where σ is the monthly standard deviation) between MAX-DOAS and MODIS AOD, for which the slope and offset were 1.13 and 0.12, respectively.

Initially, we performed a comparison of AOD retrieved from MAX-DOAS measurements without applying any scaling factor to the measured $O_4$ dSCD. While there was a general agreement between the trends in the AOD retrieved from MAX-DOAS and MODIS, the MAX-DOAS AOD showed a strong underestimation (Fig. F11). Several MAX-DOAS measurement studies and comparison with independent datasets (e.g. sun photometer) found that a scaling factor (less than one) was necessary to bring MAX-DOAS results and independent measurements into agreement (Wagner et al., 2009;Wagner et al., 2019;Beirle et al., 2019). However, a similar number of studies did not find the need to apply such a scaling factor (Wang et al., 2017a;Franco et al., 2015). Currently, the reason for the scaling factors is not understood (Wagner et al. (2019) and references therein). In most cases, when scaling factors are used, values between about 0.8 and 0.9 are found best. In our case, the application of a scaling factor was found to be necessary to bring MAX-DOAS and satellite measurements into an agreement. In order to further confirm the choice of the scaling factor, the profile inversion was performed with a variable scaling. Fig. F12 shows the distribution of the scaling factor which concludes that an $O_4$ scaling factor of 0.8 fits best for our measurements.

NASA OMAERUV data product provides the AOD at 354 nm, but the spatial resolution and temporal coverage are not as good as for the MODIS MAIAC product. From Fig. 7C and D, we observe that OMAERUV generally underestimates the AOD over Mohali and the level of agreement is also worse both for daily and monthly mean values. For OMAERUV, the mean AOD was 0.53 ± 0.26. Over central and east ASIA, independent comparison of OMI with AERONET measurements also found a ~50% underestimation by OMAERUV and a poor agreement for a 10-year period (Zhang et al., 2016). We think that the much coarser spatial resolution of OMAERUV as compared to the MAIAC data product is among the probable reasons for the worse agreement. In order to evaluate this hypothesis, we created a time series of the MODIS MAIAC data product, spatially averaged over 5 km and 25 km around Mohali.

MAX-DOAS measurements are spatially representative of a few kilometres in the field of view, depending on the ambient aerosol load and elevation angle, whereas the ground footprints of individual OMI pixels are $13\times24$ km$^2$ in the best case. We have calculated the horizontal sensitivity distance (HSD) of MAX-DOAS for low elevation angles as the e-folding distance of O$_4$ dAMF from the instrument location (Wagner and Beirle, 2016). Figure F13 shows that the mean afternoon time (between 12:00 and 15:00 local time) HSD ranges between 5 and 7 km for clear sky condition with low aerosol load and between 3 and 6 km for high aerosol conditions. Here it is important to note that this estimate is mainly representative for the near-surface layers. While for the trace gas inversions, the VCD is constrained by all elevation angles, the determination of the AOD is mostly constrained by the high elevation angles. For high elevation angles, the sensitivity range is much closer to the instrument (at distances up to 1 and 2 km for layer height of 0.5 and 1 km, respectively) (see Fig 6A). Comparing the spatially degraded time series with MAX-DOAS AOD resulted in a worse agreement (r=0.75 and 0.79 for the daily and monthly means, respectively) for 25 km but did not change significantly for 5 km (Fig. F14) as compared to the original comparison when only a 2 km area around Mohali was considered for spatial averaging. In addition to the effect of horizontal gradients, the poor agreement of the OMAERUV product might also be caused by residual cloud contamination. Further, non-representative assumptions about the aerosol types between smoke, dust and non-absorbing aerosols used for the inversion of the measured reflectances might play a role, as highlighted by *Zhang et al.*, 2016.

### 3.4 Intercomparison and temporal trends of NO$_2$ vertical columns

Fig. 8 shows the time series of the NO$_2$ vertical column densities measured by MAX-DOAS and by OMI for the three different data products. Please note that for calculating the monthly means, we only considered the days of the month when both cloud screened, and quality controlled satellite and MAX-DOAS data were available. Within the complete study period, the DOMINO, QA4ECV and OMNO2 data products had 60%, 47% and 46% days of cloud and quality screened respectively. We observe a general agreement in the trends of the NO$_2$ VCDs between the MAX-DOAS and OMI datasets. However, all OMI data products generally underestimate the NO$_2$ VCDs. The mean MAX-DOAS NO$_2$ VCD for the measurement period if averaged between 12:30 and 14:30 local time (around the OMI overpass) was $(5.4 \pm 3.0) \times10^{15}$ molecules cm$^{-2}$. The mean VCD for OMI was $(3.7 \pm 2.4) \times 10^{15}$ molecules cm$^{-2}$, $(3.7 \pm 1.3) \times10^{15}$ molecules cm$^{-2}$ and $(3.7 \pm 1.7) \times 10^{15}$ molecules cm$^{-2}$, respectively, for the DOMINO, QA4ECV, and OMNO2 data products. The reasons for the systematic difference between the MAX-DOAS and OMI measurements can be attributed to several factors including, 1) difference in spatial representation and 2) differences in vertical sensitivity of MAX-DOAS and OMI. Previous validation studies over China also found systematic underestimation of ~60% over Nanjing by the OMNO2 and ~30% over Beijing and Nanjing by the DOMINO product (Chan et al., 2019;Ma et al., 2013). The observed discrepancies were attributed to differences in spatial representativeness which introduces smoothing of the measured NO$_2$ VCD over a large satellite ground pixel and to the shielding effect of aerosols. At Mohali, the maximum disagreement of the OMI products is observed during the late post monsoon and winter months, where all satellite data products significantly underestimate the NO$_2$ VCDs. Note that a large amount of aerosol and trace gases are

emitted from the crop residue and domestic fires in these months, a major fraction of which is trapped close to the surface due to suppressed ventilation.

Fig. 8 also shows the linear regression fits of the three OMI data products versus the MAX-DOAS $NO_2$ VCDs. We observe smaller scatter in the QA4ECV and OMNO2 products for both daily and monthly values, as compared to DOMINO product. An orthogonal distance regression (ODR) fit was performed between the monthly means of MAX-DOAS and OMI $NO_2$ VCDs weighted by $\sigma^{-2}$, where $\sigma$ is the monthly standard deviation. The slopes of the ODR fit between the MAX-DOAS and OMI monthly mean $NO_2$ VCDs were 0.94, 0.59 and 0.78, respectively, for DOMINO, QA4ECV and OMNO2. The offsets of the ODR fits were $-8.1\times10^{14}$, $8.4\times10^{14}$, and $-1.7\times10^{14}$, respectively for DOMINO, QA4ECV and OMNO2, respectively. Over Mohali, we observe excellent consistency between the QA4ECV and OMNO2 products with a slope and correlation coefficient (r) of 0.94 and 0.72, respectively between both datasets.

Since we have retained the OMI pixels whose centre points lie within $0.25º \times 0.25º$ of the MAX-DOAS measurement site, differences can arise due to the smoothing effect across the OMI ground pixels. With the pristine regions of the Himalayan mountain range only ~ 35 km from Mohali, smaller $NO_2$ VCD from OMI measurements are expected due to systematic gradients towards the mountain range. The effect of smoothing over a large area can be minimized if only collocated pixels are retained for the intercomparison. However, this significantly reduced the number of available days for intercomparison. If only collocated pixels were considered, we were left with only 35%, 25% and 22% of the measurement days for DOMINO, QA4ECV and OMNO2, respectively. Due to the poor statistics, we did not observe improvements in the correlation coefficient (r) of the linear regression of the daily data and these changed from 0.38, 0.50 and 0.43 to 0.38, 0.56, 0.43, respectively, for DOMINO, QA4ECV and OMNO2. In the absence of a higher resolution $NO_2$ data for the study period, we could not quantify the effect of the different spatial representativeness of the MAX-DOAS and OMI measurements.

One of the major reasons for the disagreement between satellite and MAX-DOAS measurement is the difference in vertical sensitivity of the two measurements. Satellite instruments have limited sensitivity close to the ground. In contrast, MAX-DOAS measurements have the highest sensitivity close to the ground, while it becomes virtually zero above 3-4 km. The limited sensitivity of the satellite instruments is addressed using the box airmass factors (bAMFs) and the a priori profiles of the trace gases to be retrieved (Eskes and Boersma, 2003). In appendix D we compare the bAMFs used in the DOMINO and QA4ECV retrievals with those calculated by employing the radiative transfer model (RTM) McARTIM over Mohali using the mean aerosol extinction profiles retrieved from the MAX-DOAS measurements. A discrepancy is found between the calculated bAMFs and those used for OMI retrievals (DOMINO and QA4ECV) (Fig. D1), such that the calculated bAMFs show systematically higher values close to the surface. In such a case, attribution of a smaller fraction of $NO_2$ in layers close to the surface in the a priori profiles cause a systematic underestimation of the VCDs (please see appendix D for details). We found that the a priori $NO_2$ profiles used in the DOMINO v2 retrievals strongly differ from those retrieved using the MAX-DOAS measurements for winter and polluted post monsoon when a large fraction of $NO_2$ is present in layers close to the surface (Fig. 5 and D2).

In order to eliminate the difference caused by the non-representative a priori $NO_2$ profiles, we calculated the "modified MAX DOAS VCDs" (called $VCD_{mod}$ hereafter), which represent the MAX-DOAS $NO_2$ VCDs as observed by OMI. The application of the OMI tropospheric averaging kernels and a priori profiles also to the MAX-DOAS profiles makes the comparison independent of the a priori profiles used for the OMI retrieval. In order to do so, we apply the tropospheric averaging kernels of OMI DOMINO ($AK_{trop}$) data product to the $NO_2$ vertical profiles ($x_{doas}$) retrieved from MAX-DOAS in layers(i) from ground (i = 0) to h = 4 km (i = 20), according to the following equation (Rodgers and Connor, 2003):

$$VCD_{mod} = \sum_{i=0}^{h} AK_{trop,i}(x_{doas,i} - x_{ap,i}) + x_{ap,i} \qquad 4$$

Here, $x_{ap,i}$ represents the DOMINO a priori $NO_2$ profiles. Please note that the total averaging kernels ($AK_{tot}$) provided in the DOMINO level 2 data product are converted to tropospheric averaging kernels using the ratio of total AMF ($AMF_{tot}$) and tropospheric AMF ($AMF_{trop}$):

$$AK_{trop,i} = AK_{tot} * \frac{AMF_{tot}}{AMF_{trop}} \qquad 5$$

Fig. 9 shows the time series of original MAX-DOAS VCDs (red), OMI DOMINO VCDs (blue) and modified MAX-DOAS VCDs (black). We observe that the bias between the OMI and MAX-DOAS measurements is smaller if the averaging kernels and a priori profiles are applied to the MAX-DOAS $NO_2$ profiles. However, in contrast to the MAX DOAS VCDs, the VCD$_{MOD}$ are systematically lower than the DOMINO $NO_2$ VCDs.

While the application of the DOMINO averaging kernels and a priori $NO_2$ profiles to the MAX-DOAS profiles according to equation 4 accounts for the reduced OMI sensitivity close to the surface, it does not account for the limited sensitivity of MAX-DOAS at higher altitudes (above 3-4 km). VCD$_{MOD}$, hence represents the MAX-DOAS $NO_2$ VCD from the ground to up to ~4 km altitude, whereas the DOMINO $NO_2$ VCDs represents the $NO_2$ VCDs from the ground until the tropopause. For a qualitative estimate of the $NO_2$ column at high altitudes, we have calculated the fraction of the $NO_2$ column between 4 km altitude and the tropopause by only considering the $NO_2$ partial VCDs of the TM4 a priori profiles in various layers. The $NO_2$ partial columns in the 4 km – tropopause altitude range account for 7-18 % (interquartile range) of the total $NO_2$ a priori VCDs. Hence, due to the limited sensitivity of MAX-DOAS at higher altitudes, the VCD$_{MOD}$ is systematically smaller than DOMINO $NO_2$ VCDs.

Please note that a similar comparison with OMI QA4ECV and OMNO2 products is not possible as a priori profiles and averaging kernels, respectively, are not available for these data products. For qualitative evaluation, we have used the TM4 $NO_2$ a priori profiles with QA4ECV averaging kernels and calculated VCD$_{MOD}$. This also results in an improvement of the bias between the MAX-DOAS and QA4ECV $NO_2$ VCDs (Fig. F15). A different approach for improved agreement between MAX-DOAS and satellite VCDs is by using the MAX-DOAS $NO_2$ profiles as a priori profiles for the calculation of airmass factors for the satellite retrieval (Chan et al., 2019;De Smedt et al., 2015). We discuss this approach and its limitations in appendix D.

### 3.5 Intercomparison and temporal trends of HCHO vertical columns

Fig.10 shows the time series of HCHO VCDs measured by MAX-DOAS and by OMI for the QA4ECV (panel A) and OMHCHO products (panel C) with the respective scatter plots (panels B and D). Within the chosen quality and cloud filters, the QA4ECV and OMHCHO data sets have 42% and 67% days, respectively, out of the complete study period. The mean MAX-DOAS HCHO VCD for the measurement period if averaged between 12:30 and 14:30 local time (around OMI overpass) was $(11.3 \pm 6.9) \times 10^{15}$ molecules $cm^{-2}$. The mean VCD from OMI observations were $(14.9 \pm 11.3) \times 10^{15}$ molecules $cm^{-2}$ and $(11.3 \pm 11.7) \times 10^{15}$ molecules $cm^{-2}$ for the QA4ECV and OMHCHO data products, respectively. A small negative bias was expected in the MAX-DOAS HCHO VCDs, as its sensitivity is limited at higher altitudes where a background HCHO may be present due to the oxidation of long-lived hydrocarbon (mainly Methane). Though the non-methane VOCs dominate the formaldehyde production over land, yet methane oxidation is a ubiquitous source of formaldehyde across the globe. At high altitudes (between 3.6 and 8 km) in a pristine location (Jungfraujoch), the background HCHO VCDs have been observed between $0.75 \times 10^{15} - 1.43 \times 10^{15}$ molecules $cm^{-2}$ (Franco et al., 2015) which is equivalent to ~10% of the measured total column as measured over Mohali.

In general, we see slightly higher VCDs by the OMI QA4ECV product compared to MAX-DOAS, except for the post monsoon months, when we observe a better agreement between the two datasets. For the OMHCHO product, the monthly mean HCHO VCDs agree well with MAX-DOAS except for the post monsoon of 2015, where MAX-DOAS VCDs were higher. The generally good agreement of the OMHCHO VCDs with MAX-DOAS is in line with previous works (Chan et al., 2019;Wolfe et al., 2019). The OMHCHO product was also shown to have a good agreement with airborne measurements in the remote troposphere (Wolfe et al., 2019). However, the accountability in the range of the monthly mean HCHO VCDs was much better for the QA4ECV product (slope=0.77) as compared to OMHCHO (slope=0.41). In comparison to the QA4ECV $NO_2$ dataset, we observe a higher noise in both spatial and temporal patterns of the HCHO VCDs, which arises due to the relatively small atmospheric absorption. The larger uncertainty in the QA4ECV HCHO dataset is also evident from the scatter of daily measurements where sometimes VCDs close to zero are observed. For some months (e.g. Jan 2015, Jan 2016, Mar 2017, Apr. 2017 and June 2017) when less than 6 days of QA4ECV data is retained for calculating the monthly means for intercomparison with MAX-DOAS measurements, the values should be considered carefully. If only co-located pixels are considered for the QA4ECV HCHO product, the statistics get poorer (only 17% of valid observations) and we observe a worse coefficient of correlation (r=0.19 and 0.17 for daily and monthly means) with MAX-DOAS measurements.

The finding that in contrast to $NO_2$, no general underestimation is observed for the comparison of the satellite HCHO VCDs to the MAX-DOAS HCHO VCDs, can be attributed to the different vertical profiles and the different vertical sensitivities of MAX-DOAS and OMI observations. Since in general, the HCHO profiles reach to higher altitudes than $NO_2$, the satellite observations capture a larger fraction of the total HCHO column. However, we cannot perform a comparison of the modified MAX-DOAS VCDs for HCHO similar to that calculated for DOMINO $NO_2$, as the total AMFs and averaging kernels (needed for Eq. 5) are not available for the QA4ECV and OMHCHO data products, respectively.

**3.6 Discerning the sensitivity of ozone production on NOₓ and VOCs**

Martin et al. (2004) recommended the use of the ratio of the formaldehyde and $NO_2$ columns from satellite observations as an indicator for the ozone production regime. $HCHO/NO_2$ ratios less than 1 represent a VOC sensitive regime, whereas values greater than 2 indicate a $NO_x$ sensitive regime. Intermediate values of the $HCHO/NO_2$ ratio indicate a strong sensitivity towards both $NO_x$ and VOCs. The threshold for this indicator was initially calculated for afternoon time (between. 13:00 – 17:00 L.T.), but was later extended to also include morning period by Schroeder et al. (2017). However, Schroeder et al. (2017)also

indicated that the upper limit of the intermediate regime might vary spatio-temporally. Nonetheless, higher $HCHO/NO_2$ indicate that reduction in $NO_x$ emissions would be more effective for ozone reduction. While the *in situ* measurements of the total OH reactivity is a more robust method to evaluate the ozone production regime, due to the experimental constraints, these measurements are reported only rarely and for the short time periods (e.g. few weeks or months)(Kumar et al., 2018;Kumar and Sinha, 2014). Mahajan et al. (2015) evaluated the ozone production regime over India using the ratio of HCHO and $NO_2$

VCDs observed from SCIAMACHY for the mean of years 2002-2012. Over the north-west IGP, the $HCHO/NO_2$ was observed to be less than 1 in the winter months and between 1 and 2 in all other months. From our intercomparisons in the previous sections, we note that while the OMI $NO_2$ VCDs are generally underestimated, the HCHO VCDs are generally well accounted for. Hence the true $HCHO/NO_2$ will be smaller than those indicated by satellite observations, which indicates that the estimated sensitivity of the ozone production regime towards $NO_x$ should be smaller and shifted towards VOCs. Using WRF-CMAQ

model simulations at $36{\times}36$ km$^2$ resolution over India for 2010, Sharma et al. (2016) have evaluated the ozone production to be strongly sensitive to $NO_x$ emissions throughout the year and recommended reduction in transport emissions which account for 42% of the total $NO_x$ emissions. However, with an increase in transport and powerplant emissions (strong $NO_x$ sources) over India, the regimes are susceptible to shift away from $NO_x$ limited and need to be re-evaluated. Fig. 11 shows the monthly mean $HCHO/NO_2$ ratio calculated using the MAX-DOAS measurements for the morning (09:30-11:30 L.T.), noontime around

the OMI overpass (12:30-14:30 L.T.) and late afternoon (15:30-17:30 L.T.). We observe a stronger (smaller) sensitivity towards $NO_x$ during the late afternoon (morning) as compared to noontime similar to other urban locations in the USA (Schroeder et al., 2017).VOCs contribute to ozone production via their oxidation by OH radicals and subsequent formation of peroxy radicals. During the build-up hours of ozone (between sunrise until noontime) at Mohali, radicals' abundance is expected to be limited. Hence, the ozone production is more sensitive to VOC (or "radicals") during morning which shifts

towards $NO_x$ later during the day. In winter months, mean daytime $HCHO/NO_2$ ratios between 1 and 2 are observed, which represent sensitivity towards both $NO_x$ and VOCs. The sensitivity of the ozone production regime changes towards $NO_x$ with the onset of summer and stays like that until the end of the post-monsoon season. Over the Indo-Gangetic plain, the strongest ozone pollution episodes are observed in the summer and post monsoon months during the afternoon hours between 12:00 and 16:00 L.T.(Kumar et al., 2016;Sinha et al., 2015). Surface ozone measurements from Mohali have shown enhancement in its

ambient concentrations during the late post monsoon as compared to the early post monsoon even though the daytime temperature drops by 6 °C. During summer, enhanced precursor emission from fires lead to an increase in ~19 ppb ozone

under similar meteorological conditions. Considering the stronger sensitivity of daytime ozone production towards $NO_x$, the ozone mitigation strategies should focus on $NO_x$ emission reductions.

**3.7 Surface volume mixing ratios of $NO_2$ and HCHO**

The surface volume mixing ratios (VMR) of $NO_2$ and HCHO can be derived from the MAX-DOAS measurements using the retrieved profiles. For MAPA, the profiles are saved using output grids of a uniform thickness of 200 meters. The mean concentrations in the bottom-most layer have been used to calculate VMR by considering the measured ambient temperature. Fig. 12 shows the time series of the $NO_2$ volume mixing ratios (VMR) measured with the *in situ* chemiluminescence analyser and that in the lowest 200 m grid obtained from the profile inversion from the MAX-DOAS measurements. The measurement

frequency of the *in situ* measurements was 1 min, whereas, from MAX-DOAS, the mean concentrations are retrieved from individual complete sequences (~10 min). Hence, we averaged the in-situ measurements to coincide with the exact MAX-DOAS measurement time. We observe a reasonable agreement between the two measurements in terms of temporal variability, but the MAX-DOAS surface VMRs are systematically higher until the end of 2014. The differences between the two can probably be explained by the difference in the horizontal and vertical representativeness and resolution and have been discussed

in detail in appendix E. Briefly, the surface VMR from MAX-DOAS represents the mean in the bottom-most 200 m grid of MAPA output, while that from *in situ* measurements are more sensitive to airmass sampled closed to the inlet. Hence MAX-DOAS VMRs are representative of a larger area around the measurement site, and we infer from Fig. 1 that the measurement location in relatively cleaner from surrounding in terms of $NO_2$ levels. The mean $NO_2$ surface VMR from MAX-DOAS measurements was $8.2 \pm 6.7$ ppb, whereas that from the concurrent *in situ* measurements was $6.0 \pm 5.0$ ppb. If we consider all

*in situ* observations (which also includes the night time observation and periods when MAX-DOAS measurements were unavailable), the mean $NO_2$ VMR was $8.6 \pm 6.9$ ppb. As compared to previous MAX-DOAS measurements from India (Biswas et al., 2019), in a rural site ($0.8 \pm 0.2$ ppb), we observe much higher $NO_2$ VMR in Mohali. However, these are comparable to previous *in situ* $NO_2$ VMR measured for a period of more than one year at urban and suburban locations (distant from traffic) of India (e.g. Mohali: 8.9 ppb, Pune: ~9.5ppb/8.7ppb and Kanpur 5.7ppb) (Gaur et al., 2014;Kumar et al., 2016;Beig et al.,

2007;Debaje and Kakade, 2009), but smaller than near traffic urban measurement (e.g. New Delhi: 12.5ppb/18.6 ppb, Agra: 15-35 ppb) (Saraswati et al., 2018;Tiwari et al., 2015;Singla et al., 2011). Please note that we have used a $NO_2/NO_x$ ratio of 0.9 to estimate $NO_2$ VMR for comparison with the previous measurements which reported $NO_x$ VMR and hence have a larger uncertainty (Kunhikrishnan et al., 2006). The inset of Fig 12A shows the monthly variability of surface $NO_2$ VMR as box and whiskers plot. Similar to the VCDs, we observe maximum $NO_2$ VMR in winter followed by crop residue burning active periods

of post monsoon and summer.

Fig. 13 shows the time series of the HCHO volume mixing ratio (VMR) measured with the PTR-MS and those derived for the lowest 200 m layer obtained from profile inversion of the MAX-DOAS measurements. The mean HCHO surface VMR from MAX-DOAS measurements was $8.7 \pm 7.5$ ppb, whereas that from concurrent *in situ* PTR-MS measurements was $3.3 \pm 1.7$ ppb. For HCHO, higher VMR from MAX-DOAS measurements can be explained its photochemical formation at altitudes

higher than the inlet of PTR-MS. This has been further discussed in appendix E. The measured HCHO VMRs are comparable to previous MAX-DOAS measurements from India (Pantnagar: 2-6 ppb), but much lower than those measured previously in India using offline techniques (e.g. north Kolkata:16 ppb, south Kolkata: 11.5ppb) (Dutta et al., 2010;Hoque et al., 2018). From the monthly variation of HCHO VMRs shown in the inset of Fig 13A, we observe that the maximum is observed in late post monsoon and winter followed by late summer (crop residue burning active period). This monthly variability is slightly

different from that of the HCHO VCDs (discussed in section 3.1). A shallower profile (Fig 5) in the late post monsoon and winter leads to high surface VMR even though the VCD is smaller than that in late summer.

Fig. F7 shows the mean diurnal profiles of the surface $NO_2$ VMRs for different months using MAX-DOAS and *in situ* measurements. A typical diurnal feature representative of a suburban location was observed from the *in situ* measurements, which is explained in detailed elsewhere for Mohali for different seasons (Kumar et al., 2016). For the daytime hours, when

MAX-DOAS observations are also available, we observe a general agreement in the absolute values and temporal evolution between the two measurements in all the seasons except winter. The occurrence of a morning peak for the *in situ* measurements between 07:00 and 09:00 local time (L.T.) in all months is primarily driven by emissions (traffic and biofuel combustion). For the MAX-DOAS observations, a similar maximum occurs between 08:00 and 10:00 L.T., and between 09:00 and 11:00 L.T. in winter months. The time shift compared to the maximum in the *in situ* measurements can be explained by the accumulation

of the surface emission in the boundary layer and breaking of the nocturnal boundary layer. The latter causes the $NO_x$ present in high altitudes (e.g. from power plants >200 m, Fig. 5) to mix in with the surface layers. A modelling study performed using the CMAQ model has shown that the peak in the diurnal profiles of $NO_2$ columns occurs 2-3 hours later than that for surface concentrations (Fishman et al., 2008). Since MAX-DOAS surface VMR represents the mean over a few hundred meters, we expect similar behaviour in its comparison with *in situ* measurements. In winter months, though the diurnal trends are similar

in both observations, a general overestimation by MAX-DOAS is found, which is caused by a shallower boundary layer and the presence of high $NO_2$ mixing ratios above the inlet of the *in situ* analyser.

**4.0 Conclusions**

We have presented long term (from Jan 2013 until June 2017) MAX-DOAS measurements of $NO_2$, HCHO and aerosols from a regionally representative suburban site Mohali in the densely populated north-west Indo-Gangetic plain. MAX-DOAS

radiance measurements at 360 nm and color index (ratio of measured radiances at 330 and 390 nm) were employed to quantitatively determine the prevalent sky conditions. Clear sky with high aerosol load conditions (AOD at 360 nm >0.85) was observed for about half of the measurement period. The profile inversion algorithm MAPA was used to derive the aerosol optical depth, vertical column density of $NO_2$ and HCHO as well as vertical profiles of aerosol extinction, $NO_2$ and HCHO. Mean AOD at 360 nm, tropospheric $NO_2$ VCD and HCHO VCD for the measurement period were observed to be $0.63 \pm 0.51$,

$(6.7 \pm 4.1) \times 10^{15}$ molecules cm$^{-2}$ and $(12.1 \pm 7.5) \times 10^{15}$ molecules cm$^{-2}$ respectively, with substantial seasonal variations in all the measured parameters. While the $NO_2$ VCDs are generally lower than those observed in the suburban and urban location

of China and western countries, the HCHO VCDs are comparable to previously reported values. Despite the rapid urbanization, no evident annual trends were observed in AOD, $NO_2$ and HCHO VCDs for the measurement period. The seasonal trends are rather driven by non-organized anthropogenic sources of emissions, e.g. agricultural residue burning, waste burning, and biofuel burning for heating and cooking. Early summer (March) and monsoon (July and August) months are the cleanest with respect to the measured $NO_2$ and HCHO VCDs, but high AOD was observed during monsoon likely due to hygroscopic growth of the aerosol particles. Maximum $NO_2$ VCDs were observed in winter months, followed by periods of post-monsoon and summer when extensive crop residue burning is practised in the agricultural regions near the measurement site. Maximum formaldehyde VCDs were observed in the summer and post-monsoon seasons when photochemical production from precursors emitted from agricultural residue fires is favoured by meteorological conditions (high temperature and strong solar radiation). Biogenic sources were also found to be crucial for formaldehyde during the monsoon and early post monsoon periods. The vertical profiles retrieved from MAX-DOAS measurements show that the major fraction of the $NO_2$ column is located close to the surface (0-200 meters) in all the seasons, while the same is true for HCHO only in winter. In other seasons, comparable HCHO mixing ratios are also observed in higher layers until 600 meters indicating active photochemistry at higher altitudes. Interestingly, the seasonal variation of the vertical profiles of aerosol extinction did not depend on the significant change of the ERA5 boundary layer heights between summer and monsoon. In addition to serving as an input for retrieval of VCDs from the preliminary satellite data analysis, vertical profiles retrieved from MAX-DOAS measurements can also be used to validate the regional atmospheric chemistry models around the measured location.

We observed a very good agreement in the temporal trend, and a slight overestimation by the high-resolution MODIS MAIAC data product around Mohali with respect to the AOD retrieved from MAX-DOAS $O_4$ measurements. The OMI aerosol data product "OMAERUV" generally underestimated the AOD at Mohali compared to MAX-DOAS and the accountability is also not as good as for the MAIAC product. In an ideal case, sun photometer measurements, which provide AOD at 360 nm, are best suited for an intercomparison with the MAX-DOAS measurements. However, such measurements were not available at our measurement location, and for the future studies, sun photometer measurements are recommended for a direct comparison of the AOD at 360nm. We use the MAX-DOAS measurements of $NO_2$ and HCHO to evaluate the three widely used $NO_2$ data products (DOMINO v2, QA4ECV and OMNO2) and two HCHO data products (QA4ECV and OMHCHO) of OMI for the first time over India and the Indo-Gangetic plain. Among the three OMI $NO_2$ data products, we observe reasonable agreement between the MAX-DOAS and OMI for the latter two. However, all three OMI data products underestimate the MAX-DOAS $NO_2$ VCDs by 30-50%. The maximum discrepancy is observed in late post monsoon and in the winter months when a large amount of trace gases and aerosol is trapped in the air close to the surface. A major reason behind the discrepancy between the MAX-DOAS and OMI observations is the inaccurate representation of the a priori $NO_2$ profiles and aerosol profiles used for calculation of airmass factors in satellite retrievals. If we account for the decreased satellite sensitivity close to the ground, a smaller bias is observed between the MAX-DOAS and OMI $NO_2$ VCDs. Because of its large ground footprint, OMI measurements are representative of a larger area around the MAX-DOAS measurement site which is also responsible for part of the discrepancy. Two OMI data products of HCHO are also compared against MAX-DOAS observations. The QA4ECV

HCHO product was found to exceed the MAX-DOAS HCHO VCDs by ~30% overall, whereas a generally good agreement was found with the OMHCHO data product. Using the ratio of HCHO and $NO_2$ VCDs measured from MAX-DOAS, we found that the daytime ozone production is sensitive towards $NO_x$ in summer, monsoon and post monsoon, whereas a strong sensitivity towards both $NO_x$ and VOCs was observed in winter. We observed a transition from stronger VOC sensitivity to stronger $NO_x$ sensitivity on ozone production from morning to late afternoon.

The mean surface volume mixing ratios (VMR) of $NO_2$ and HCHO retrieved from MAX-DOAS observations for the measurement period were $8.2 \pm 6.7$ ppb and $8.7 \pm 7.5$ ppb, respectively. We have compared the surface volume mixing ratios retrieved from MAX-DOAS measurements with those measured using *in situ* analysers. The temporal intraday and the day to day variations of the $NO_2$ surface VMR agree well for both measurements, but the MAX-DOAS measurements were generally higher than those measured using the *in situ* analysers, with a bias of $0.67 \pm 2.85$ ppb. For HCHO, however, a poorer agreement in the temporal trend and a large bias of $2.29 \pm 4.47$ ppb was observed between MAX-DOAS and *in situ* measurements. The observed differences can be mainly attributed to the differences in vertical and horizontal representativeness of both the measurements. The MAX-DOAS surface VMRs are representative of trace gases (e.g. HCHO or $NO_2$) located from surface to altitudes up to a few hundred meters. We found evidence of vertical gradients of photochemically formed compounds and hence, stronger differences were observed when significant fraction of trace gases is present at altitudes higher than the inlet of *in situ* analysers.

MAX-DOAS instruments can provide standalone routine measurements of trace gases and aerosol from remote locations, which can be further employed for air quality assessment, similar to AERONET global network. (e.g. PANDORA, NDACC : http://www.ndaccdemo.org/instruments/uv-visible-spectrometer: last access 03.09.2020). A plethora of studies related to satellite observations of tropospheric pollutants from China and western countries have sensitized the authorities to reduce emissions to curb the pollution (de Foy et al., 2016). MAX-DOAS measurements were of substantial importance to validate and complement the satellite observations in these studies. With the Indian subcontinent being projected as the new hotspot of anthropogenic emissions (Li et al., 2017;De Smedt et al., 2018), similar (or new generation, e.g. TROPOMI, GEMS) satellite and corresponding MAX-DOAS observation will be of great importance for future studies.

**Competing interests**

The authors declare that they have no conflict of interest.

**Acknowledgements**

We acknowledge IISER Mohali atmospheric chemistry facility for the support with logistics related to the operation of the MAX-DOAS instrument and sharing of meteorological and *in situ* measurement data. V.K. acknowledges the Alexander von Humboldt Foundation and Max Planck Society for supporting postdoctoral stipend. A.K.M acknowledges MHRD, India, for

support regarding PhD fellowship. V.S. thanks the Max Planck Society. Max Planck Institute for Chemistry and the Department of Science and Technology, India for funding a Max Planck India partner group at IISER Mohali through which this long term collaboration could be accomplished. We acknowledge the free use of DOMINO $NO_2$ and QA4ECV $NO_2$ and HCHO level 2 data products from OMI from www.temis.nl. We thank the Level-1 and Atmosphere Archive & Distribution
System (LAADS) Distributed Active Archive Center (DAAC) for providing free access to the OMAERUV, OMNO2, OMHCHO and MAIAC data products. We thank Dr. Philippe Goloub and Dr. Brent. N. Holben for establishing and maintaining the AERONET sites New Delhi and Lahore, respectively, whose data has been used in this study. We thank the two anonymous reviewers for their constructive feedback to the manuscript.

**Data Availability**

The MODIS MAIAC data, OMI OMAERUV, OMNO2 and OMHCHO data can be downloaded from LAADS website (https://ladsweb.modaps.eosdis.nasa.gov/). The OMI DOMINO and QA4ECV data are available at www.temis.nl. The MAX-DOAS measurement data, spectral analysis and profile inversion results can be obtained from the corresponding author. The *in situ* measurement data can be obtained from IISER Mohali atmospheric chemistry facility by contacting Vinayak Sinha (vsinha@iisermohali.ac.in)

**Author contributions**

V.K. and T.W. prepared the manuscript with contributions from all co-authors. V.K., S.Dö., A.K.M and S. Do. operated the MAX-DOAS instrument, and V.K. performed spectral analyses and profile inversion with help from S.B., S.D., and T.W. Y.W. helped V.K. with cloud classification. V.K., A.K.M and V.S. operated the in situ analyzers and contributed to the analyses. All co-authors contributed to modifications and discussions in preparing the manuscript.


**Figures**

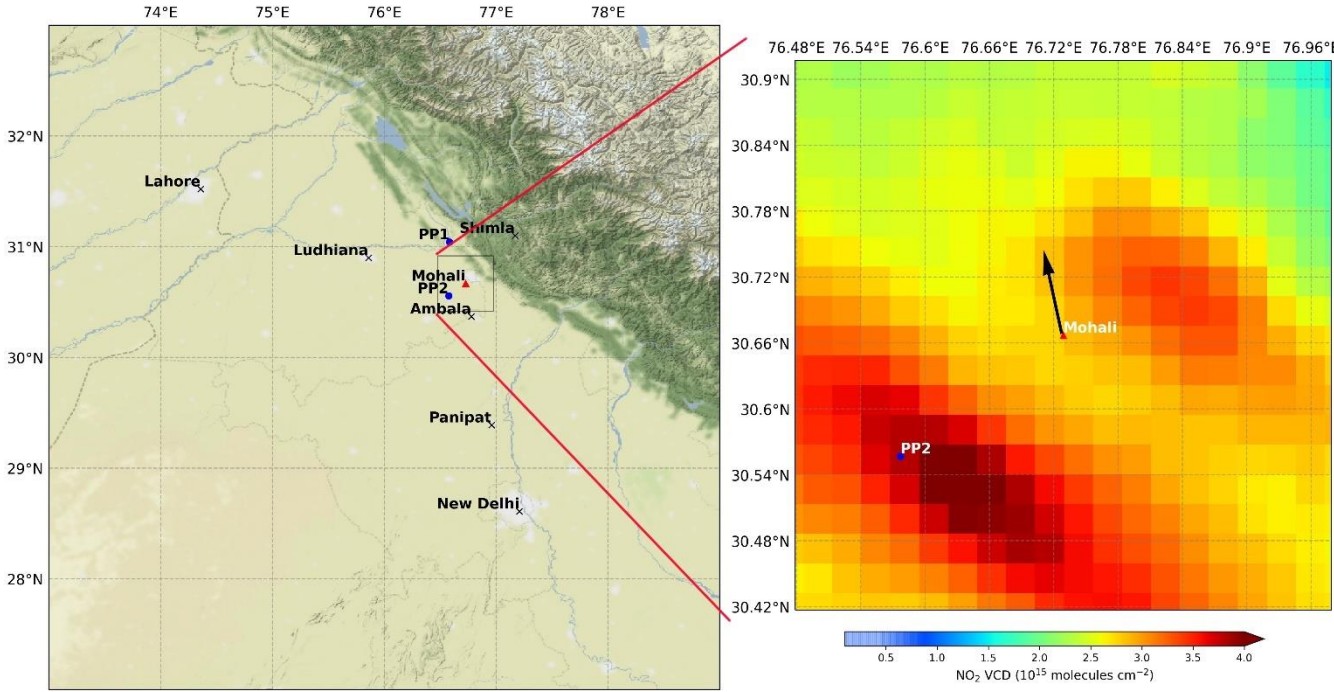

**Figure 1 Left:** Terrain map showing the location of Mohali (red triangle) in North India, along with major cities (crosses) and two major thermal power plants near Mohali (blue circles, PP1: Guru Gobind Singh Super Thermal Power plant and PP2: Larsen & Toubro Super Thermal Power Plant (NPL), Rajpura) and province boundaries. **Right:** Mean tropospheric $NO_2$ VCD measured by TROPOMI overlaid on the terrain map of $0.5° \times 0.5°$ box around Mohali (shown in the left panel) for the period December 2017-October 2018. The black arrow indicates the viewing direction of the MAX-DOAS instrument. Terrain maps are adapted from Stamen (http://maps.stamen.com/terrain, last access 03.09.2020)


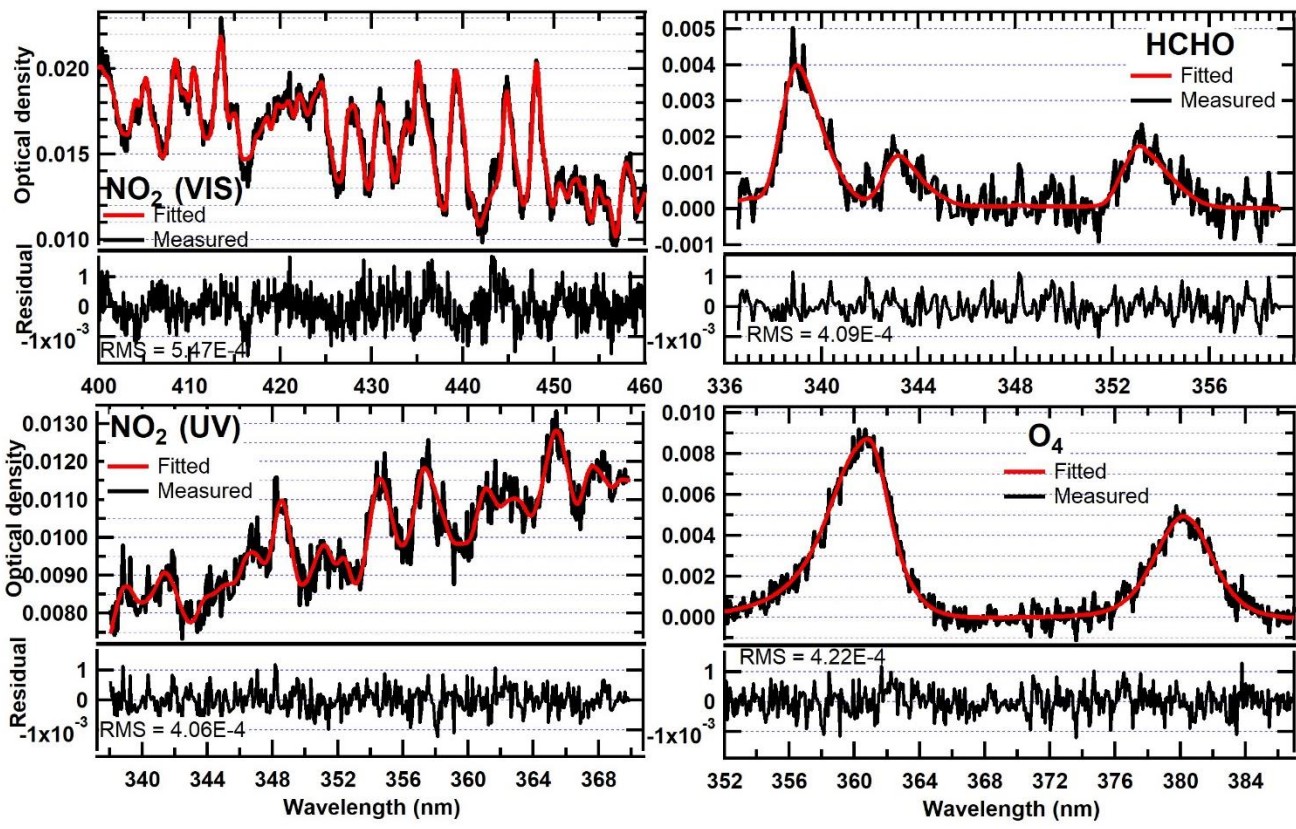

**Figure 2: Example DOAS fits and residuals for NO₂ in the Visible (dSCD = 2.91×10¹⁶ ± 2.89×10¹⁴ molecules cm⁻²) and UV (dSCD = 2.09×10¹⁶ ± 8.17×10¹⁴ molecules cm⁻²), HCHO (dSCD = 7.23×10¹⁶ ± 4.32×10¹⁵ molecules cm⁻²) and O₄ (dSCD = 2.06×10⁴³ ± 4.25×10⁴¹ molecules² cm⁻⁵) for a typical spectrum measured on 26.07.2015 at a solar zenith angle of 20° and 2° elevation angle.**

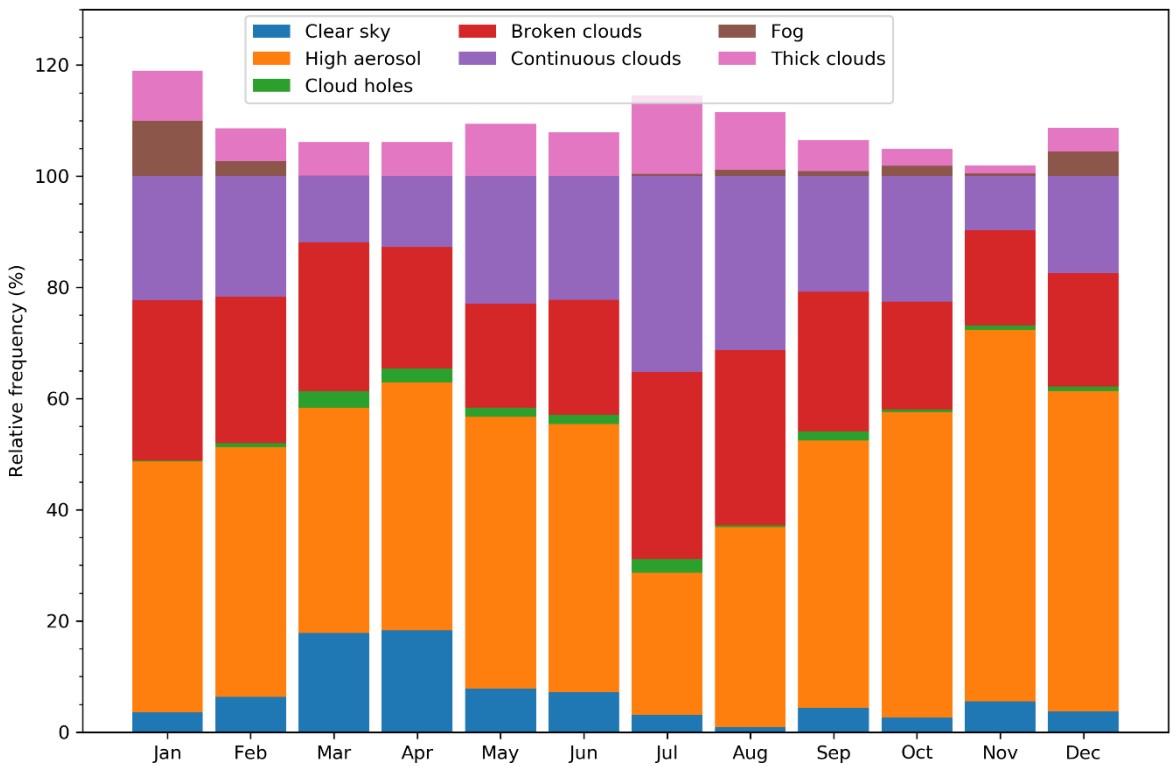

**Figure 3: Relative frequencies of occurrence of various sky conditions in different months of the year over Mohali as derived from 4.5 years of MAX-DOAS observations. Note that the secondary cloud classifications of fog and optically thick clouds are not mutually independent and exclusive of the primary classification (clear sky, clear sky with high aerosol load, broken clouds, cloud holes and continuous clouds). Hence, these are shown separately above the 100% mark.**

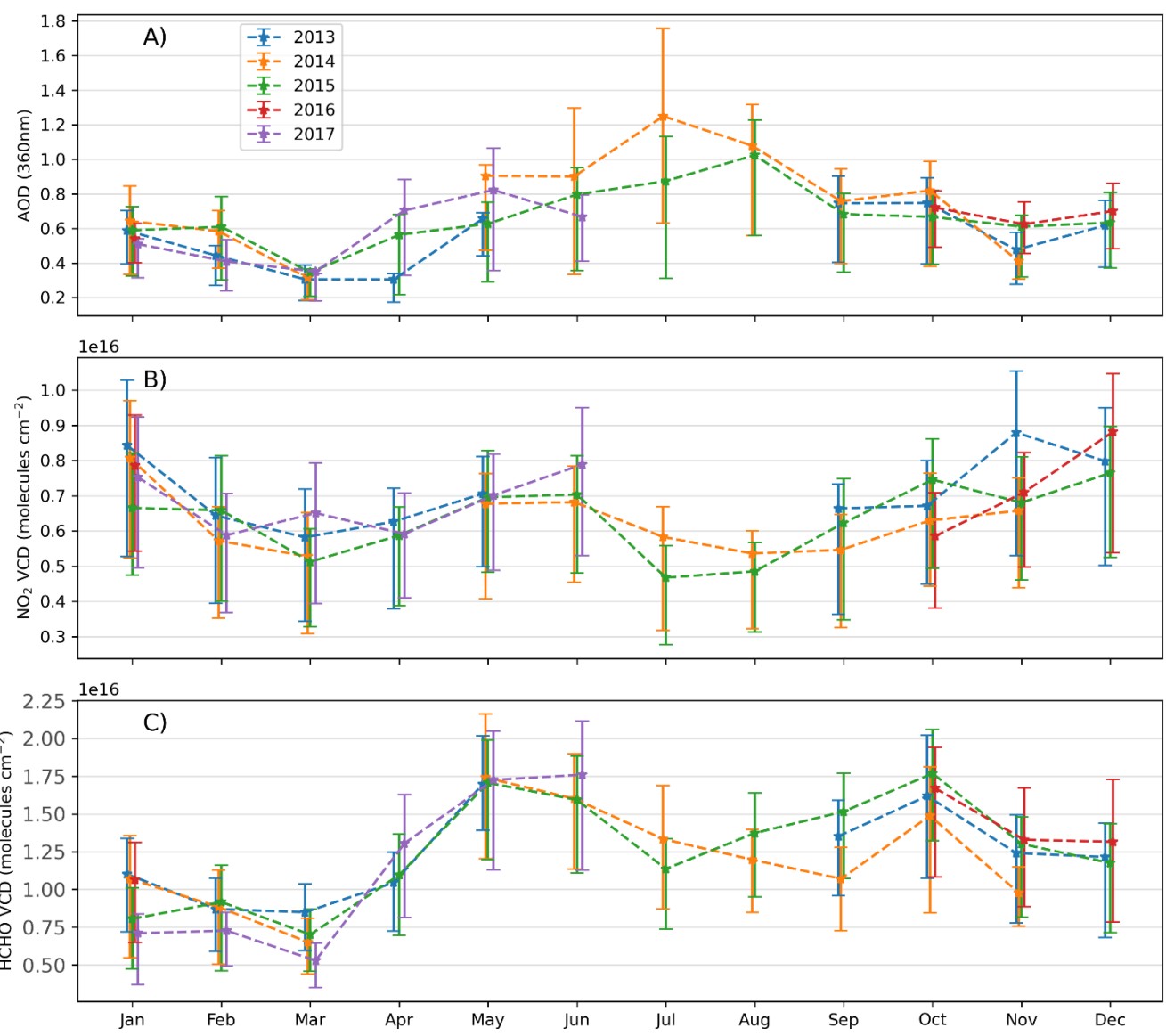


**Figure 4: Monthly and annual variation of (A) AOD, (B) NO₂ and (C) HCHO vertical column densities derived from MAX-DOAS measurements. The lower and upper vertical error bars represent the 25th and 75th percentiles, respectively.**

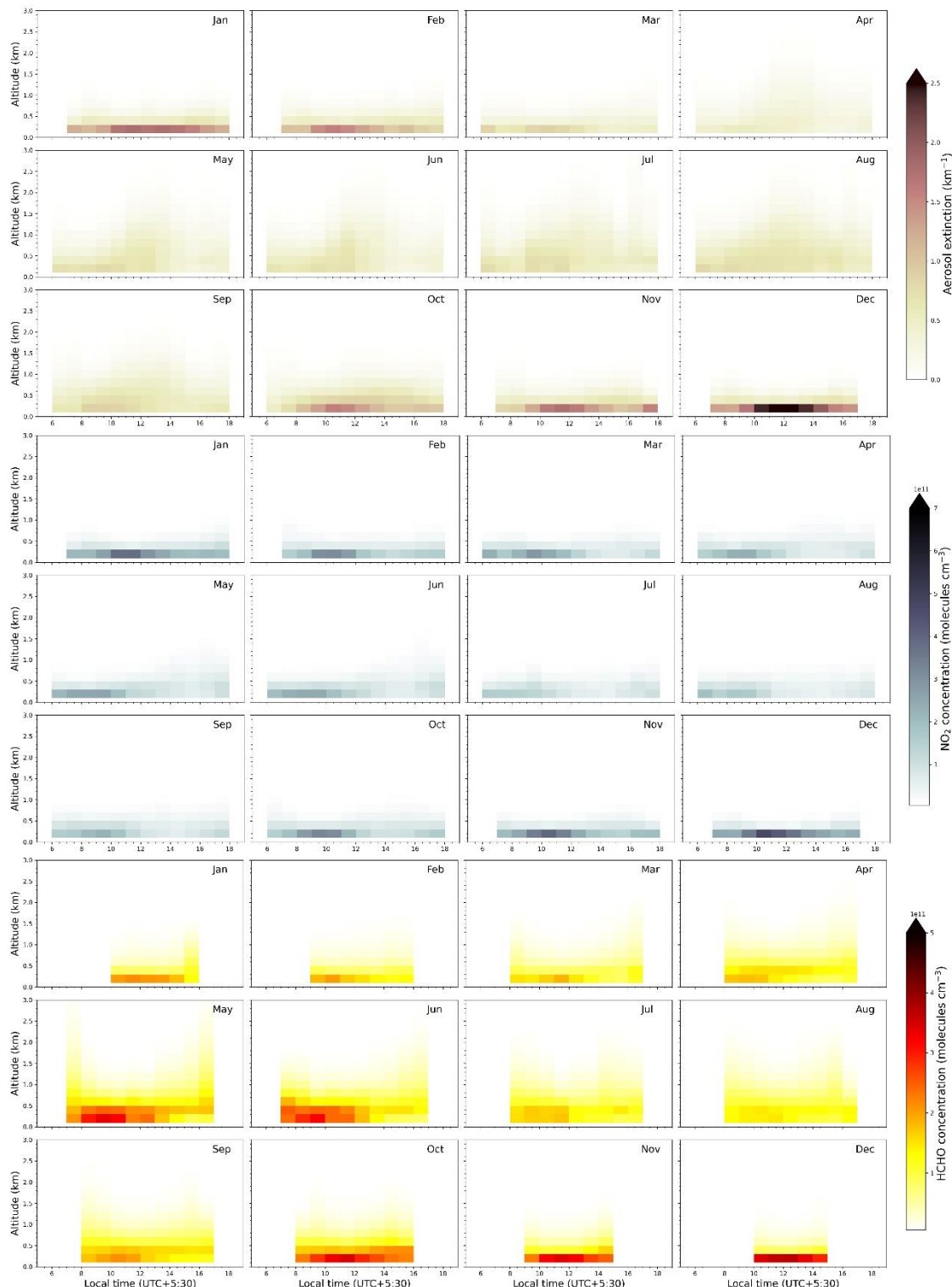

**Figure 5: Hourly mean vertical profiles of aerosol extinction (top 3 rows, shades of olive), NO₂ concentrations (middle three panels, shades of blue) and HCHO concentrations (bottom three panels, shades of red) in different months retrieved from 4.5 years of MAX-DOAS measurements over Mohali.**

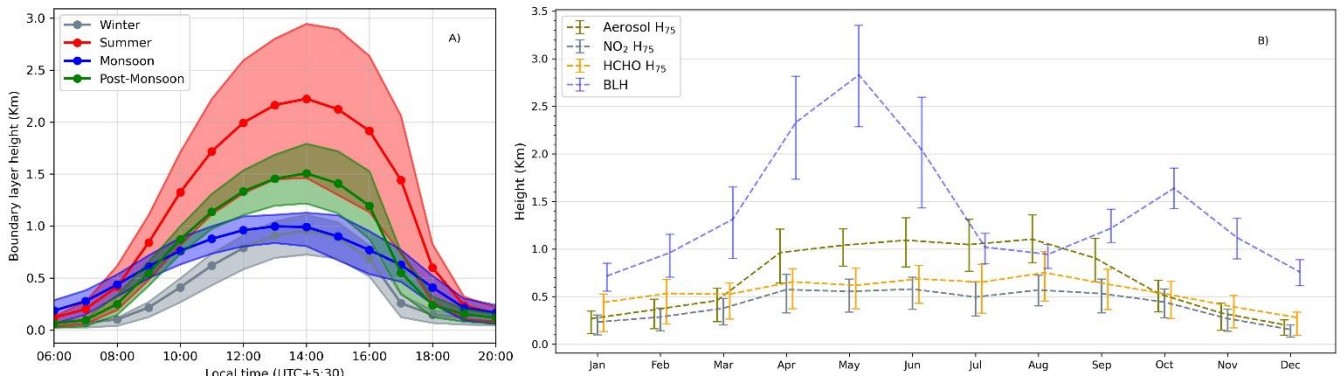


**Figure 6: A) Diurnal evolution of the hourly means ERA5 boundary layer height (BLH) at Mohali for the four major seasons of the year. B) Mean afternoon time (between 12:00 and 15:00 Local time) profile height (with 75% of the total amount below) for aerosols, NO₂ and HCHO, and the ERA5 BLH for different months. The upper and lower vertical error bars represent the monthly variability as 75th and 25th percentiles, respectively.**

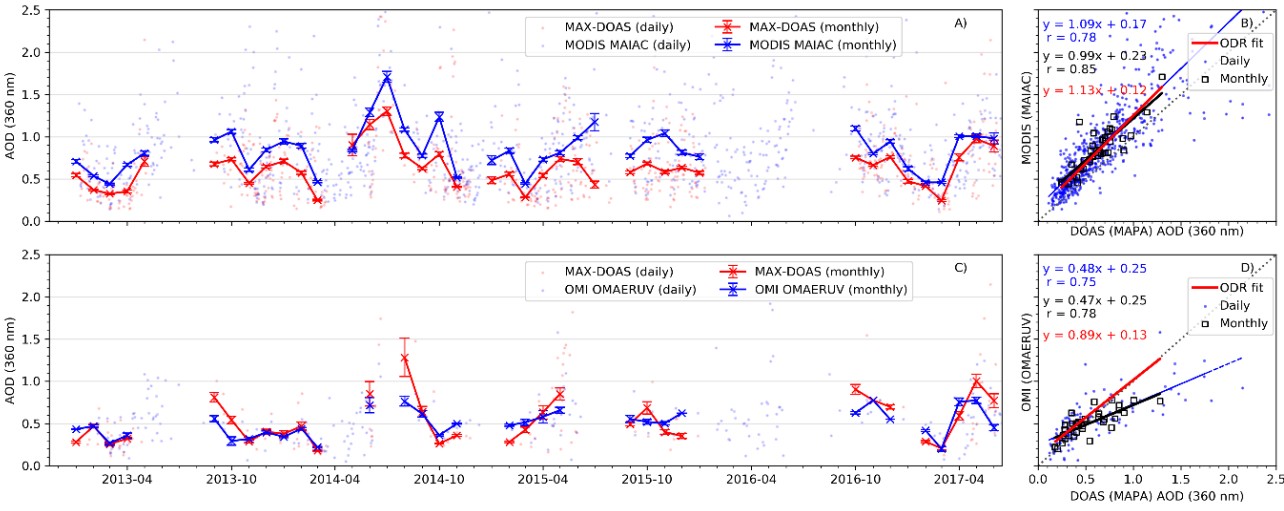


**Figure 7: Intercomparison of daily (dots) and monthly mean (lines and markers) AOD at 360nm retrieved from ground-based MAX-DOAS measurements and from the MODIS MAIAC data product (top panel) and OMI AERUV data product (bottom panel). The monthly mean of the MAX-DOAS and satellite data products were calculated by considering only the days of the month when both the measurements were available, causing different MAX-DOAS means in A) and C).**


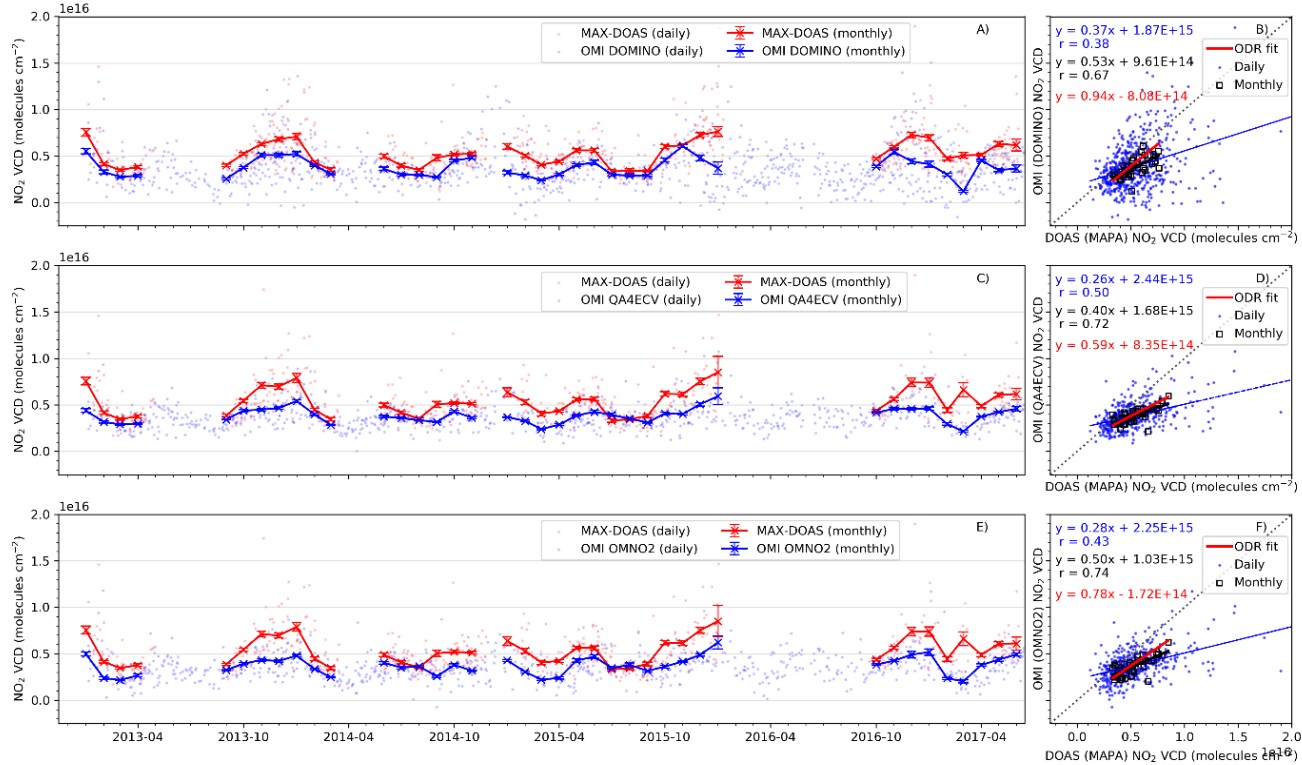

**Figure 8: Intercomparison of time series of daily (dots) and monthly mean (lines and markers) NO₂ VCDs retrieved from ground-based MAX-DOAS measurements and the OMI DOMINO data product (A), the OMI QA4ECV data product (B) and the OMI OMNO2 data product (C). The vertical error bars represent the monthly variability as the standard error of the mean. The monthly mean of the MAX-DOAS and satellite data products were calculated by considering only the days of the month when both the**
**measurements were available, causing different MAX-DOAS means in A), C) and E). Scatter plots (panels B, D and F for DOMINO, QA4ECV and OMNO2 respectively) using the daily and monthly mean values are shown adjacent to the time series.**

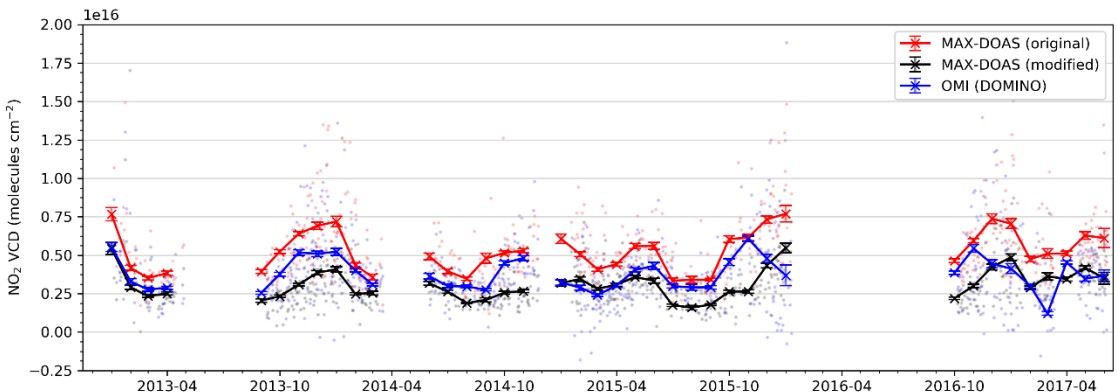

**Figure 9: Time series of daily (dots) and monthly means (lines and markers) of MAX-DOAS NO₂ VCDs, OMI DOMINO NO₂ VCDs and modified MAX-DOAS VCDs modified by using the DOMINO averaging kernels and a priori profiles.**

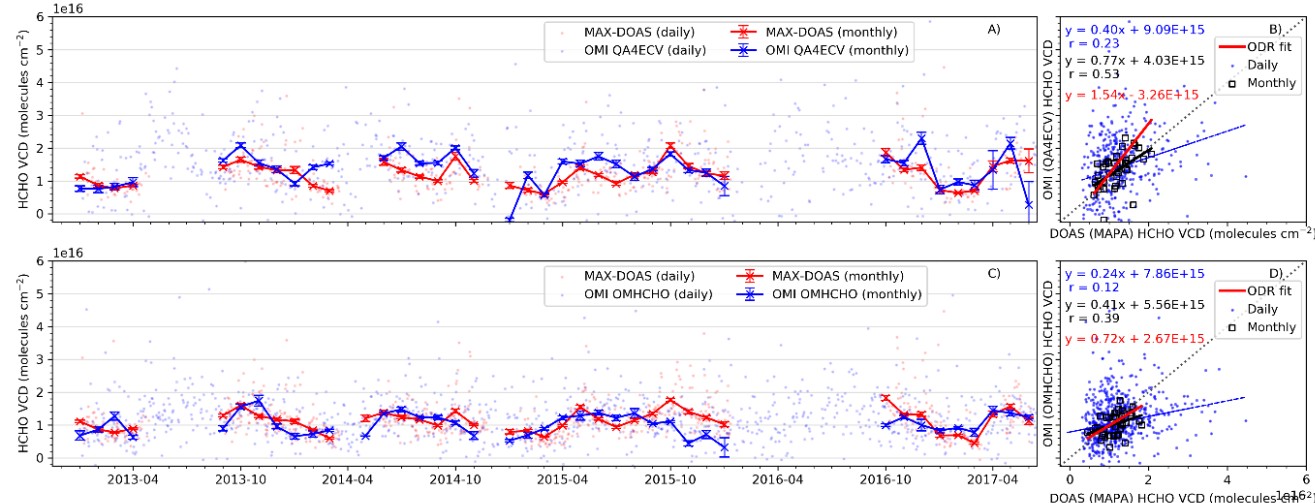

**Figure 10: Intercomparison of time series of daily (dots) and monthly mean (lines and markers) HCHO VCDs retrieved from ground-based MAX-DOAS measurements and the OMI QA4ECV data product (A) and the OMI OMHCHO data product (C). The respective vertical error bars represent the 1σ monthly variability as the standard error of the mean. The monthly means of the MAX-DOAS and satellite data products were calculated by considering only the days of the month when both the measurements were available, causing different MAX-DOAS means in A) and C). Scatter plots (B and D) using the daily and monthly mean values are shown adjacent to the time series.**

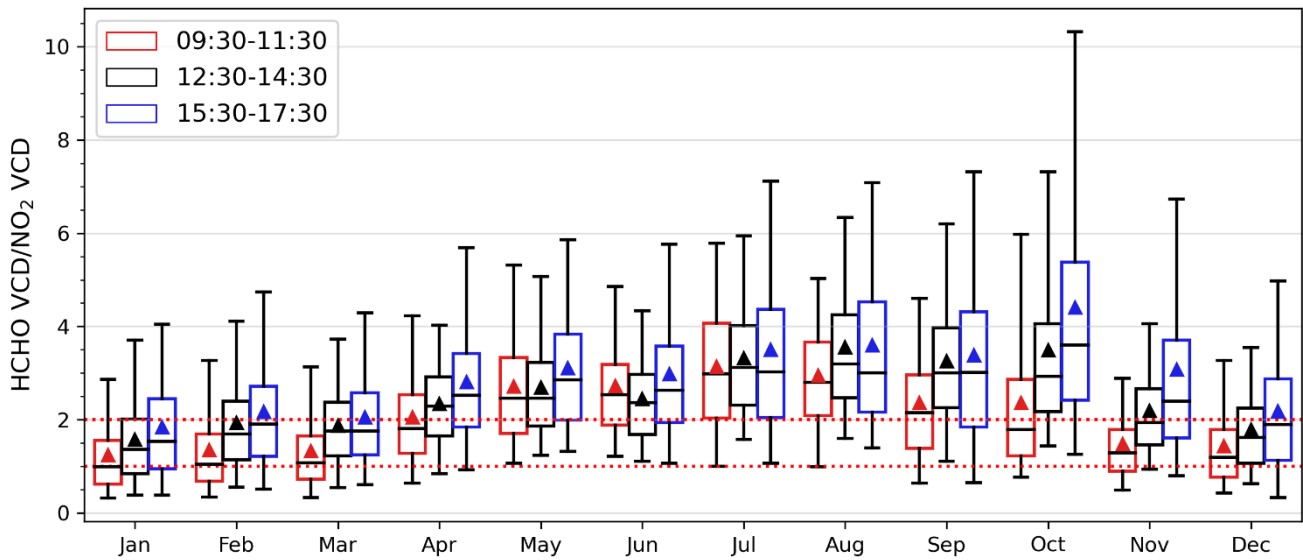

**Figure 11: Monthly mean HCHO VCD/NO₂ VCD ratios (triangles) calculated from MAX-DOAS measurements for the morning (09:30-11:30 L.T., red), noon around the OMI overpass time (12:30-14:30 L.T., black) and late afternoon (15:30-17:30 L.T., blue) over Mohali. The lines at the centres of the boxes represent the median; the boxes show the interquartile ranges whereas the whiskers show the 5th and 95th percentile values.**

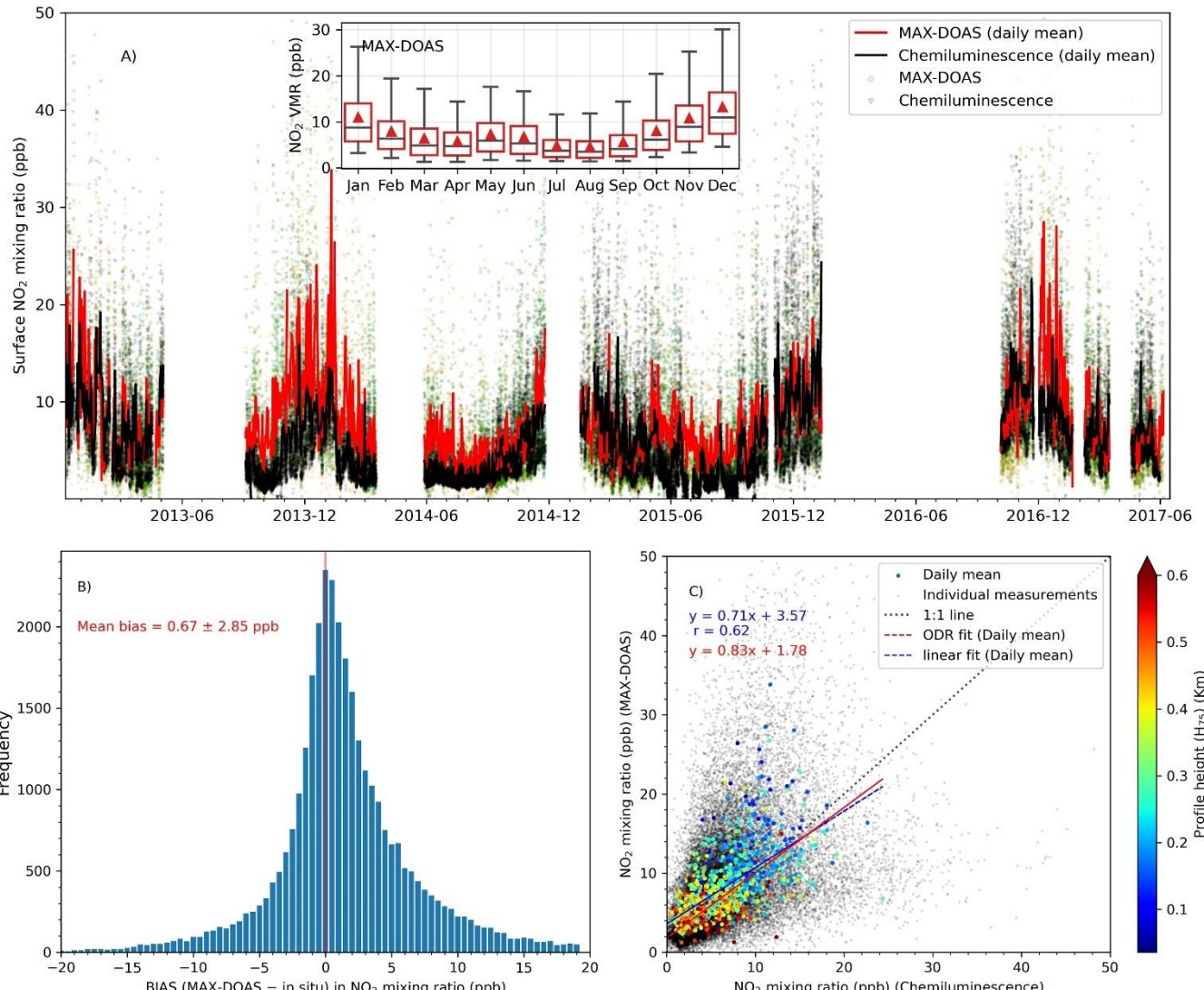

Figure 12: A: Daily mean NO₂ mixing ratios measured using an *in situ* chemiluminescence analyser (black line) and average mixing ratio in the lowest layer (0-200m) retrieved by the profile inversion of MAX-DOAS measurements (red line). The green and orange dots in the background show the mixing ratios corresponding to the individual MAX-DOAS elevation sequences flagged as valid and warning respectively. The inset of panel A shows the monthly variability of the MAX-DOAS surface NO₂ VMR as a box and whiskers plot. Panel B shows the frequency distribution of the BIAS (MAX-DOAS – *in situ*) for the individual surface VMR measurements. Panel C shows the scatter plot between the daily mean surface NO₂ VMR from MAX-DOAS and *in situ* analyser colour-coded according to the profile height (H₇₅). The individual measurements are shown as black dots in the background of panel C.

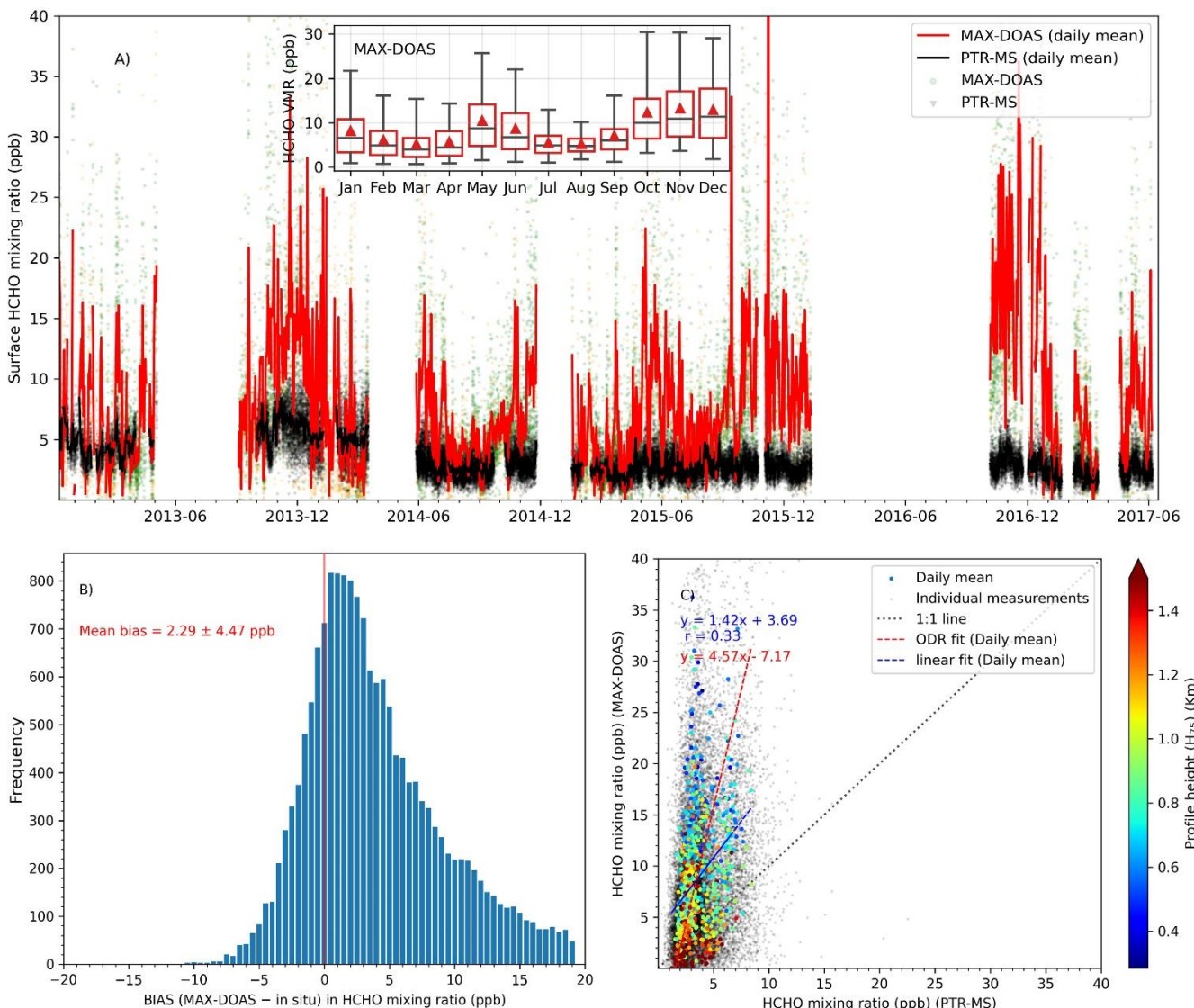

Figure 13: A: Daily mean HCHO mixing ratios measured using an *in situ* chemiluminescence analyser (black line) and average mixing ratio in the lowest layer (0-200m) retrieved by the profile inversion of MAX-DOAS measurements (red line). The green and orange dots in the background show the mixing ratios corresponding to the individual MAX-DOAS elevation sequences flagged as valid and warning respectively. The inset of panel A shows the monthly variability of the MAX-DOAS surface HCHO VMR as a box and whiskers plot. Panel B shows the frequency distribution of the BIAS (MAX-DOAS – *in situ*) for the individual surface VMR measurements. Panel C shows the scatter plot between the daily mean surface HCHO VMR from MAX-DOAS and PTR-MS colour-coded according to the profile height ($H_{75}$). The individual measurements are shown as black dots in the background of panel C.

**Tables**

**Table 1: Spectral analysis settings and considered cross-sections in QDOAS for retrieval of dSCDs of $NO_2$ (in UV and Vis), HCHO and $O_4$**

| Species | NO$_2$ (UV) | NO$_2$ (Vis) | HCHO | O$_4$ |
|---|---|---|---|---|
| Fit window | 338-370 nm | 400-460 nm | 336.5-359 nm | 352-387 nm |
| Fitted absorption cross sections | NO$_2$ (298 K with I$_o$ correction corresponding to a SCD of $10^{17}$ molecules cm$^{-2}$)[1], NO$_2$ (220K pre-orthogonalized to NO$_2$ cross section at 298K)[1], HCHO (297 K)[2], O$_3$ (223K with I$_o$ correction corresponding to a SCD of $10^{20}$ molecules cm$^{-2}$)[3], BrO (223K)[4], O$_4$ (293K)[5], Ring | NO$_2$ (298 K with I$_o$ correction corresponding to a SCD of $10^{17}$ molecules cm$^{-2}$)[1], H$_2$O(296K)[6], O$_3$ (223K with I$_o$ correction corresponding to a SCD of $10^{20}$ molecules cm$^{-2}$)[3], O$_4$ (293K)[5], Ring | HCHO (297 K)[2], BrO (223K)[4], O$_3$ (223K with I$_o$ correction corresponding to a SCD of $10^{20}$ molecules cm$^{-2}$)[3], NO$_2$ (298 K with I$_o$ correction corresponding to a SCD of $10^{17}$ molecules cm$^{-2}$)[1], O$_4$ (293K)[5], Ring | O$_4$ (293K)[5], NO$_2$ (298 K with I$_o$ correction corresponding to a SCD of $10^{17}$ molecules cm$^{-2}$)[1], HCHO (297 K)[2], O$_3$ (223K with I$_o$ correction corresponding to a SCD of $10^{20}$ molecules cm$^{-2}$)[3], BrO (223K)[4], Ring |
| Polynomial order | 5 | 5 | 5 | 5 |
| Intensity offset | Constant and first order | Constant and first order | Constant and first order | Constant, first, and second order |
| Fraunhofer reference selection | Sequential | Sequential | Sequential | Sequential |
| Shift and stretch | Spectrum | Spectrum | Spectrum | Spectrum |

[1](Vandaele et al., 1998), [2](Meller and Moortgat, 2000), [3](Serdyuchenko et al., 2014), [4](Fleischmann et al., 2004), [5](Thalman and Volkamer, 2013), [6](Rothman et al., 2010)

**Table 2: Frequency of MAPA retrievals flagged as "valid", "warning" or "error" for various species and different sky conditions.**

| | All-sky conditions (n = 63107) | | | | Condition except for thick clouds and fog (n = 58209) | | | |
|---|---|---|---|---|---|---|---|---|
| | O$_4$ | NO$_2$ (UV)* | NO$_2$ (VIS) | HCHO | O$_4$ | NO$_2$ (UV)* | NO$_2$ (VIS) | HCHO |
| Valid | 27530 | 24287 | 24981 | 21671 | 27096 | 23899 | 24637 | 21462 |
| Warning | 10281 | 12893 | 12634 | 15491 | 9699 | 12304 | 11981 | 14758 |
| Error | 25296 | 25927 | 25492 | 25945 | 21414 | 22006 | 21591 | 21989 |

* The number of valid retrievals for NO$_2$ (UV) is calculated using relaxed RMS flag criteria (Appendix A)

**Appendix**

**A Performance of the NO₂ retrieval in the UV and Visible spectral range and comparison of the NO₂ VCD obtained from the profile retrievals to the results from the geometric approximation**

The DOAS spectral analysis and profile inversion have been performed both in the UV and Visible spectral ranges. As mentioned in section 2.2 of the main text and also shown in Fig. A1 for a smaller subset of data from 2015, the analysis in the UV results in larger fit uncertainties, which eventually leads to a poorer performance of the profile inversion.

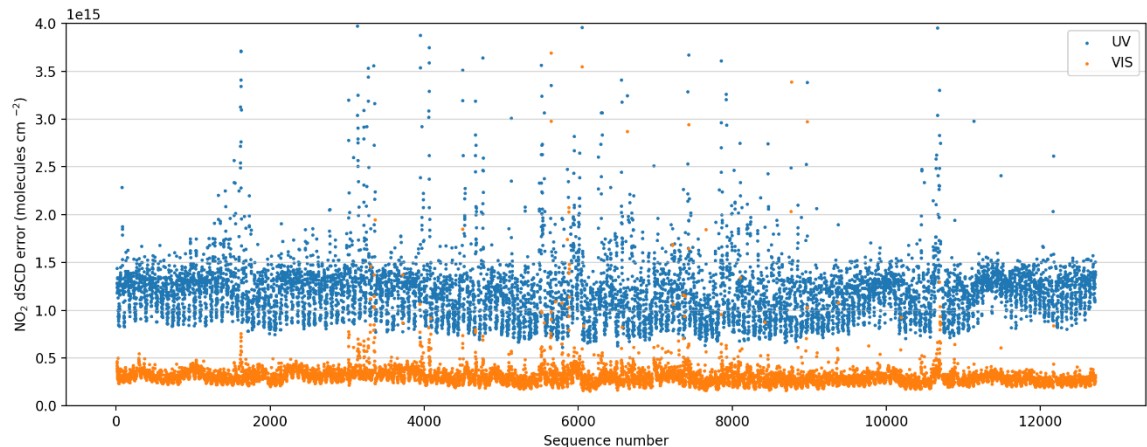

**Figure A1: DOAS fit errors of the NO₂ retrievals in the UV and Visible wavelength windows.**

For an elevation angle sequence, MAPA finds the best fit of the model dSCDS by minimizing the difference compared to the measured dSCDs. The difference is quantified by the RMS ($R$), which is defined as:

$$R = \sqrt{\frac{(S_{fm} - S_{ms})^2}{N_{eas}}} \qquad\qquad \text{A1}$$

Here $S_{fm}$ and $S_{ms}$ represent dSCDs sequences from model and measurements, and $N_{eas}$ represents the number of elevation angles in the measurement sequence. Warning and error flags are raised for the retrieved profile if the following two conditions are fulfilled:

1. $R_{bm}$ (RMS for the best matching model dSCD) exceeds the sequence median dSCD uncertainty.
2. $R_n$ (the ratio of $R_{bm}$ and maximum dSCD of a sequence) exceeds the predefined thresholds (Beirle et al., 2019).

The larger fit uncertainties of the NO₂ analysis in the UV directly result in larger errors in the NO₂ dSCDS. These larger errors lead to larger residuals and hence larger $R_{bm}$, leading to more flagged sequences. Here we check if the threshold for $R_n$ can be relaxed from that (0.05) recommended by Beirle et al. (2019) for the NO₂ retrieval in the UV while still retaining reasonable retrievals. We have calculated the total retrieval quality flag as described in section 2.8.6 of Beirle et al. (2019) by varying $R_n$ from 0.05 to 0.15. A larger $R_n$ leads to a larger number of retrievals being flagged as valid. The retrieved VCDs and concentrations in the layer closest to the surface derived in the UV for different $R_n$ values are compared to those in the Vis as

reference. We find that increasing $R_n$ from 0.05 to 0.15 for the UV retrieval almost doubles the number of valid retrievals while keeping similar statistical agreement as compared to Vis retrievals. From Fig. A2, we see that the slope of the linear regression of the NO2 VCDs between UV and Vis remains close to 0.95, while the correlation coefficient (r) only changes from 0.91 to 0.89. Similar results are obtained for the surface concentration, where the slope of the linear regression and correlation coefficient (r) only slightly change from 0.87 and 0.94 to 0.85 and 0.93, respectively.

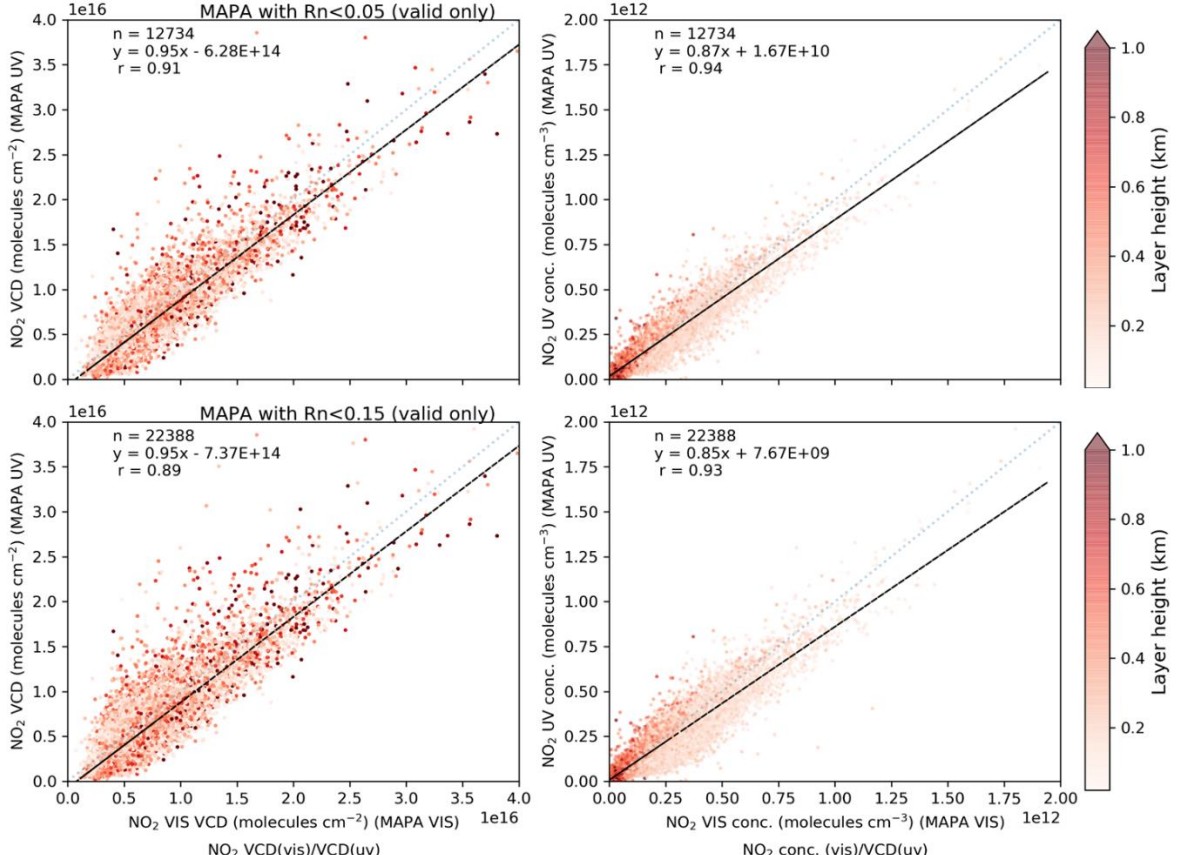

**Figure A2: Comparison of NO2 VCDs (left) and surface concentrations (right) retrieved in the UV against those retrieved in the Vis. Please note that the number of valid data points shown in this figure corresponds to the number of valid overlapping retrievals in UV and Vis. The number of valid retrievals for NO2 in UV shown in Table 2 correspond to $R_n$ flag with threshold <0.15.**

For several years, the geometric approximation has been used for the calculation of VCDs from the dSCDs retrieved from DOAS spectral analysis. If only the photon light path is taken into account with the assumption that the last scattering event before the photon reached the telescope of the instrument, has happened above the trace gas layer, the geometric airmass factors for an elevation angle α can be calculated trigonometrically:

$$AMF_{\alpha,Geo} = \frac{1}{\sin \alpha} \qquad\qquad A2$$

Since the geometric approximation relies on the single scattering approximation above the trace gas layer, it is expected to work well under clear sky conditions with low aerosol loads and for trace gases confined to layers close to the surface. Generally, there is a trade-off between the sensitivity and validity of geometric approximation for the choice of elevation angle. At low elevation angles, though the measurements are more sensitive, the probability of scattering within the trace gas layer is rather high. We have chosen an elevation angle of 15° for the calculation of geometric VCDs. Several studies (Bösch,

2018;Wagner et al., 2011;Jin et al., 2016) have shown that the geometric VCD usually agree within 20% to the VCDs that are retrieved using radiative transfer simulations if the trace gas is not located at higher altitudes (>1000m) and trace gas is confined within the aerosol layer. In Fig. A3, we compare the VCDs calculated from the geometric approximation and from valid MAPA retrievals for various sky conditions, both for $NO_2$ in the UV and visible and for HCHO (in the UV).

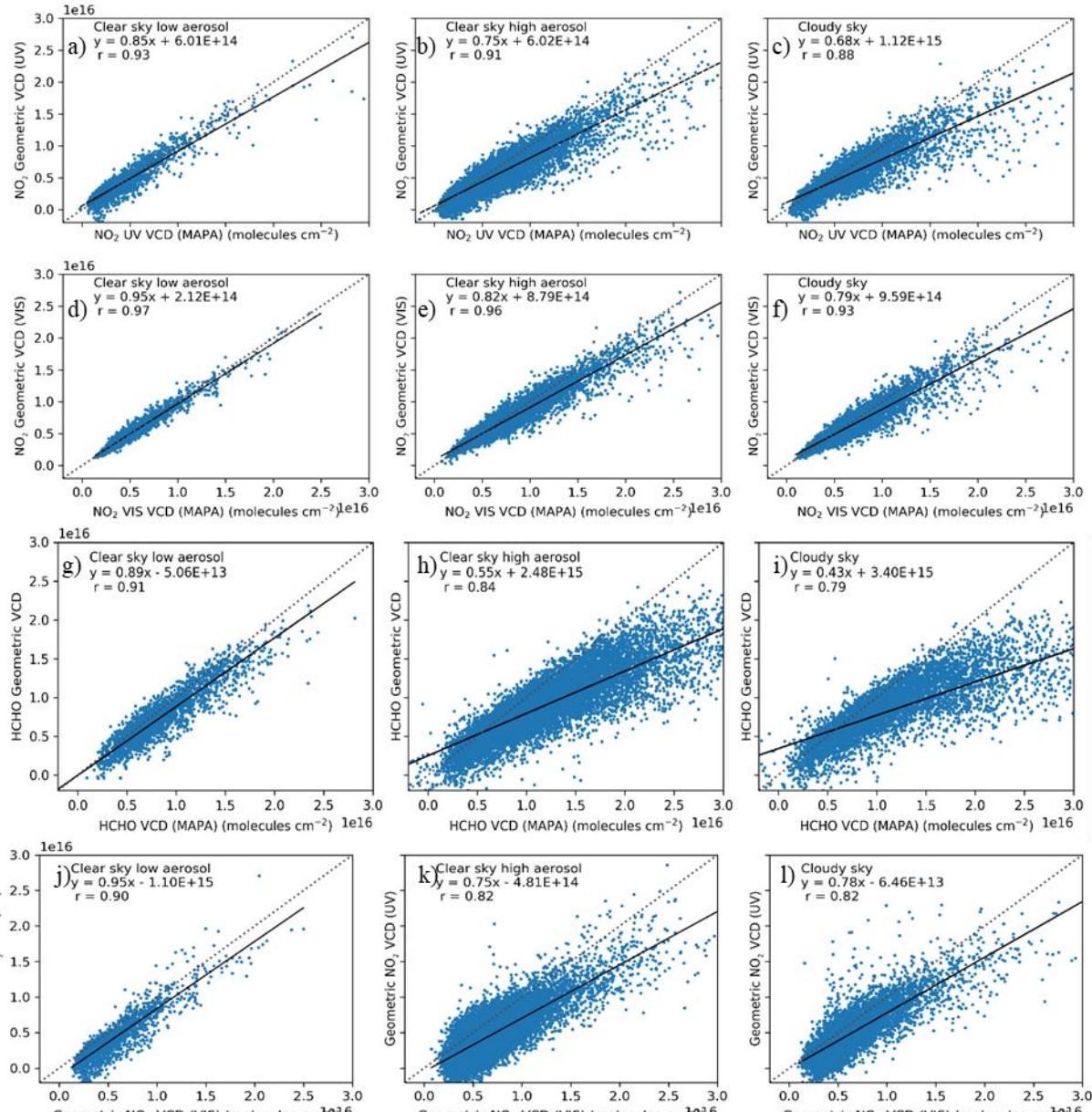

**Figure A3: Comparison of VCDs calculated using the geometric approximation at 15° elevation angle and that calculated using MAPA for NO₂ in the UV (top panel), NO₂ in the VIS (second panel) and HCHO (third panel) for various sky conditions: clear sky with low aerosol load (left), clear sky with high aerosol load (middle) and cloudy sky conditions except thick clouds and fog (right). The bottom panel shows the comparison of NO₂ VCD calculated using the geometric approximation for spectral analyses in the UV and Visible for the three different sky conditions.**

As expected, we see an excellent agreement between the NO₂ geometric VCDs and those retrieved from the profile inversion in clear sky conditions with low aerosol load both for the UV and visible retrievals. The level of agreement is still reasonable

in high aerosol condition and cloudy cases, however, for such conditions, the geometric VCDs are systematically lower. This finding is in line with that observed by Wagner et al. (2011). It indicates the effect of photon scattering within the trace gas layer. We also observe that the agreement of $NO_2$ VCDs between geometric approximation and profile inversion is better for the UV than for the Visible retrieval. If we chose 30° elevation angle for the calculation of the geometric VCDs, reasonable agreement (r>0.9) with the VCDs retrieved from profile inversion is observed only for the visible retrievals. For the $NO_2$ retrievals in the UV, reasonable agreement (r=0.86) was observed for clear sky with low aerosol load, but a larger scatter (r<0.8) is found for clear sky conditions with high aerosol load and for cloudy sky conditions.

Similar to $NO_2$, we also observe the best agreement for the HCHO VCDs retrieved using the geometric approximation and those using the profile inversion for clear sky conditions with low aerosol load. For high aerosol loads and cloudy sky conditions, the VCDs from the geometric approximation are significantly lower. In contrast to $NO_2$, a significant fraction of formaldehyde is usually found in higher layers, which limits the validity of geometric approximation. The bias gets higher for HCHO VCDs greater than ~$1.5 \times 10^{16}$ molecules $cm^{-2}$. If we chose observations at 30° elevation angle for the calculation of the geometric VCDs, lower correlation with the VCDs retrieval using the profile inversion was observed (r<0.8) for all sky conditions.

We also checked the internal consistency of the $NO_2$ dSCDS retrieved from the DOAS analysis in the UV and Visible. For the comparison, even for geometric approximation, we only consider those sequences for which the profile retrieval was flagged valid. From Fig. A3 (panel J-L), we see that in clear sky cases, $NO_2$ VCDs from the geometric approximation in the visible are systematically higher than those retrieved in the UV. The Rayleigh scattering probability is inversely proportional to the fourth power of the wavelength, and therefore in the UV, the higher scattering probability results in shorter light paths within the trace gas layer. Since the AMFs from the geometric approximation are independent of the wavelength, the shorter dSCDs results in smaller VCDs from the geometric approximation in the UV. The radiative transfer models, however, account for different path length in UV and VIS. Hence, the VCDs retrieved using MAPA do not show any systematic bias in UV or VIS (Fig. A2). It is interesting to note that the difference of the $NO_2$ VCDs retrieved from the geometric approximation in the UV and visible is much smaller for cloudy sky conditions. For such conditions, the differences of the light path lengths in the UV and visible are usually much smaller than for clear sky conditions.

**B Identification of thick clouds**

For the identification of thick clouds, Wagner et al. (2016) proposed the use of an absolute calibration method for the $O_4$ absorption in the Fraunhofer reference spectrum. However, this method can only be applied for relatively short time periods, over which the spectral properties of the MAX-DOAS instrument stay almost the same. For the measurements used in this study, this method cannot easily be applied, because the spectral properties of the Mini-MAX-DOAS instruments are known to vary rather strongly even within rather short periods of time (a few days to weeks). In addition, for the measurements used in this study, the detector temperature had to be changed frequently according to the seasonal variation of the ambient temperature. Therefore, it was impossible to create a consistent time series of absolutely calibrated measured $O_4$ absorption

over the complete measurement period. Therefore, the identification of thick clouds was performed based on the measured radiances (Wagner et al., 2014;Wang et al., 2015)

While optically thin clouds often lead to an increase in the measured radiance, optically thick clouds lead to a strong decrease (compared to clear sky radiance). However, such an approach can only be applied if there is no significant degradation in the measured radiance over the whole measurement period. Fig. B1 shows the time series of the measured radiances (at 360nm) for a SZA interval between 55° and 60°. A significant decrease is not observed in the measured radiance during the more than four years measurement period, albeit the intra-annual seasonal variation. Hence, we can use the measured radiance for the identification of thick clouds. Please note that the strongly increased values of the radiances are caused by optically thin clouds. Note that the linear trend was fitted only for the period from 01-01-2013 to 31-12-2016 to minimise the impact of the seasonal variation.

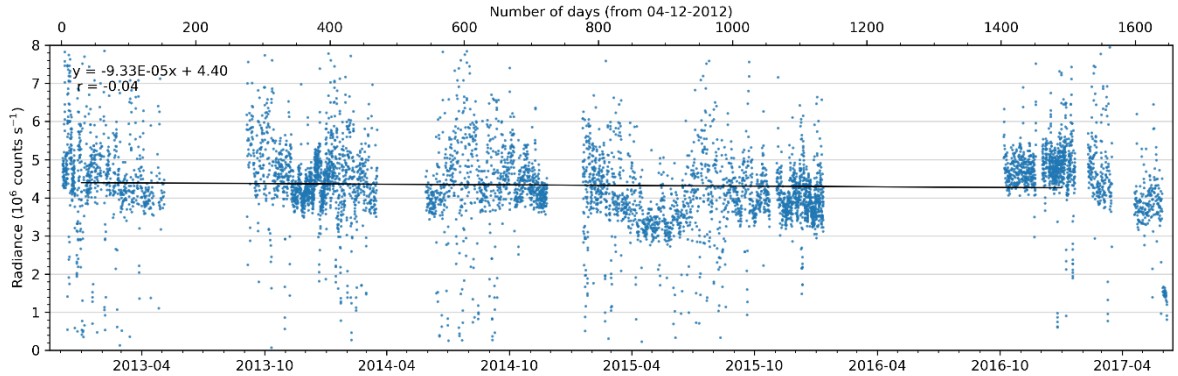

**Figure B1: Time series of the measured radiances at 360nm derived from the MAX-DOAS measurements in the zenith viewing direction for SZA between 55° and 60°. Please note that the trend is calculated only for the measurements from 01-01-2013 to 31-12-2016 to minimize the effect of seasonal variation. The dark black line indicates the linear regression of the measured radiance as a function of day number.**

In order to identify the optically thick clouds, the measured radiances in the zenith direction have to be compared to a SZA-dependent reference value (see Wagner et al. (2014) and Wagner et al. (2016)), which is obtained from RTM simulations for a well defined atmospheric scenario. However, as these instruments are usually not radiometrically calibrated, the measured radiances cannot be directly compared to the reference radiances derived from the RTMs. Hence, we first have to perform a calibration of measured radiances.

For the calibration, we first calculated the radiance in the zenith direction at 360 nm for an SZA range between 7° and 95° using the RTM McARTIM (Deutschmann et al., 2011). The simulations are done for an AOD = 0.3, aerosol particles properties as proposed by Wagner et al. (2014) and a surface albedo of 5%. In the next step, two clear days in summer (such that the measurements are available even at solar zenith angles less than 10°) with AOD (at 360 nm) close to 0.3 (as measured by MODIS) are selected. By comparing the SZA-dependence of the measured radiances with the simulated radiances (Fig. B2), we determined a calibration factor to be $1.48 \times 10^{-8}$ counts$^{-1}$ for our measurements. Here it should be noted that from the similar

relative SZA dependences of the measured and simulated radiances at larger SZA we can conclude that the AOD at 360 nm was similar to 0.3.

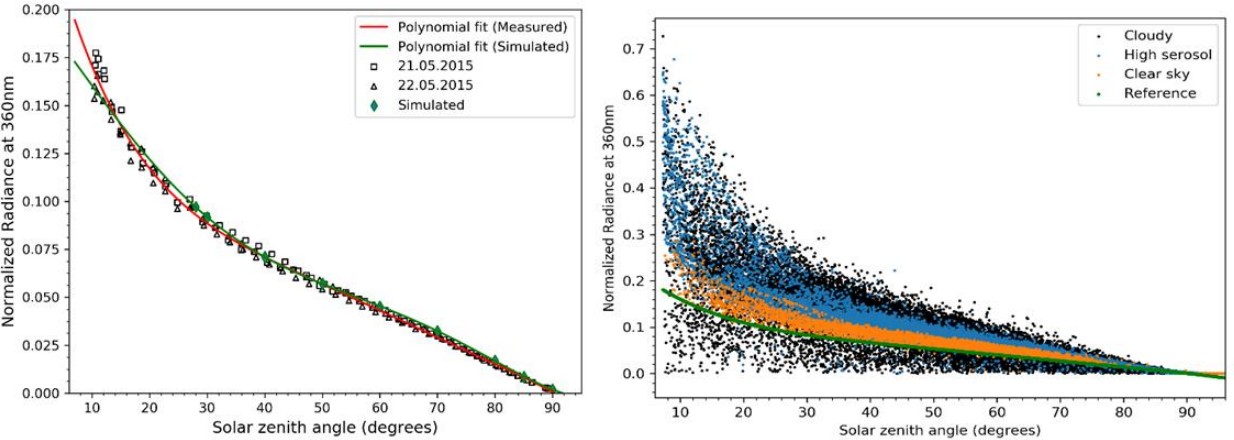

**Figure B2: Dependence of the measured and simulated radiances in zenith direction (at 360nm) on the solar zenith angle**
**for the two selected days (left). The right panel shows all measured normalized radiances (after calibration) in zenith direction colour coded for three sky conditions: black: cloudy sky, orange: clear sky with low aerosol load, blue: clear sky with high aerosol load. All measurements corresponding to thick clouds lie below the threshold (shown as green curve) in the right panel.**

Fig. B2 (left panel) shows the variation of the calibrated measured radiances on the two selected days and the polynomial fit
of the simulated radiances as a function of the SZA. It should be noted that in contrast to Wang et al. (2015), we have observed good agreement between the measured and simulated radiances even for solar zenith angles less than 40°. In the study by Wang et al. (2015) in Wuxi, the disagreement was attributed to a possible deviation from the parametrised Henyey–Greenstein aerosol phase function. For Mohali, a better agreement is probably related to the presence of a different aerosol type compared to that in Wuxi. A single fifth-order polynomial was fitted to the calibrated measured radiances in the complete SZA range to
derive a SZA dependent function of the clear sky normalized radiance, which is subsequently used as SZA-dependent reference.

For the identification of thick clouds, we have used a threshold of 0.94 times the SZA-dependent threshold radiance similar to Wang et al. (2015). In order to check the consistency of the thick cloud identification using the normalized radiances, we have also performed thick cloud identification based on the $O_4$ AMFs as described by Wagner et al. (2016) for a small period in
July 2014. Figure B3 shows an excellent consistency of the thick cloud identification using the two methods. On the top panel, the periods having smaller radiances than the threshold show larger $O_4$ AMFs than the respective threshold in the bottom panel.

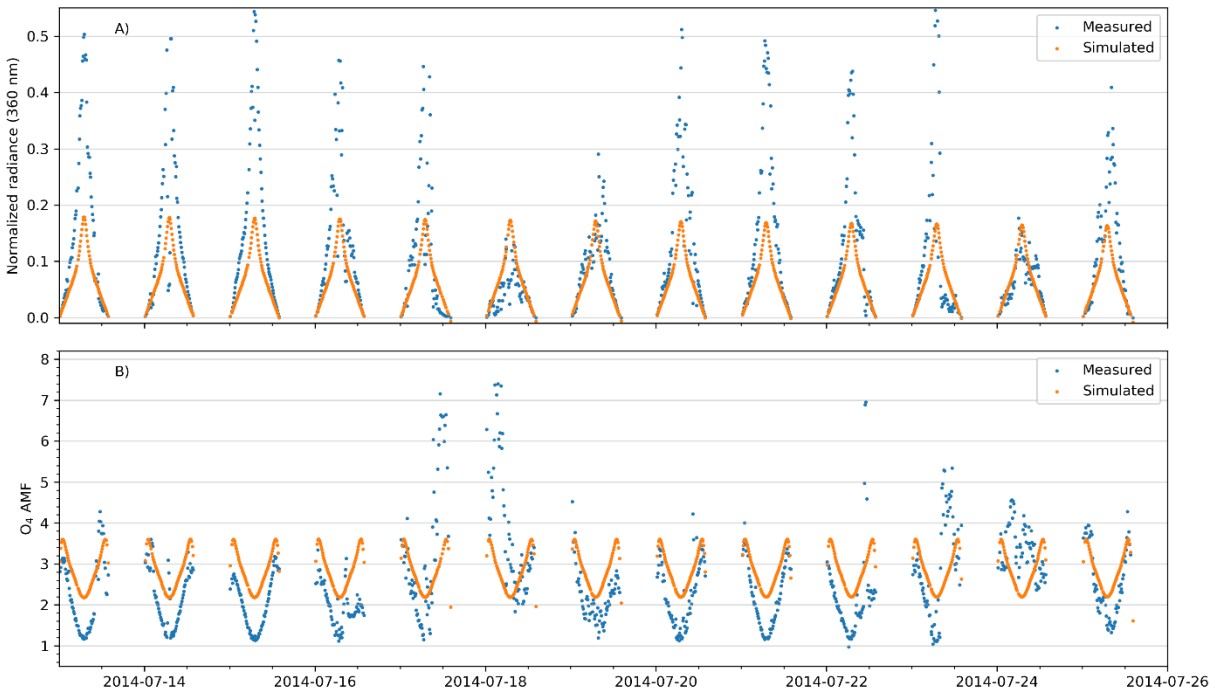

**Figure B3 A: Measured calibrated radiances in the zenith direction and B: O₄ AMFs derived for zenith direction for a selected short period of the measurements, over which the instrument properties were almost constant. Also shown are the simulation results for the SZA-dependent thresholds. On some days largely enhanced O4 AMFs are found indicating the presence of optically thick clouds. On these days, also largely decreased radiances are measured.**

## C Satellite data products used for comparison against MAX-DOAS measurements

**OMAERUV:** The OMAERUV algorithm uses the Lambert Equivalent Reflectivity (LER) calculated from the measured radiance at 388 nm to first yield the AOD and aerosol absorption optical depth (AAOD)at 388 nm. Subsequently, using an inherent aerosol model, the AOD at 354 nm and 500 nm are calculated (Torres et al., 2007). The OMAERUV V1.8.9.1 product used in this study also provides an indicator for the data quality named "FinalAlgorithmFlags" corresponding to every measurement. For the intercomparison in this study, we have only retained the data corresponding to a "FinalAlgorithmFlags" value of 0, which represent the most reliable retrievals of AOD.

**DOMINO V2:** The DOMINO version 2.0 NO₂ product (Boersma et al., 2011) makes use of the measurements in the 405-465 nm wavelength interval. The NO₂ slant column density (SCD) retrieved by employing the DOAS technique in the first step is separated for stratospheric and tropospheric composition. The DOAS fit includes the absorptions due to NO₂, O₃, O₄, H₂O (l), H₂O (g) and the Ring effect. Subsequently, the tropospheric SCD is converted to the tropospheric VCD using the tropospheric airmass factors. Both the calculation of the airmass factor using a radiative transfer model and the estimation of the stratospheric NO₂ uses the NO₂ fields from a $3° \times 2°$ spatial resolution global chemistry transport model TM4.

**OMNO2 v 3.0:** The OMNO2 algorithm developed by NASA (Marchenko et al., 2015) uses the 402-465 nm fit window for the $NO_2$ retrieval. In the first step, the wavelength calibration and Ring correction are performed in 7 sub-windows within the spectral range. The DOAS fit involves an iterative process, in which first a preliminary estimate of $NO_2$, $H_2O$ and glyoxal is made using fits in smaller sub-windows of the full spectral range of 402-465nm. The absorption due to this initial estimate is used for the determination of instrumental noise in the original spectrum. The noise corrected spectrum, in the next stage is again subjected to a DOAS fit including the absorptions from $NO_2$, $H_2O(g)$, glyoxal and the Ring effect in the 402-465nm spectral range to retrieve the $NO_2$ SCDs. A stratospheric correction is applied, and airmass factors are applied to retrieve the tropospheric $NO_2$ VCDs. The $NO_2$ a priori vertical profiles used for calculation of the AMF are taken from the climatology of the 4 year-long global model initiative (GMI) $2° \times 2.5°$ spatial resolution chemistry transport model simulation. The cloud information is incorporated from the external OMCLDO2 product, which calculates the cloud fraction using the contrast in measured radiances between clear and cloudy pixels. The cloud pressure (an indicator of cloud height) is calculated using the $O_4$ absorption at 477nm.

**QA4ECV:** In this work, we also use the OMI QA4ECV (quality assurance for essential climate variables) data product for $NO_2$ (Boersma et al., 2018) and HCHO (De Smedt et al., 2018) for comparison with respective MAX-DOAS measurements. The DOAS retrieval of $NO_2$ SCDs uses a similar wavelength window and absorbers as DOMINO v2, but including an additional intensity offset. Also, an optical depth fit is performed in place of an intensity fit. While for DOMINO, the wavelength calibration was performed prior to the fit in the 409-428 nm window, for QA4ECV, it is performed along with the DOAS fit in the 405-465 nm wavelength window. The most significant improvement in the QA4ECV $NO_2$ retrieval concerns the AMF calculation and the stratospheric background correction. QA4ECV uses the $1° \times 1°$ spatial resolution TM5 model for the calculation of a priori $NO_2$ profiles. In several studies, QA4ECV $NO_2$ products are shown to have a better agreement with ground-based measurements and a smaller uncertainty than the other OMI $NO_2$ data products (Boersma et al., 2018;Chan et al., 2019;Zara et al., 2018).

The QA4ECV HCHO algorithm performs a DOAS optical depth fit including HCHO, $O_3$, BrO, $NO_2$, $O_4$ and the Ring effect in the wavelength interval 328.5 – 359 nm to obtain the HCHO SCDs (Zara et al., 2018). The across-track stripes observed in the retrieved DSCDs are corrected by subtracting an OMI detector row dependent mean equatorial pacific HCHO SCD. The a priori vertical profiles of HCHO are also calculated using the $1° \times 1°$ spatial resolution TM5 model.

**OMHCHO:** The OMHCHO v003 (González Abad et al., 2015) is a level 2 formaldehyde data product from NASA. The HCHO slant column densities are retrieved by employing a DOAS intensity fit in the 328.5 – 356.5nm wavelength window which includes the absorptions from HCHO, $O_3$, $NO_2$, BrO, $O_4$ and the Ring effect. Similar to OMNO2, the airmass factor for the conversion of the SCD into the VCD is calculated by considering climatological HCHO vertical profiles. The cloud information is also taken from the OMCLDO2 product. In order to account for the observed across-track stripes in the VCD, a normalization is performed with respect to the GEOS-Chem model calculated monthly climatological means over remote pacific.

**MAIAC:** The collection 6 MCD19A2 level 2 gridded product from MODIS provides the $1\times 1$ km$^2$ spatially resolved AOD at 470 nm and 550 nm based on the MAIAC (Multi-Angle Implementation of Atmospheric Correction) algorithm. In contrast to previous swath based retrievals for individual ground pixels, the MAIAC algorithm grids the L1B top of atmosphere reflectance in $1\times 1$ km$^2$ predefined sinusoidal grids prior to further processing. For each grid, data up to 16 days are accumulated, which include measurements at various viewing geometries from different orbits. The AOD retrieval relies on the ratio of measured spectral regression coefficients (SRC) at Band3(459-479nm)/Band7(2105-2155nm) and Band3/Band4(545-565nm). The analysis of time series of SRC enables the separation of the relatively static surface properties and fast varying atmospheric properties (e.g. AOD).

For generating a time series of AOD for comparison with MAX-DOAS, we have extracted the MODIS data spatially averaged within 2 km of the measurement location in Mohali. We have assumed a linear dependence between the logarithms of the wavelength and the AOD in the wavelength range 360 and 550nm to convert the MODIS AOD measured at 470 nm using the Ångström exponent calculated according to Eq. 3. We have only retained the highest quality MODIS AOD measurements corresponding to a QA value of "0000" in bits 8-11 provided in the MAC19A2 dataset (Lyapustin et al., 2018). This criterion removes all the data contaminated by clouds and those adjacent to cloudy pixels.

**D Evaluation of the airmass factors and a priori profiles in the OMI retrievals**

The sensitivity of satellite instruments is usually characterized by the so-called box airmass factors (bAMFs) profiles (Eskes and Boersma, 2003). bAMFs can be regarded as the airmass factors for discrete atmospheric layers. They can be integrated from surface until the tropopause (Trop), weighted by the trace gas profile, according to the following equation to get tropospheric airmass factors.

$$AMF_{Trop} = \sum_{i=0}^{Trop} bAMF_i \frac{VCD_i}{VCD_{Trop}}$$

D1

The vertical profiles of trace gases and aerosol extinction is a piece of important information needed to derive VCDs from the satellite measurements. Due to the absence of such measured information, global chemistry, usually transport models (e.g. a coarse $2.5°\times 2.5°$ spatial resolution TM4 for DOMINO and finer $1°\times 1°$ spatial resolution TM5 for QA4ECV) are employed. From Eq. D1 it is evident that if a relatively small fraction of an absorber (for e.g. NO$_2$) is located close to the ground in the a priori profiles, the resulting AMFs become positively biased, and finally, the VCDs become negatively biased. The finer horizontal resolution of the a priori NO$_2$ profiles in the QA4ECV product probably results in a more accurate representation of the NO$_2$ vertical profiles, especially close to strong emission sources like Mohali and thus improves the retrieved NO$_2$ VCD. In order to further investigate the underestimation (in particular late post monsoon and winter months), we first calculate the box airmass factors (with the RTM McARTIM) at 30° SZA over Mohali using mean aerosol extinction profiles (Fig. F16) retrieved from MAX-DOAS measurements and compare them with bAMFs used for the DOMINO and QA4ECV NO$_2$

retrievals. We perform this comparison for September (representative of early post monsoon or clean conditions) and October, November (representative of late post monsoon and winter respectively) months (Figure D1). Two striking features are observed:

1. Overall vertical variability of the bAMFs is different in the satellite products compared to the bAMF calculated for the MAX-DOAS aerosol profiles. The calculated bAMFs indicate a rather sharp increase with altitude until the first 1000m.

2. The calculated bAMFs show systematically higher values close to the surface than those used in the satellite retrievals. The largest underestimation is found for the DOMINO bAMF, which uses the rather coarse TM4 model input.

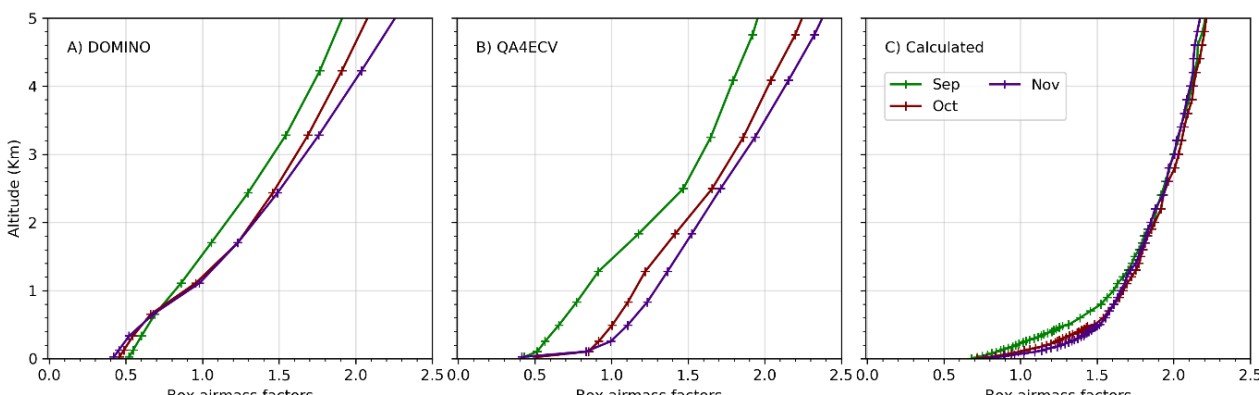

**Figure D1: Monthly mean box airmass factors used in the DOMINO (A) and QA4ECV (B) NO₂ data products under clear sky conditions (cloud fraction <0.1) over Mohali for September (when the aerosol load is small) and October and November (when the aerosol load is high close to the surface). The right panel (C) shows the corresponding box airmass factors calculated using MAX-DOAS aerosol extinction profiles.**

The smaller bAMF of the satellite product in the lower layers indicate a smaller weight of these layers where a major fraction of NO₂ is present. Hence, the AMFs will become smaller and the VCDs become higher if the relative a priori NO₂ profiles for satellite data retrieval would be adjusted to the observed profiles. However, if the a priori profiles assume a smaller fraction of NO₂ in layers close to the surface, higher layers will get a larger weight in Eq. D1, resulting in larger AMF and smaller VCD.

Hence, in the next step, we compare the a priori NO₂ profiles of the satellite data product with those retrieved from MAX-DOAS measurements. Unfortunately, our comparison is limited to the DOMINO product, as the a priori NO₂ profiles are not available for the other products.

Fig D2 shows monthly mean relative vertical profiles of NO₂ retrieved by MAX-DOAS and the corresponding TM4 profiles. We can clearly notice two differences:

1. The vertical gradient in the relative profile shape is stronger in the MAX-DOAS profiles than in the TM4 product.

2. The TM4 relative a priori shape is somewhat similar in all months whereas the profile shape retrieved from DOAS changes strongly with season. More than half of the NO₂ is located in the bottom-most layer in the winter months.

The consequence of the first observation is that the total airmass factors will generally be higher if the TM4 profiles are used as a priori which results in smaller VCDs as also observed in Fig 8. The consequence of the second observation is that in the winter months, the discrepancy due to a priori profile is even stronger, resulting in a larger disagreement with the measured MAX-DOAS VCDs.

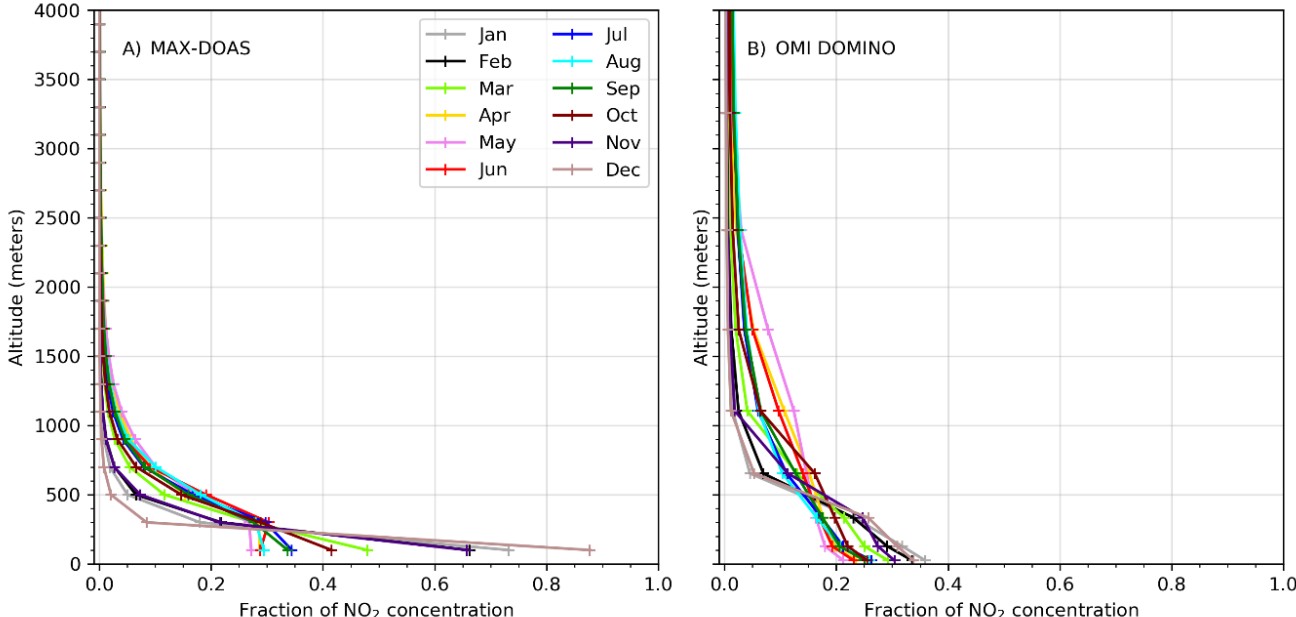

**Figure D2: Monthly mean relative a priori NO₂ profiles over Mohali retrieved from MAX-DOAS measurements around the OMI overpass time (between 12:30 and 14:30 local time) (A) and the TM4 a priori profiles used for the DOMINO retrieval (B). The a priori profiles are not available for the QA4ECV NO₂ product.**

Accurate satellite AMFs can be recalculated using MAX-DOAS vertical profiles as a priori, which could improve the agreement with MAX-DOAS observations as also shown by De Smedt et al. (2015). The satellite AMFs corresponding to the MAX-DOAS a priori profiles ($x_m$) can be recalculated according to the following equation:

$$AMF_{trop}(x_m) = AMF_{tot}(x_a) \frac{\sum_{l=1}^{L} A_l x_{m,l}}{\sum_{l=1}^{L} x_{m,l}}$$   D2

Here,

$x_a$ : Original satellite (TM5) a priori trace gas profile

$L$ : Tropopause level index

$A$ : Satellite averaging kernels

$AMF_{tot}$   Total airmass factor

Total AMF ($AMF_{tot}$) and satellite averaging kernels used for OMI retrievals are crucial information required to recalculate the satellite (OMI) AMF. For NO$_2$, only the DOMINO product provides both $AMF_{tot}$ and satellite averaging kernels in the data product. We attempted to recalculate the $AMF_{trop}$ using the MAX-DOAS profiles, but this resulted in very small airmass factors (and very large recalculated OMI VCDs). The small $AMF_{trop}$ is due to the fact that the MAX-DOAS profiles do not account for the background NO$_2$ in the free troposphere, where the satellite averaging kernels are large. In the next step, we used hybrid profiles such that we only replaced the profiles in the lowest 5 km and 2.5 km of the TM5 profile with those retrieved from MAX-DOAS measurements. The observations are summarized in Fig D3. We note that even with the hybrid approach, there is an overestimation of VCDs for many months. This is probably caused by the incorrect aerosol profiles used for the calculation of the averaging kernels in the satellite analyses.

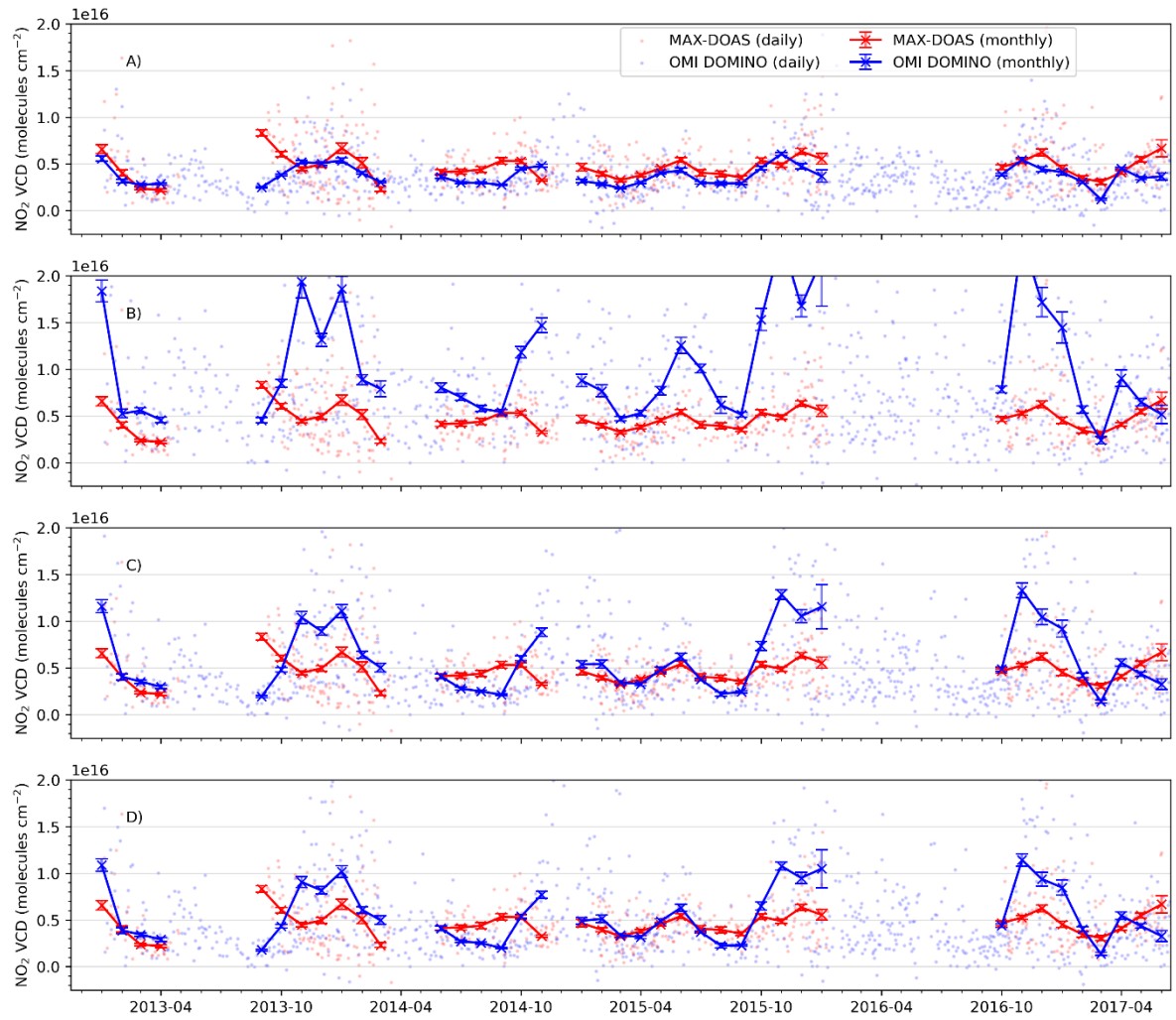

Figure D3: Time series of the MAX-DOAS and OMI DOMINO NO$_2$ tropospheric VCDs for A) when no modifications in the a priori profiles are applied, B) when the a priori profiles of DOMINO are replaced by the MAX-DOAS profiles, C) when the lowest

**5km of the profiles of the a priori are replaced by the corresponding MAX-DOAS profiles and D) when the lowest 2.5 km of the profiles of the a priori are replaced by the corresponding MAX-DOAS profiles.**

For formaldehyde, $AMF_{tot}$ is not provided in the QA4ECV products, while averaging kernels are not available for OMHCHO products. However, we approximated $AMF_{tot}$ to be close to $AMF_{trop}$ because of the negligible amount of HCHO present in the stratosphere. Using this approximation, we have recalculated the modified $AMF_{trop}$ using MAX-DOAS profiles as a priori. Similar to NO$_2$ we have replaced the profile in the lowest 2.5 km of the TM5 profile with that retrieved from MAX-DOAS measurements. For HCHO, the modified VCDs are

largely positive biased. Like for NO$_2$, this overestimation might be caused by incorrect aerosol profiles used for the calculation of the averaging kernels in the satellite analyses.

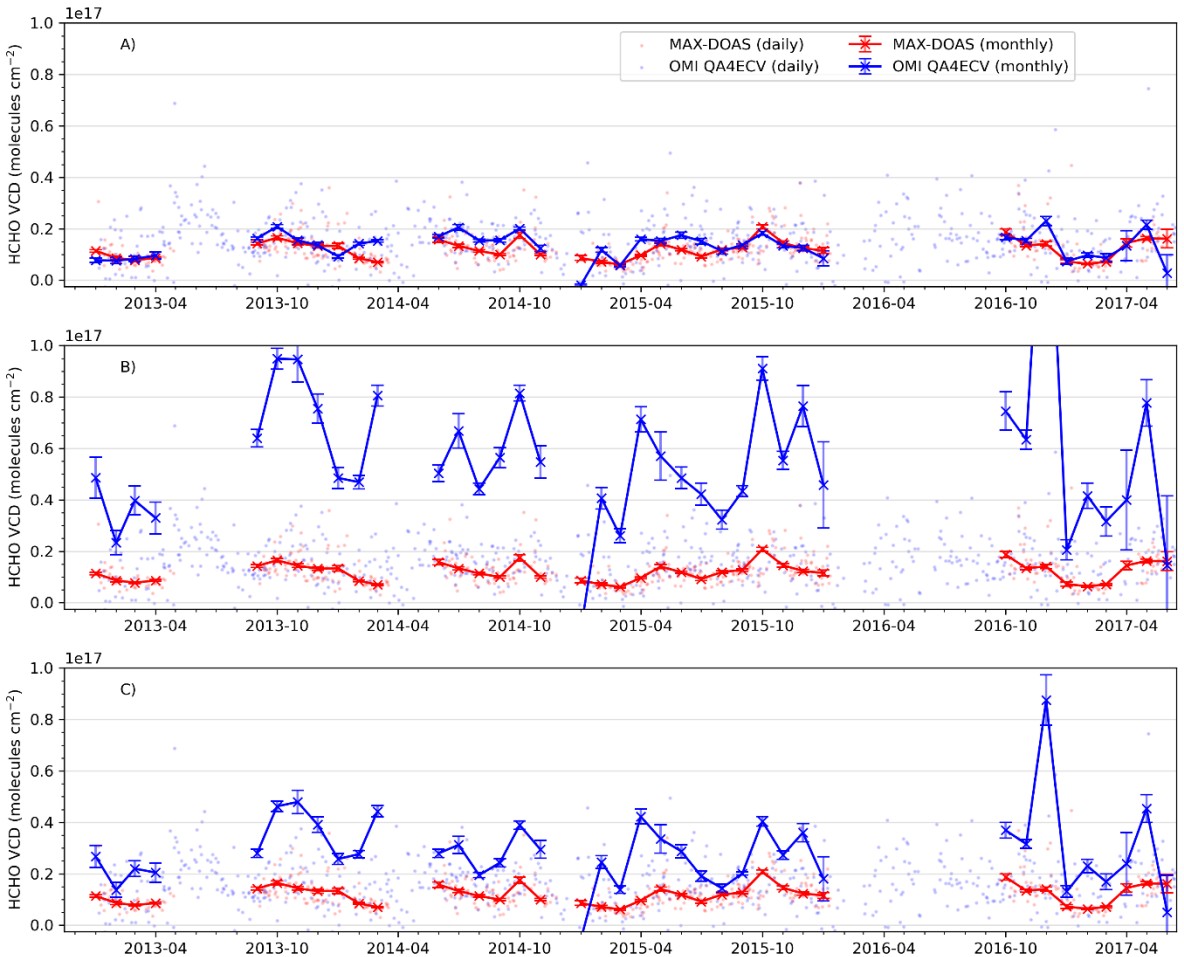

**Figure D4: Time series of the MAX-DOAS and OMI QA4ECV HCHO tropospheric VCDs for A) when no modifications in the a priori profiles are applied, B) when the a priori profiles of QA4ECV are replaced by the MAX-DOAS profiles and C) when the lowest 2.5 km of the a priori profiles are replaced by corresponding MAX-DOAS profiles.**

**E Comparison of surface concentration of NO₂ and HCHO from MAX-DOAS and *in situ* measurements**

In addition to the comparison with the $NO_2$ and HCHO VCDs from satellite data products, also the surface concentration derived from MAX-DOAS is compared with *in situ* measurements. From this comparison, a first-order assessment of the quality of profile retrieval is obtained (Wang et al., 2019;Vlemmix et al., 2015). Often systematic differences up to 30% are found between MAX-DOAS and *in situ* measurements which are mainly related to the limited vertical (and horizontal) resolution of the MAX-DOAS profiles (Vlemmix et al., 2015;Wang et al., 2019;Li et al., 2013). The vertical resolution of the profiles retrieved from MAX-DOAS measurements depends strongly on altitude.

Taking the standard deviation of the daily means into account for an ODR fit further improves the agreement between the MAX-DOAS and in situ measurements of the $NO_2$ surface VMRs. The frequency distribution of the bias between the two measurements shows a normal distribution which peaks at ~0.7 ppb. Please note that in Fig. 12, we have used all profiles which were flagged as valid and warning by MAPA. The linear correlation coefficient (r) changes slightly from 0.62 to 0.60 if only valid retrieval results were considered, while for the ODR fit, the slope and intercept change from 0.83 and 1.78 to 0.90 and 1.39, respectively. MAX-DOAS is sensitive towards airmasses in the viewing direction of the instrument whereas *in situ* analysers are sensitive for air directly sampled by the inlet. $NO_2$ is primarily emitted (or converted very fast from NO near the source) close to surface. So, if the measurements are performed in the vicinity of emission sources, we expect higher $NO_2$ from *in situ* measurements than MAX-DOAS, which provides a mean concentration in the 0-200m output grid. This was also observed in previous intercomparison studies (Wang et al., 2019;Li et al., 2013;Piters et al., 2012). To a surprising extent, we observe that until the end of 2014, most surface VMR from MAX-DOAS are systematically higher than the *in situ* measurements, while afterwards, the differences become smaller. A plausible reason for the positive bias is the plumes from the Rupnagar power plant (PP1, Fig. 1), ~45km far from Mohali in the north-west direction, which also is the viewing direction of the MAX-DOAS instrument. Pawar et al. (2015) have previously shown back trajectories of air mass arriving at Mohali for a period of 2 years (2011-2013). Except for monsoon, more than 80% of the back trajectories were among the clusters 'westerlies', 'local' or 'calm', all of which include the location of PP1. In monsoon, these clusters accounted for more than 50% of the total. From the wind rose plot of Fig F1, we also observe that in all the seasons except monsoon, the major fetch region includes PP1. The Rupnagar power plant was active with 90% of its capacity until October 2014, and operated only with 20% of its capacity till the ceasing of its operation in 2018 (https://timesofindia.indiatimes.com/city/chandigarh/Punjab-shuts-10-of-14-thermal-power-plants/articleshow/44730937.cms: last access 03.09.2020). The power plant plume is emitted directly at an altitude (~100m), much higher than the inlet height of the *in situ* measurements (~15m). Due to its coarser vertical resolution, the MAX-DOAS surface VMRs are also influenced by the $NO_2$ at higher altitudes (e.g. from the power plant plume). During stagnant conditions, the vertical mixing is suppressed, and we expect a larger bias between the two measurements. From the MAX-DOAS $NO_2$ profiles, we can also estimate the extent of the vertical mixing of $NO_2$ in terms of the characteristic profile height ($H_{75}$). Fig. 12c shows the scatter plot between the MAX-DOAS and *in situ* surface VMRs of

$NO_2$ colour-coded according to $H_{75}$. We observe that for profile heights less than 200 meters, the MAX-DOAS surface VMRs are larger positively biased than for higher $H_{75}$. During the summer months (Mar-June), due to the radiative heating of the surface, vertical mixing is enhanced and leads to a higher $H_{75}$ (Fig. 6). Also, the downmixing of the power plant plume to the surface is more efficient during such conditions. Hence, during summer 2013, even though the power plant was operational at high capacity, we see a smaller bias between the two data sets. For some applications, the limited vertical resolution of MAX-DOAS instrument can be regarded as an advantage in terms of robustness against stratification in stable meteorological conditions and yield a more spatially representative value.

The horizontal heterogeneity of the $NO_2$ VMR and differences in the spatial representativeness of the measurements can also add to the observed bias in the overall measurement period. The measurements were performed within an educational institute campus, located in the south-east corner of the tri-city Panchkula-Chandigarh and Mohali. From the high-resolution TROPOMI $NO_2$ maps for the year 2018 (Fig. 1), we can observe that the measurement location is relatively clean (with respect to the $NO_2$ VCD) compared to the surrounding regions. The viewing direction of the MAX-DOAS instrument is towards the city, and the horizontal sensitivity along the range of sight is typically a few kilometres. Thus, the MAX-DOAS measurements are sensitive for an urban air mixture consisting of higher $NO_2$ than at the measurement location. Post-2014, the bias is within 20%, similar to those observed in previous studies, which can be attributed to these factors.

The frequency distribution of the bias (MAX-DOAS – PTR-MS) in the individual measurements of the HCHO surface VMRs shows a distribution similar to lognormal with a maximum at ~1.1 ppb and skewed towards positive values. The large bias can also be inferred from the large offset (3.18 ppb) and slope (1.14) in the linear regression of HCHO VMRs measured by MAX-DOAS and by PTR-MS. We observe that until May 2015, there was a general agreement between the two measurements regarding their temporal variability, but the *in situ* VMRs were generally lower. Post-May 2015, the bias between the two measurements became larger. The reason for the larger bias is not well understood. We also observe a large variability in the MAX-DOAS HCHO VMRs, which possibly arises due to a larger uncertainty in the MAX-DOAS HCHO measurements as compared to the random uncertainty of ~30% in the PTR-MS HCHO measurements. The major contribution to the error budget is from fitting errors in the DOAS fit in addition to the uncertainties in the profile inversion algorithm and cross sections. The HCHO surface VMRs retrieved using the MAX-DOAS measurements have an uncertainty of ~50% (as compared to only ~20% for $NO_2$ surface VMR)(Wang et al., 2017b).

Secondary photochemical production is the major source of atmospheric formaldehyde. The photo-oxidation of primarily emitted VOCs occurs during the course of their mixing up in the boundary layer, and hence, a significant amount of formaldehyde is observed at altitudes up to 600 m or even higher in some cases. The surface VMRs from MAX-DOAS shown in Fig. 13 represent the mean in the lowest 200 m layer of the MAPA output; which might also be influenced by higher altitudes due to limited vertical resolution of MAX-DOAS. Surface VMRs from the PTR-MS measurements are sensitive to the inlet height (~15m). Hence, a higher VMR from MAX-DOAS measurement was expected. This is further supported by our observations in Fig. F8, where we observe that for the periods when the emissions of precursors of HCHO are higher (e.g., from crop residue fires in May, June, October and November and from burning for domestic heating in Dec. and Jan.), the bias

between the MAX-DOAS and *in situ* VMRs is also higher. Nevertheless, keeping in mind the systematic uncertainty of the *in situ* measurements (which could not be quantified within the scope of this study due to unavailability of calibration standards) and the high uncertainty of MAX-DOAS measurements, we cannot further interpret the comparison results.

**F Additional figures**

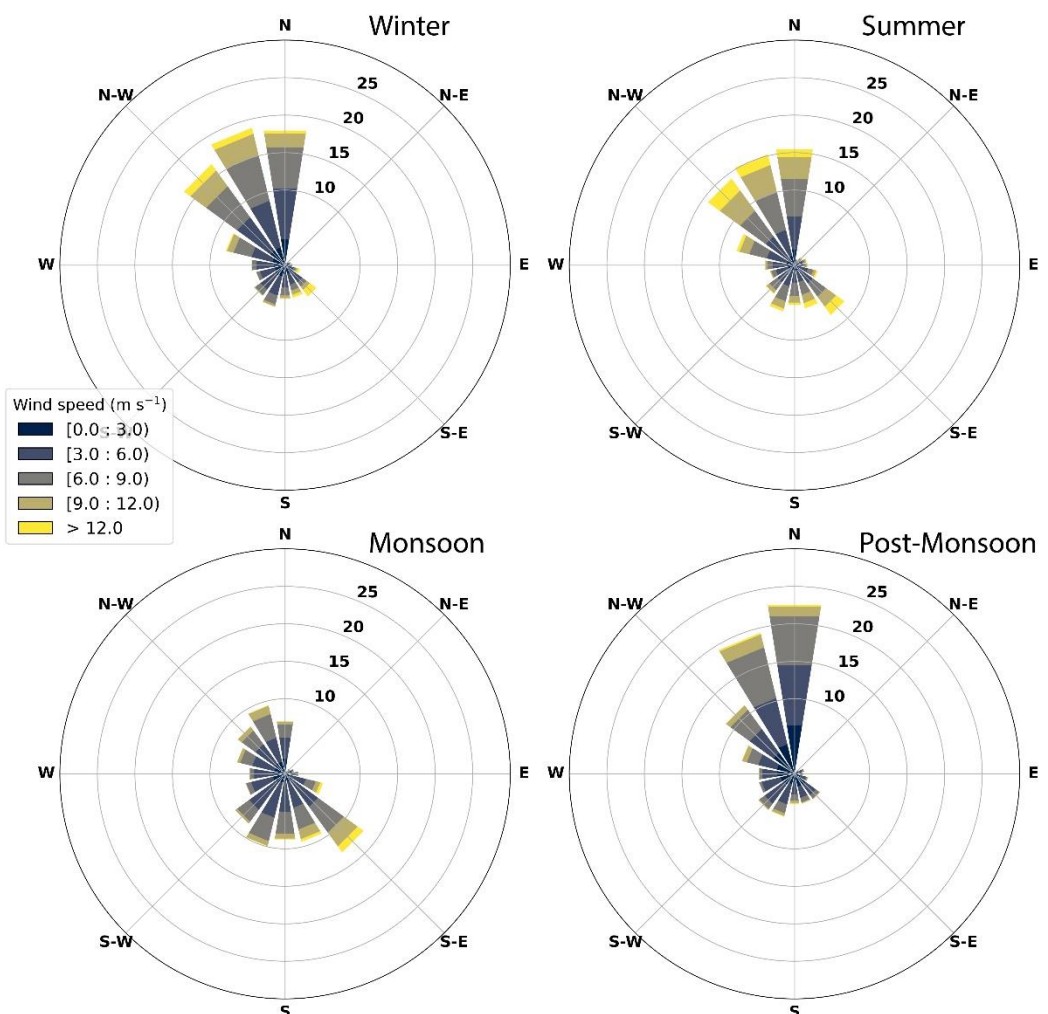

**Figure F1: Wind rose plots showing the major fetch region of air mass arriving at Mohali for the four major seasons of the year.**

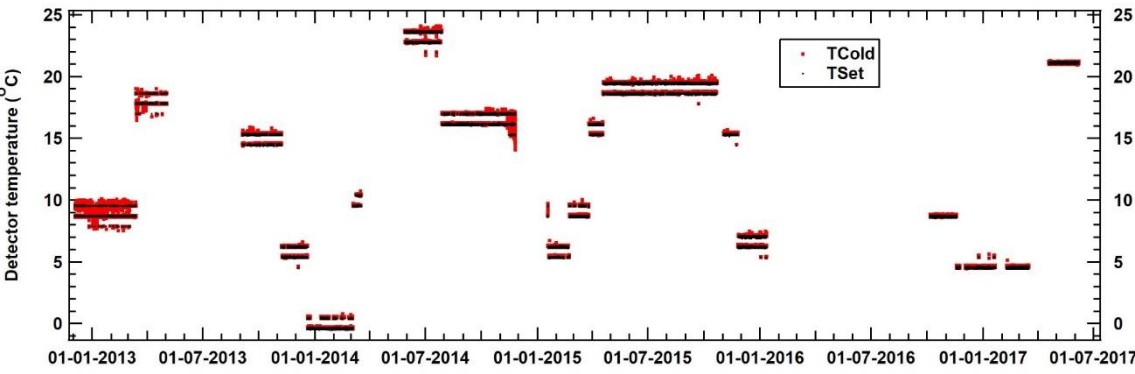

**Figure F2: The time series of the nominal ($T_{set}$) and the actual temperature ($T_{cold}$) of the detector within the MAX-DOAS instrument.**


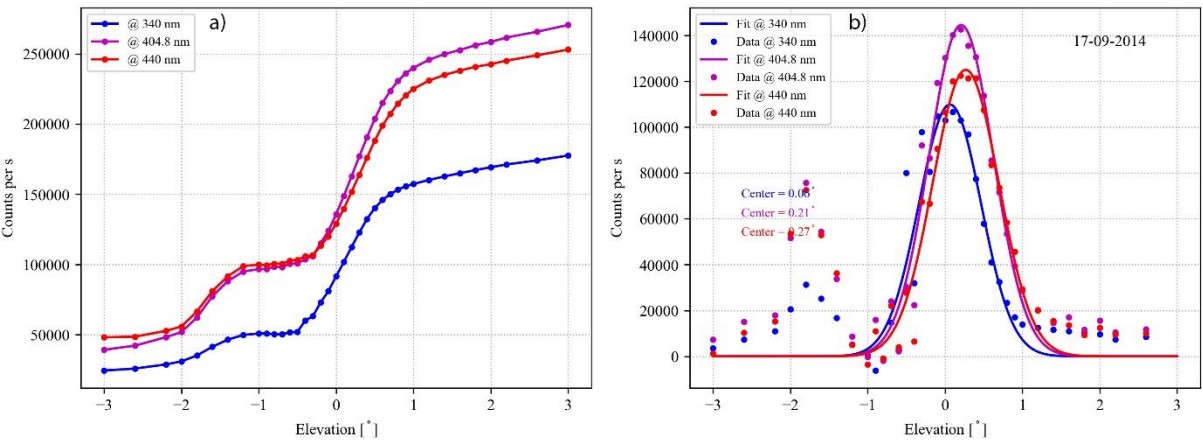

**Figure F3: Example plots showing the measured radiance (left) and the derivative of the radiance (right) during a horizon scan spanning elevation viewing angles from -3° to 3°. The figures correspond to the horizon scan performed on 17-09-2014.**

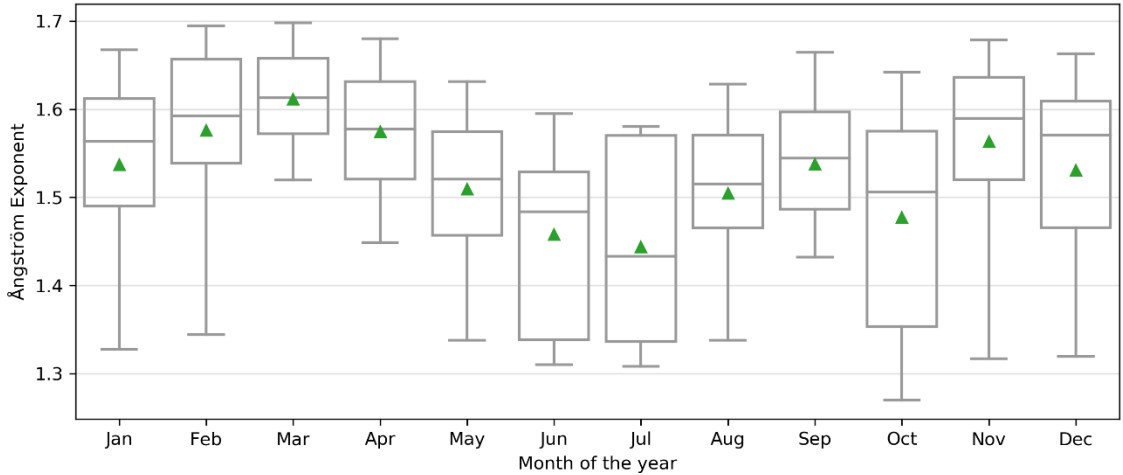

**Figure F4: Monthly variation of Ångström exponent for the wavelength pair 470 and 550 nm derived from MODIS measurements over Mohali. The green triangles represent the monthly means. The centre line of each box represents the median values, whereas the box represents the interquartile range. The whiskers represent the 5th and 95th percentiles.**

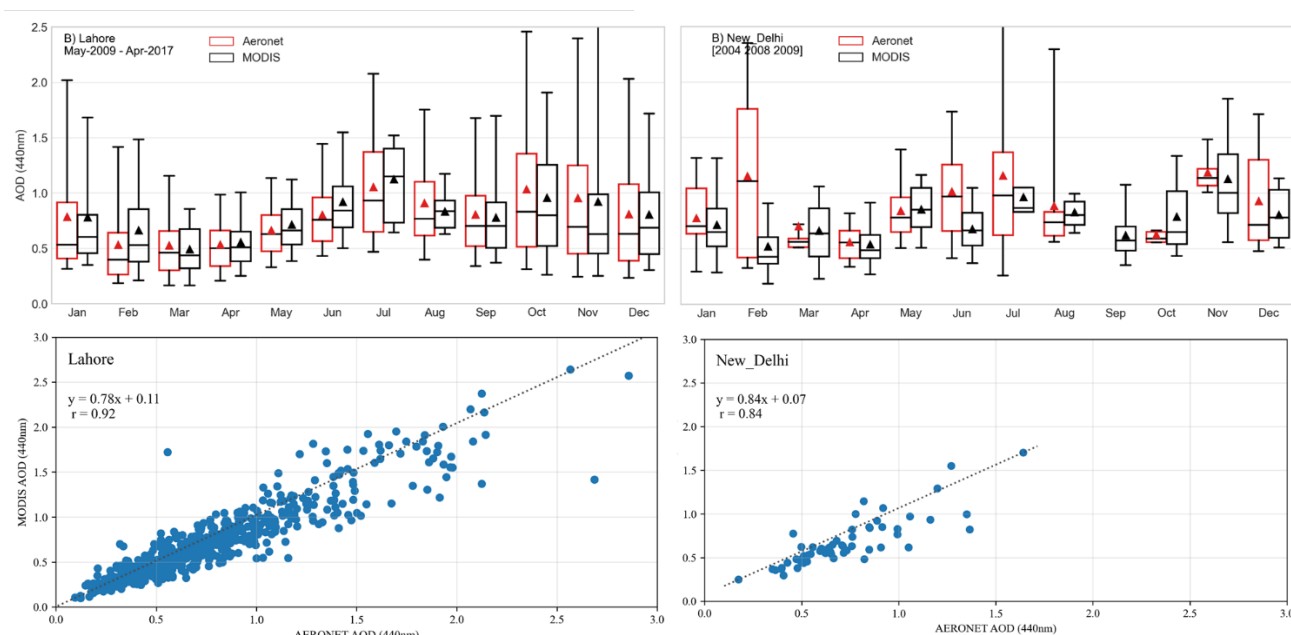

**Figure F5: Monthly variation of the AOD (at 440nm) as observed by AERONET sun photometers (red boxes) and MODIS (black boxes) at two sites (A. Lahore and B. New Delhi), which are the nearest stations to Mohali in the Indo-Gangetic Plain. The bottom panel shows the corresponding scatter plots indicating the agreement in the daily MODIS and AERONET measurements.**

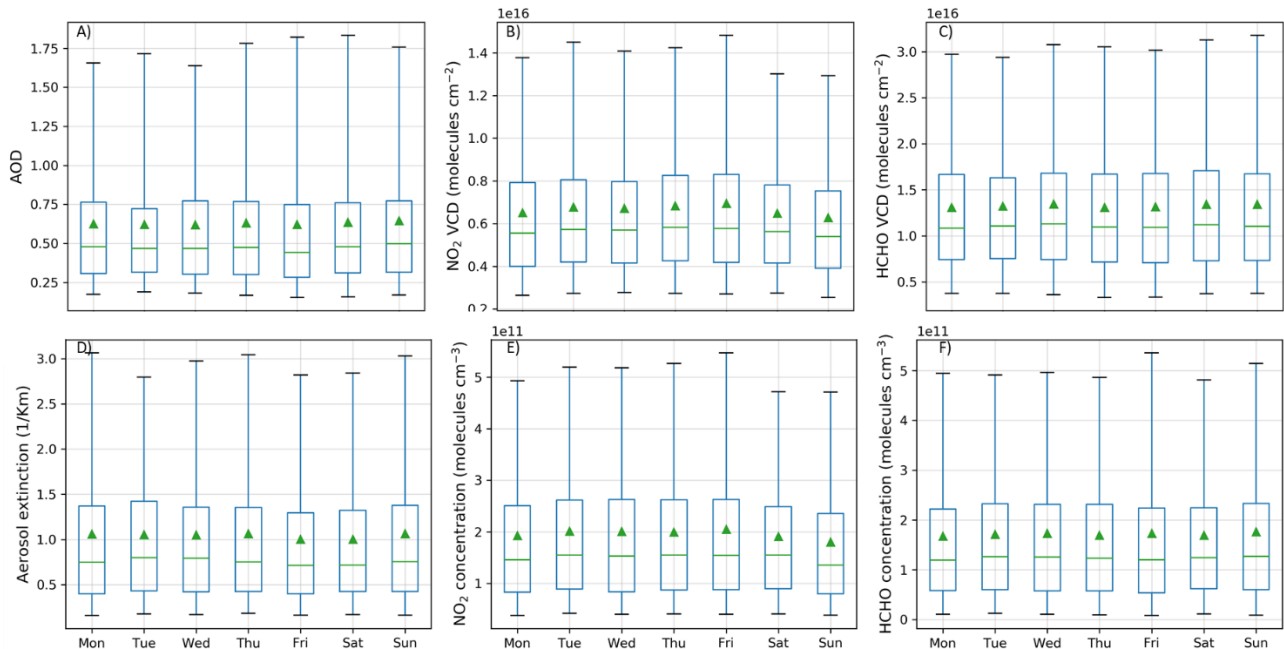

**Figure F6: Mean weekly variation of A) AOD, B) NO₂ VCD and C) HCHO VCD in the top panel. The bottom panel shows the mean weekly variation of D) aerosol extinction E) NO₂ concentration and F) HCHO concentration in the bottom-most layer (0-200m) retrieved from the profile inversion of the MAX-DOAS measurements.**


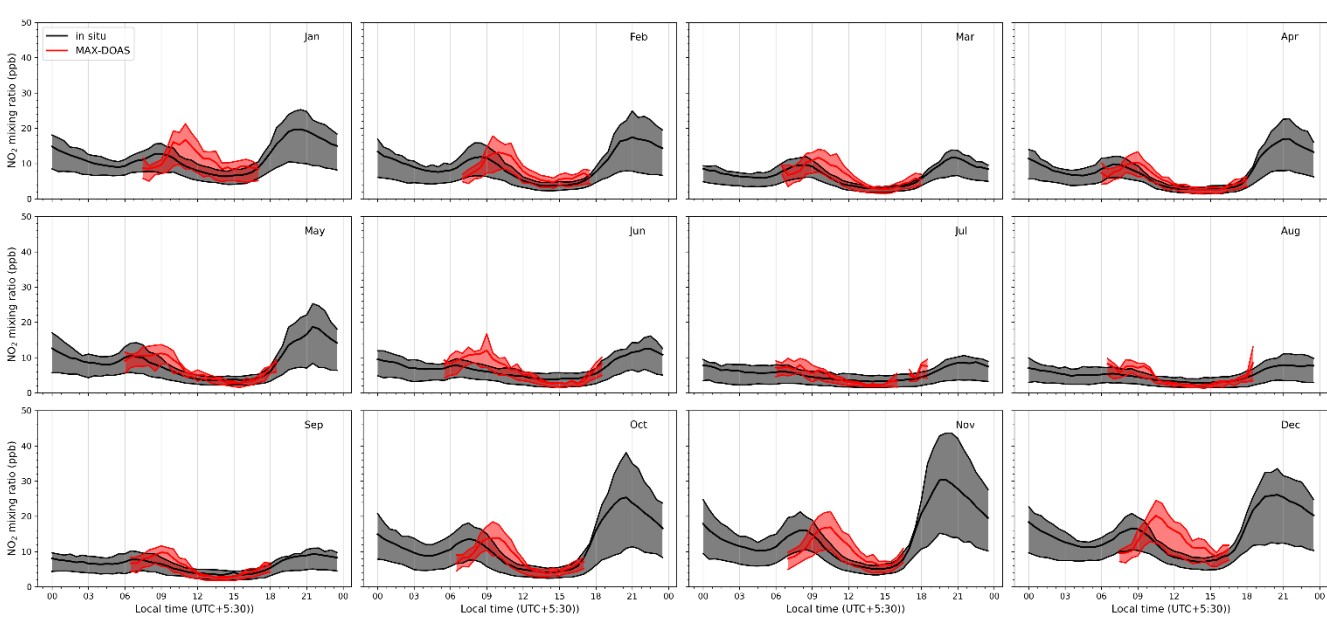

**Figure F7: Mean diurnal profiles of surface NO₂ mixing ratios measured by an *in situ* analyser (black) and retrieved from the MAX-DOAS measurements (red) for different months of the year. The dark line represents the mean value while the shaded region above and below the dark lines represent the 75ᵗʰ and 25ᵗʰ percentiles, respectively.**


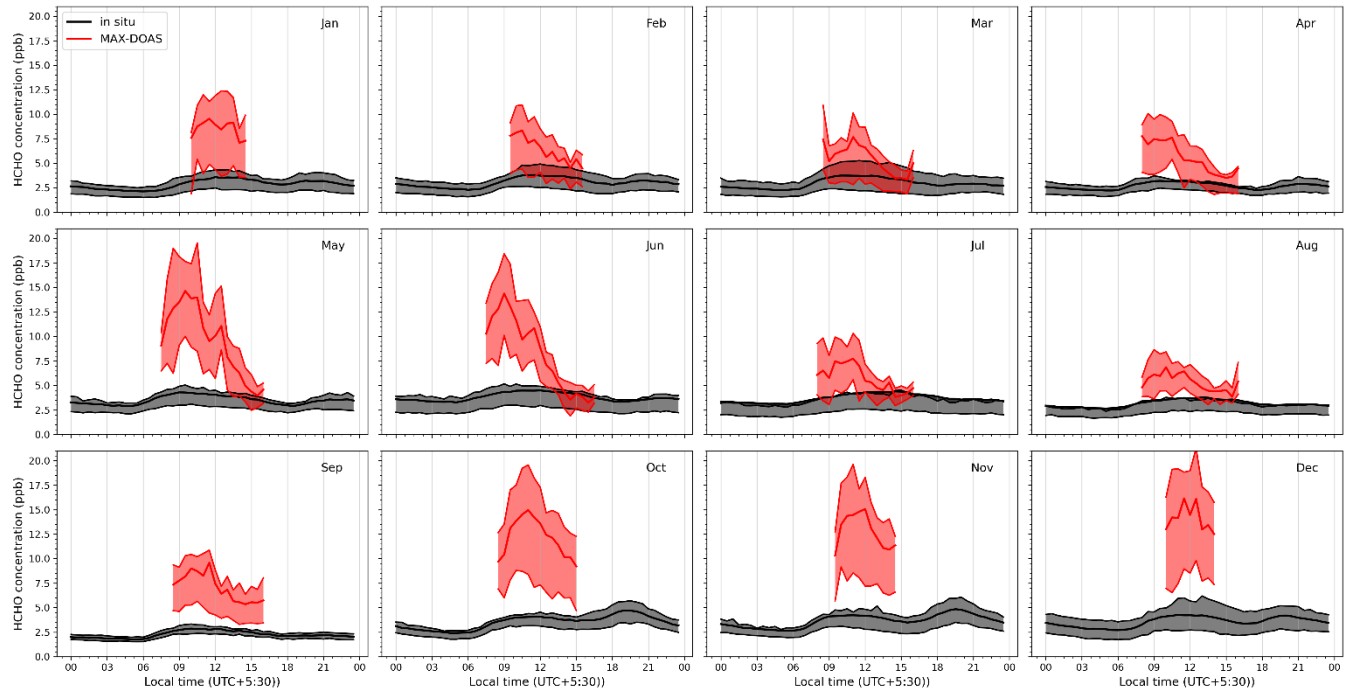

**Figure F8: Mean diurnal profiles of surface HCHO mixing ratios measured by the PTR-MS (black) and retrieved from the MAX-DOAS measurements (red) for different months of the year. The dark line represents the mean value while the shaded region above and below the dark lines represent the 75th and 25th percentiles, respectively.**


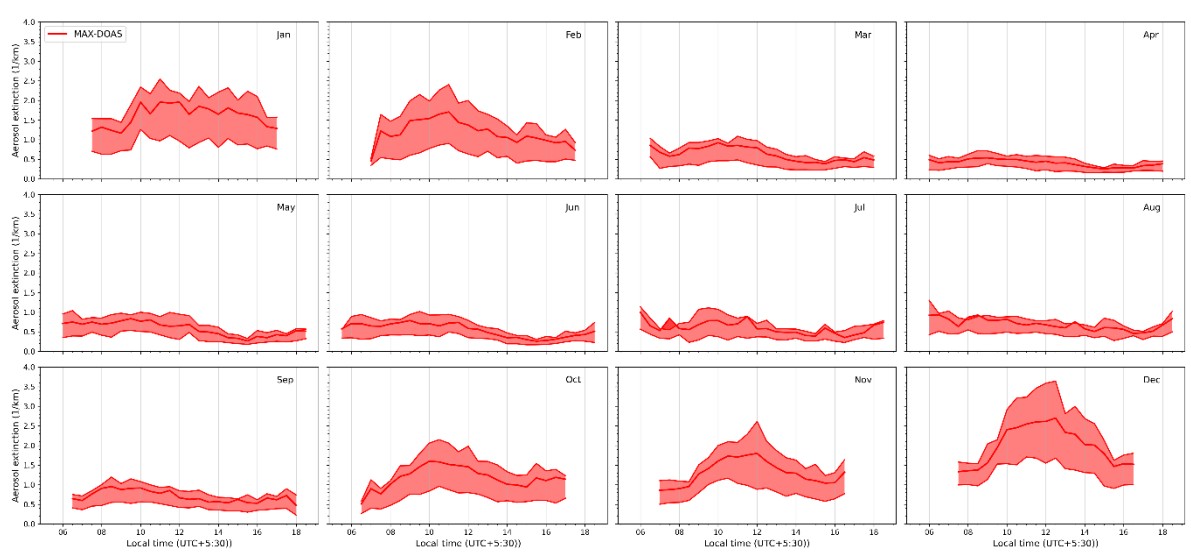

**Figure F9: Mean diurnal profiles of the aerosol extinction in the bottom-most layer (0-200m) retrieved from the MAX-DOAS measurements for different months of the year. The dark line represents the mean value while the shaded region above and below the dark lines represent the 75ᵗʰ and 25ᵗʰ percentiles, respectively.**

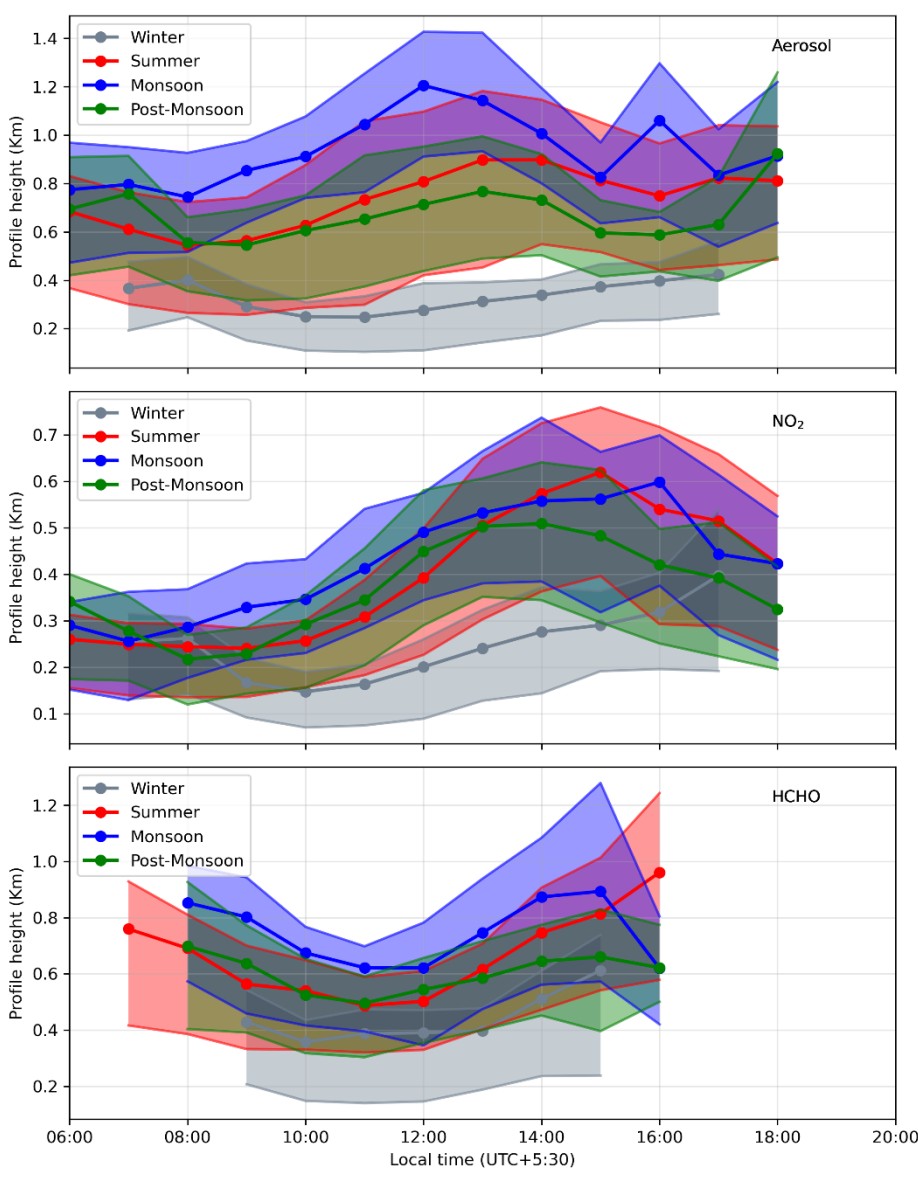


**Figure F10: Diurnal variation of characteristic profile heights of aerosol (top panel), NO₂ (centre panel) and HCHO (bottom panel) for the four major seasons.**

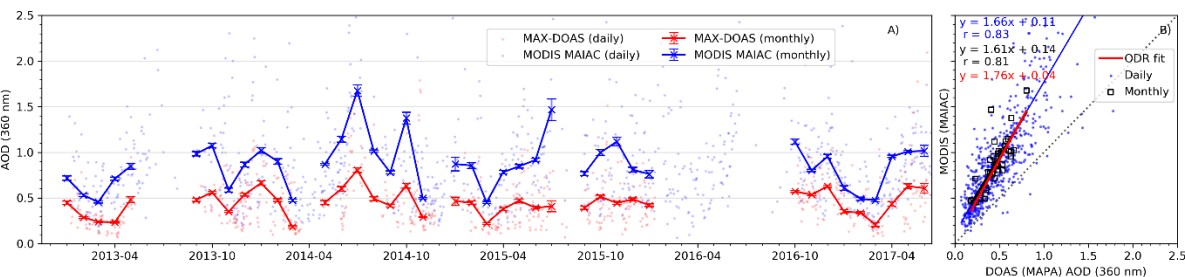

**Figure F11: Intercomparison of daily (dots) and monthly mean (lines and markers) AOD at 360 nm retrieved from ground-based MAX-DOAS O₄ measurements and from the MODIS MAIAC data product when no scaling factors were applied for the O₄ dSCDs.**

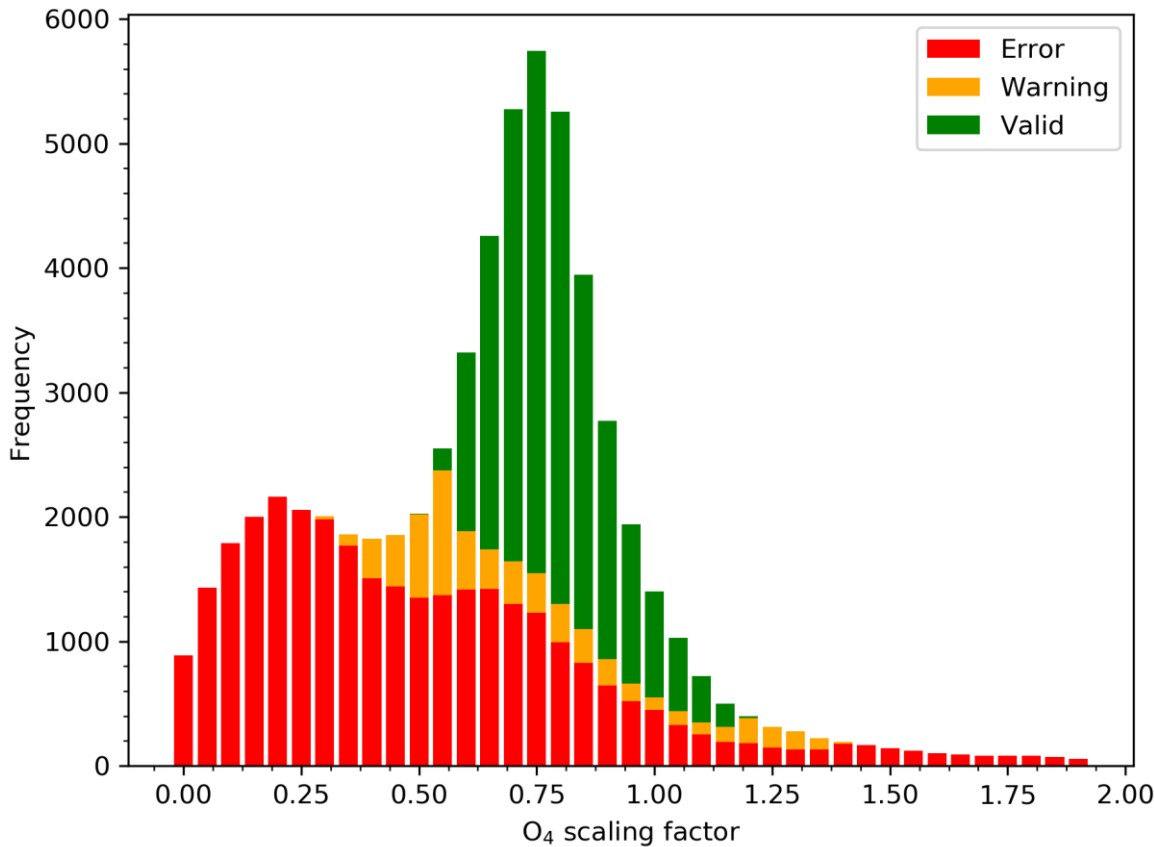

**Figure F12: Frequency distribution of O₄ scaling factors derived from profile inversions, which allowed to vary the O₄ scaling factors in order to achieve an agreement between measurements and forward model. The green bars show retrievals which are flagged valid while the orange and red bars indicate retrievals with warning and error flags.**

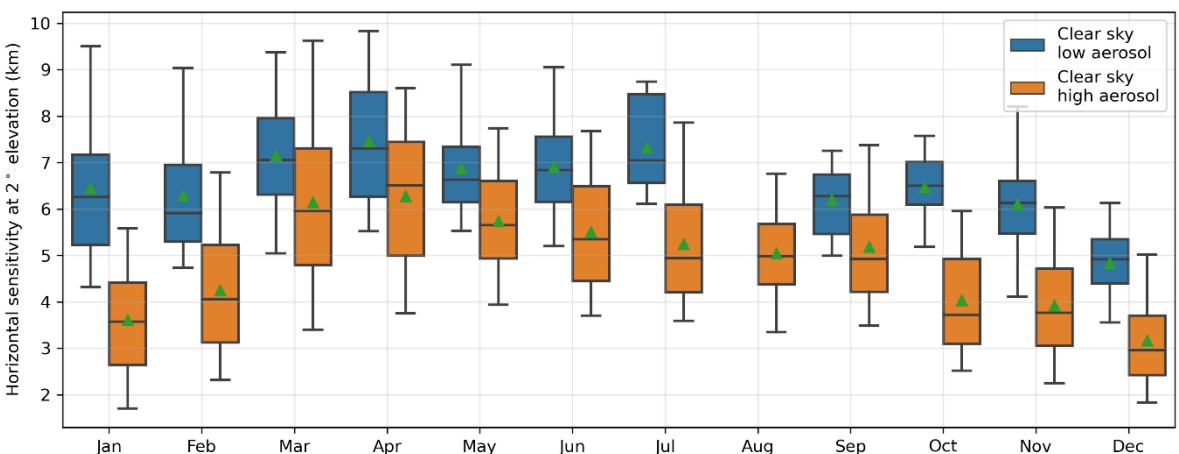

**Figure F13: Box and whiskers plot showing the horizontal sensitivity distance of MAX-DOAS measurements during afternoon hours (between 12:00 and 15:00 local time) for 2° elevation angle. The blue boxes represent clear sky conditions with low aerosol load, and the orange boxes indicate clear sky conditions with high aerosol load.**

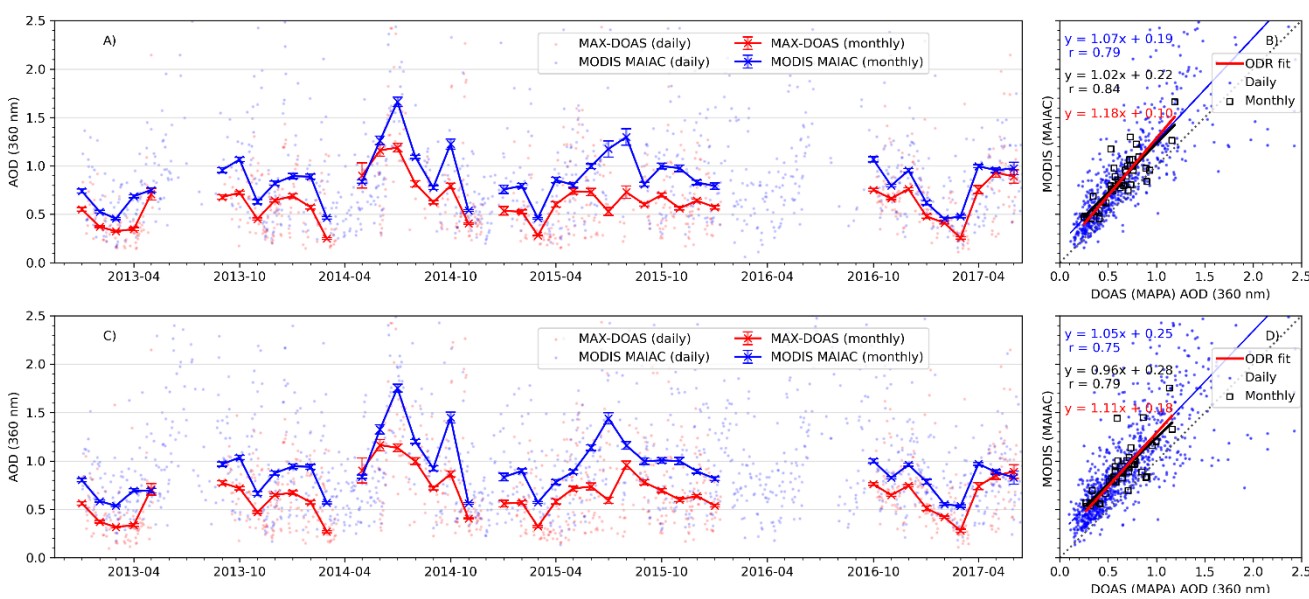

**Figure F14: Intercomparison of daily (dots) and monthly mean (lines and markers) AOD at 360 nm retrieved from ground-based MAX-DOAS $O_4$ measurements and from the MODIS MAIAC data product when spatially averaged over 5 km (top panel) and 25 km (bottom panel) around Mohali. $O_4$ dSCD were scaled by a factor of 0.8, as discussed in the main text.**

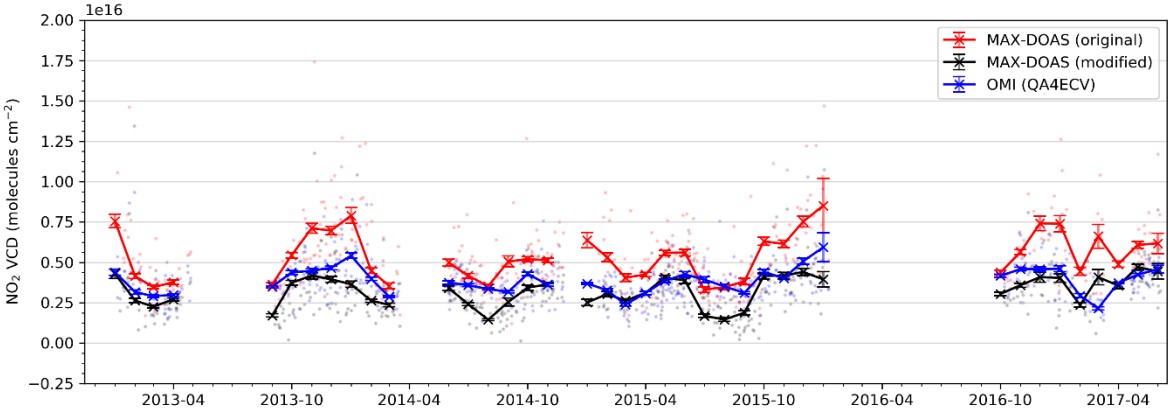

Figure F15: Time series of daily (dots) and monthly means (lines and markers) of MAX-DOAS NO₂ VCDs, OMI QA4ECV NO₂ VCDs and MAX-DOAS VCDs modified using the QA4ECV averaging kernels and the TM4 a priori profiles.

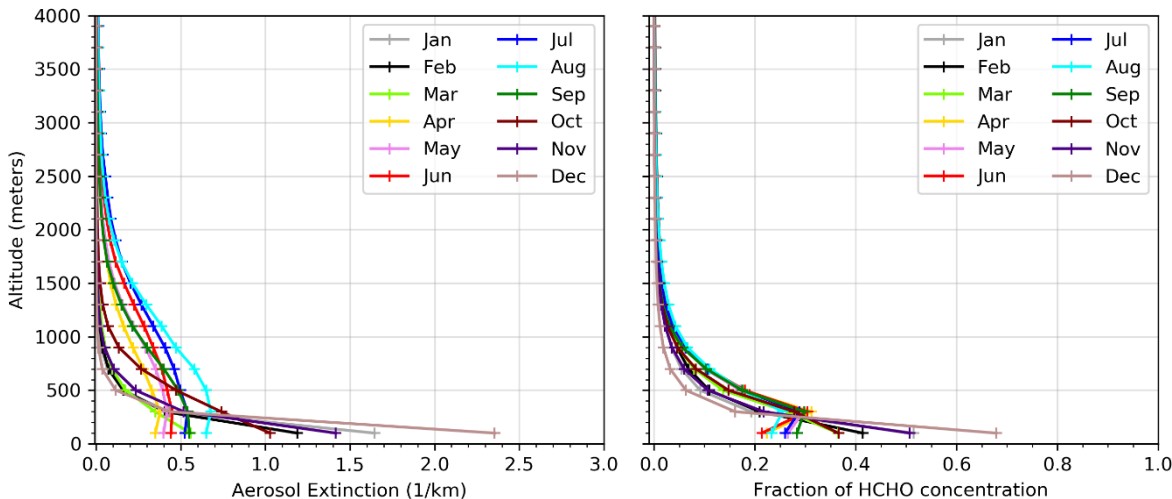

Figure F16: Monthly mean aerosol extinction profiles (left) retrieved from MAX-DOAS O₄ measurement and HCHO profiles (right) over Mohali at around the OMI overpass time (12:30-14:30 local time).

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
