# Peer review of "Long term MAX-DOAS measurements of NO2, HCHO and aerosols and evaluation of corresponding satellite data products over Mohali in the Indo-Gangetic plain"

_Atmospheric Chemistry and Physics, 2020_

## Referee Comment (RC1) · Anonymous Referee #1 · 17 Jul 2020

This paper investigates the temporal variation of the vertical distributions of aerosols, NO2, and HCHO and corresponding AOD and vertical column densities (VCDs) retrieved from MAX-DOAS observations performed at Mohali, north-west Indo Gangetic plain. The measurements presented in the study cover the period from January 2013 to June 2017. The different factors driving the seasonal and diurnal variations of the above parameters are identified and extensively discussed. The MAX-DOAS data sets are also used to validate co-located observations from the OMI and MODIS satellite instruments.

[Figure]

This paper is well written, clearly structured, and presents very interesting results which fit well with the scope of ACP. In addition, I would like to say that for me this study is a breakthrough in the evaluation of the air quality in India using ground-based and space-borne remote sensing observations. I therefore strongly recommend the final publication of the manuscript after addressing the following comments:

Specific comments:

Page 3, lines 97-98: It is written that the two stationary MAX-DOAS measurements over India did not report trace gas VCDs? What do they report then? Only aerosol measurements? The locations of those stationary MAX-DOAS measurements are also not clear.

Page 5, lines 160-163: You should add in the legend of Fig D2 to which day those horizon scan results corresponds? Is the steep increase in the measured intensity always centred between $0°$ and $0.3°$ during the 4.5 years of measurements?

Page 6, line 176 (Eq. 1): How do you select SCD90? Do you use the zenith dSCD of the scan for correcting all the off-axis dSCDs, or do you interpolate the zenith dSCD at the time of the off-axis measurements of a scan by using the zenith dSCDs just before and just after those off-axis measurements? From my experience, it can have an impact on the resulting off-axis dSCDs, especially in the case of HCHO.

Page 14, lines 447-448 and next paragraphs on page 15: are there any measurements of the boundary layer height at Mohali? It could be an added value to the discussion. If no measurements exists, maybe ECMWF Era-interim BLH could be used.

Page 18, line 562 and page 23, line 723: MAX-DOAS VCD measurements are spatially representative of a few kilometres in the field of view and this horizontal sensitivity strongly depends on the aerosol load. I think it would be useful to have to estimate this horizontal sensitivity for the main sky conditions presented in Figure 3. Numerous studies have also shown that taking into account this horizontal sensitivity in the

selection of the co-located satellite data can have an impact on the agreement with ground-based MAX-DOAS observations. This is especially the case in the present study where MODIS data are given on a 1x1 km2 grid, i.e. at a horizontal resolution which is significantly higher than the typical horizontal distance of several kilometres representative of the MAX-DOAS measurements.

Page 21, lines 646-648: Since satellite total HCHO AMFs and averaging kernels are missing in the files, you could proceed the other way round for eliminating the difference caused by the non-representative satellite a priori HCHO profiles, i.e. recalculating satellite AMFs using MAX-DOAS vertical profiles and dividing the satellite slant column densities by those new AMFs (see e.g. De Smedt et al., Atmos. Chem. Phys., 15, 12519–12545, 2015). Maybe something worth to try.

Page 50, Figure D4: How MODIS would compare to the AERONET sun photometer measurements at these two sites? Angstrom exponent could be used to convert MODIS AOD from 470 nm to 360 nm.

Technical corrections:

Page 1, line 11: 'We investigate the temporal variation and the vertical profiles...'. I find the sentence a bit misleading since you also investigate the temporal variation of the vertical profiles (-> see Figure 5). Maybe some rephrasing is needed here.

Page 2, line 59: 'ground based' -> 'ground-based'; There are some other places where this should be corrected too.

Page 5, line 138: Multi Axis Differential Optical Absorption Spectroscopy (MAX-DOAS).

Page 14, line 444: 'diurnal trends' -> 'diurnal variations' ?

Page 28, Figure 1: The names of the cities and the x and y axes labels are difficult to read. Maybe you could use a larger font size.

Page 37, Figure 13: in the legend, it should be 'Daily mean HCHO mixing ratios...' and

not NO2 mixing ratios.

---

## Referee Comment (RC2) · Anonymous Referee #2 · 5 Aug 2020

Review for 'Long term MAX-DOAS measurements of NO2, HCHO and aerosols and evaluation of corresponding satellite data products over Mohali in the Indo-Gangetic plain'

The paper titled 'Long term MAX-DOAS measurements of NO2, HCHO and aerosols and evaluation of corresponding satellite data products over Mohali in the Indo-Gangetic plain' reports long term MAX-DOAS observations of AOD, NO2 and HCHO from Mohali, a suburban site in the Indo-Gangetic plain covering the period of January 2013 – June 2017. The MAPA algorithm is used to retrieve vertical profile and

vertical column densities (VCD) and the results are discussed in detail. Seasonal and annual trends along with diurnal variation of vertical profiles are discussed and inter-comparison of the MAX-DOAS measured AOD, NO2 and HCHO with satellite observations are reported. Finally, they have compared surface volume mixing ratios of NO2 and HCHO with in situ observations. Whilst the study is strong on analysis, with the methodology explained in detail, the paper is weak on new results. At present, the manuscript as it stands is more appropriate for a methods journal, for example Atmospheric Measurement Techniques. No novel results are presented in terms of improving our understanding of the chemistry or physics of the region, or of the two chemical compounds and their impacts. The main conclusions of the paper are twofold – presenting an inter-comparison with satellite products, which is a methodology based conclusion and second that ozone production is sensitive to NOx and VOCs in winter, but more to NOx during the other seasons – not a novel result considering the past publications globally and in this region. Even the methods section is not new considering it has been developed in the past by some of the co-authors and has already been used in different parts of the world.

Hence, I would reject the current paper and encourage submission in a methods journal or ask the authors to focus more on the observations rather than the observation methodology to decipher novel results. At present the manuscript, although replete with instrument and retrieval details, does not include significant results on atmospheric chemistry.

Specific comments:

Measurement technique:

1. Why did the detector temperature have to be adjusted for different seasons? If the temperature was not stabilized for different seasons, would the diurnal temperature change not lead to the same issue? Was the DC and offset measured for different temperatures and removed from the spectra according to the ambient temperature?

(line 151-152)

2. 'During the period of measurement, the horizon in the viewing direction was deter-mined by a residential building with a height of about 40m at a distance of 3 km.' - This part is not clear. Does it mean the line of sight has an obstacle within 3 km? Is this why a 1 degree elevation angle was not used? (line 164-165)

3. What was the calibration process for the instrument? Did the authors perform any spectral calibration?

4. 'Based on the measured radiances at 360 nm, colour index (ratio of measured radiances at 330 and 390 nm) and measured O4 airmass factors (O4 SCD/ O4 VCD), we can classify the sky conditions into the following seven categories:' The difference between the upper and lower wavelengths for the classification is small. Why have the authors have not used radiances from ends of the measured window? (Line 204-205)

5. The sky conditions defined by colour indexes (CI) were not supported by any other supporting information. Were there any other methods like visual inspection, or com-parison with sky images done for validation? Without validation how can the CI be used for cloud classification – especially since it is based only on radiances? What were the thresholds used for the classification – what determines a cloud hole as compared to broken clouds? Although the authors have cited some past work, that is from different parts of the world, with very different aerosol loading and SZAs – this information is missing from appendix B. (Line 206-207)

6. What was the threshold for RMS used for filtering the QDOAS analysis? Was there a reason for using this filter?

7. 'This value was derived as the mean of the Ångström exponent (AE) between 470-550 nm measured by MODIS for the measurement period, where we do not observe a strong intra-annual variation (Fig. D3)' The calculation of the AE value from MODIS poses two concerns: The wavelength range is not same and satellite instrument viewing geometry is different than ground based observation. The overpass times of the satellite will also determine what AE is measured, which will change drastically in places with high aerosol loading, or in seasons of biomass burning etc. Why not to use any ground based AOD observations if available – if not, what is the sensitivity of MAPA to the AE values used? (Line 246-247)

8. Were there any radiosonde or BL height measurements available? This would add to the discussion on the differences in the in situ and MAX-DOAS profiles and also to the seasonal variation.

Chemistry:

9. 'Fig. 11 shows the afternoon time (12:30-14:30) monthly mean HCHO/NO2 ratio calculated using the MAX-DOAS observations. We observe that in winter months, mean daytime HCHO/NO2 ratios between 1 and 2 are observed, which represents sensitivity towards both NOx and VOCs.'- The HCHO/NO2 ratio was calculated only for 12:30-12:30 hrs. What was the ratio during rest of the day – emissions would show a diurnal profile, affecting this ratio. (line 650-655)

10. 'However, these are comparable to previous in situ NO2 VMR measured for a period of more than one year at urban and suburban locations of India (e.g. Mohali 8.9 ppb, Pune ∼9.5ppb and Kanpur 5.7ppb)(Gaur et al., 2014;Kumar et al., 2016).'What about other locations from India such as Debaje and Kakade (2009) and Beig et al. 2007. There are many other observations of NO2 and HCHO from urban and suburban regions in India. Please update the ciations.

11. The explanation for higher mixing rations in the MAX-DOAS as compared to in situ observations is not satisfactory. The fact that both the species how higher values for the MAX-DOAS are indicative of an instrumental bias, rather than source regions as speculated in the paper. If the authors are convinced that the power plant, or VOC degradation at higher altitudes contributes to this effect, it can be checked using air mass back trajectories.

12. Figures D6 and D7 need to be discussed in more details in terms of the chemistry leading to large diurnal differences between the in situ and MAX-DOAS observations.

---

## Author Comment (AC1) · 3 Sep 2020

We thank the reviewer for her/his very encouraging comments regarding the significance of the submitted work and positive feedback with a strong recommendation for final publication. We respond to the specific comments below, where the reviewers' comments are marked in blue, our responses are shown in black, and the modification in the manuscript is shown in red.

This paper investigates the temporal variation of the vertical distributions of aerosols, NO2, and HCHO and corresponding AOD and vertical column densities (VCDs) retrieved from MAX-DOAS observations performed at Mohali, north-west Indo Gangetic plain. The measurements presented in the study cover the period from January 2013 to June 2017. The different factors driving the seasonal and diurnal variations of the above parameters are identified and extensively discussed. The MAX-DOAS data sets are also used to validate co-located observations from the OMI and MODIS satellite instruments. This paper is well written, clearly structured, and presents very interesting results which fit well with the scope of ACP. In addition, I would like to say that for me this study is a breakthrough in the evaluation of the air quality in India using ground-based and space-borne remote sensing observations. I therefore strongly recommend the final publication of the manuscript after addressing the following comments:

We thank the reviewer for her/his encouraging feedback and highlighting the importance of our work.

Page 3, lines 97-98: It is written that the two stationary MAX-DOAS measurements over India did not report trace gas VCDs? What do they report then? Only aerosol measurements? The locations of those stationary MAX-DOAS measurements are also not clear.

The previous two stationary MAX-DOAS measurements over India were reported from Pantnagar (29.03° N, 79.47° E), (Hoque et al., 2018) and Barkachha (25.06°N, 82.59°E) (Biswas et al., 2019) respectively. We have modified the corresponding text in manuscript to make the locations of these measurements clearer (lines 65-68 of the revised manuscript):

'The few studies are limited only to four days of mobile measurement around Delhi (Shaiganfar et al., 2011) for estimation of NOx emission from Delhi and satellite validation and more recently at a suburban site Pantnagar (29.03° N, 79.47° E), (Hoque et al., 2018) and a rural site Barkachha (25.06°N, 82.59°E) in the Indo Gangetic plain (Biswas et al., 2019).'

The latter two studies focussed primarily on surface VMRs of $NO_2$, HCHO and CHOCHO. For example, the study from Pantnagar reports the near-surface mixing ratio of $NO_2$, HCHO and Glyoxal without providing the trace gas dSCDs and aerosol information. The study from Barkachha reports the dSCDs and "path average" surface concentration of $NO_2$ and HCHO derived from MAX-DOAS measurements at 3° elevation angle. We have modified lines 97-99 of the original manuscript to include this information (lines 101-103 of the revised manuscript).

"The two stationary MAX-DOAS measurements so far over India focussed primarily on surface volume mixing ratios (VMRs) and have not reported the VCDs of trace gases, and hence lack intercomparison with the satellite observation."

Page 5, lines 160-163: You should add in the legend of Fig D2 to which day those horizon scan results corresponds? Is the steep increase in the measured intensity always centred between 0° and 0.3° during the 4.5 years of measurements?

We have updated Fig D2 of the original manuscript (Fig. F3 of the revised manuscript) to add the day of the horizon scan measurements (17-09-2014). Concerning the second question, we would like to mention that we did not always observe a steep increase in measured intensity centred between 0° and 0.3°. Our method of horizon scans yields the most accurate results in clear sky conditions. In the presence of low lying and rapidly changing clouds, abrupt changes in intensity are observed, whereas in the presence of fog a steep increase in intensity is not observed, resulting in poor performance of this method under such conditions (Donner et al., 2020). During the 4.5 years of measurements, the $5^{th}$ and $95^{th}$ quantiles of the variation of horizon position determined by measured intensity variation at 404nm were -0.22° and 0.46°, respectively. Here it should be noted that part of this variability is probably related to changing weather conditions.

Page 6, line 176 (Eq. 1): How do you select SCD90? Do you use the zenith dSCD of the scan for correcting all the off-axis dSCDs, or do you interpolate the zenith dSCD at the time of the off-axis

measurements of a scan by using the zenith dSCDs just before and just after those off-axis measurements? From my experience, it can have
an impact on the resulting off-axis dSCDs, especially in the case of HCHO.
We thank the reviewer for her/his question about this important detail. We have chosen the zenith dSCDs of the off-axis measurements of a scan by interpolating the zenith spectra before and after the elevation sequence at the time of off-axis measurement. We have also added this information in the revised manuscript (lines 183-185).

"For analysing the off-axis spectra measured at time 't', we calculate the FRS at the time of the measurement by interpolating the zenith spectra measured before and after the complete measurement sequence."

Page 14, lines 447-448 and next paragraphs on page 15: are there any measurements of the boundary layer height at Mohali? It could be an added value to the discussion. If no measurements exist, maybe ECMWF Era-interim BLH could be used.
We thank the reviewer for her/his suggestion. Unfortunately, measurements of boundary layer heights are not available at Mohali. Following the reviewer's recommendation, we now use the boundary layer height from ERA5 land reanalysis data. It should be noted that there are various parametrisations for calculation of boundary layer height, and ERA5 uses the bulk Richardson method recommended by Seidel et al. (2012), which is based on datasets from Europe and the United States. In the revised Figure 6, we also show the diurnal evolution of BLH in the four major seasons, as well as the monthly variation of the afternoon time boundary layer height. Additionally, we show the diurnal evolution of the profile heights of aerosol, $NO_2$ and HCHO in the appendix (Fig. F10). While performing these comparisons, we realised that for HCHO, often unrealistic values at high SZA occur, which is probably related to the spectral interference with ozone absorption. Hence in the revised manuscript, we limit the analysis of the HCHO profiles results to SZA < 60°. We mention this in lines 259-262 of the revised manuscript:
"For the HCHO profile inversion, we observed unrealistic h and s at high solar zenith angles (SZA> 60°), which are probably related to spectral interferences with the ozone absorption within the DOAS analysis. Therefore, we only consider HCHO profile results for measurements with SZA less than 60°."
We have revised lines 446-450 and 461 of the original manuscript as follows to append this new information (lines 407-425 and 436-444 of the revised manuscript)
 "The vertical profile of aerosol extinction is expected to be primarily driven by the boundary layer height (BLH) and to some extent, the photochemistry, which eventually drives secondary aerosol formation (Wang et al., 2019). At Mohali, the diurnal evolution of the aerosol extinction profile heights reaches its maximum during afternoon hours. In Fig. 6A, we show the typical diurnal evolution of BLH from the ERA5 reanalysis data for the four major seasons. We observe a growth of the BLH from morning until noon with a maximum at 14:00 L.T. and a subsequent decline. The maximum BLH up to 3 km is observed in summer. Shallow daytime BLH up to 1.2 km are observed in the monsoon period due to overcast sky condition, stronger wind and high surface moisture, and in winter due to low surface temperature and low surface heat flux (Sathyanadh et al., 2017). We observe that the aerosol is trapped in the bottom layers (within 400m) in winter, whereas during the afternoon hours in summer, monsoon and early post-monsoon months, aerosol extinction up to 0.2 $km^{-1}$ is observed even at around 1.5 km altitude. Though the ERA5 BLH is shallow in monsoon, yet we observe similar aerosol profiles during that period as those during summer, which indicates that at Mohali, the vertical distribution of aerosol does not follow ERA5 BLH transition from summer to monsoon. Over India, the monsoon months are characterised by strong convective activity which can bring the surface air aloft to several km despite a shallow ERA5 BLH (Lawrence and Lelieveld, 2010). The convection is rather strong in the Himalayan foothill region (which also includes Mohali) and pumps the surface pollutants even into the UTLS (upper troposphere/lower stratosphere) (Fadnavis et al., 2015). The evidence of pollutant transport associated with deep convection is crucial for PAN formation in the UTLS, which is observed by the modelling studies over the IGP and Himalayan region. Long-lived non-methane VOCs (e.g. ethane) can be transported to the UTLS

where both convective transported $NO_x$ from the surface and exchanged from stratosphere serve as fuel for PAN formation."

"We show the diurnal evolution of characteristic profile heights ($H_{75}$) in Fig. F10 for the four major seasons. Fig. 6B shows the mean afternoon time characteristic profile heights ($H_{75}$) for aerosol, $NO_2$ and HCHO for different months, together with the mean ERA5 BLH. Due to their short atmospheric lifetime (< 6 hours) during daytime, $H_{75}$ for $NO_2$ and HCHO are lower than those for aerosol. $H_{75}$ for the measured species are observed to be smaller than the typical boundary layer heights. In the monsoon season, we observe $H_{75}$ comparable to those in summer, even though the boundary layer height is shallow and comparable to that in winter. Trace gases and aerosol from the surface are lofted up due to deep convection in the monsoon leading to high $H_{75}$. This indicates that the vertical mixing of aerosol during the monsoon is not driven by the parameters used to calculate the ERA5 BLH, but rather follows the trend of ambient daytime temperature, which does not show such large difference between summer and monsoon (e.g. Fig. S2 of Kumar et al. (2016))."

[Figure]

Figure 6: A) Diurnal evolution of hourly means ERA5 boundary layer height (BLH) at Mohali for the four major seasons of the year. B) Mean afternoon time (between 12:00 and 15:00 Local time) profile height (with 75% of the total amount below) for aerosols, $NO_2$ and HCHO and the ERA5 BLH for different months. The upper and lower vertical error bars represent the monthly variability as 75th and 25th percentiles, respectively.

[Figure]

Figure F10: Diurnal variation of characteristic profile heights of aerosol (top panel), NO$_2$ (centre panel) and HCHO (bottom panel) for the four major seasons.

Page 18, line 562 and page 23, line 723: MAX-DOAS VCD measurements are spatially representative of a few kilometres in the field of view and this horizontal sensitivity strongly depends on the aerosol load. I think it would be useful to have to estimate this horizontal sensitivity for the main sky conditions presented in Figure 3. Numerous studies have also shown that taking into account this horizontal sensitivity in the selection of the co-located satellite data can have an impact on the agreement with ground-based MAX-DOAS observations. This is especially the case in the present study where MODIS data are given on a 1x1 km2 grid, i.e. at a horizontal resolution which is significantly higher than the typical horizontal distance of several kilometres representative of the MAX-DOAS measurements.

We thank the reviewer for her/his suggestion to estimate the horizontal sensitivity of MAX-DOAS. The horizontal sensitivity distance (HSD) of MAX-DOAS measurements can be estimated as the e-folding distance of $O_4$ differential box airmass factors at the location of the instrument.

For elevation angles smaller than 6°, Wagner and Beirle (2016) have introduced a 2D function of solar zenith angle and relative azimuth angle to derive an estimate of the HSD from the measured $O_4$ dAMF. Hence using the MAX-DOAS measured $O_4$ dSCD, we have calculated the HSD for clear sky cases with low aerosol (AOD < 0.85 at 330nm) and clear sky cases with high aerosol (AOD > 0.85 at 330nm) for 2° elevation angle.

[Figure]

Figure F14: Box and whiskers plot showing the horizontal sensitivity distance of MAX-DOAS measurements during afternoon hours (between 12:00 and 15:00 local time) for 2° elevation angle. The blue boxes represent clear sky conditions with low aerosol load, and the orange boxes indicate clear sky conditions with high aerosol load.

We observe that for clear sky cases with high aerosol load, the mean daytime HSD is in the range 5-7 km, whereas, for high aerosol load cases, it is in the range of 3-6 km. The horizontal sensitivity is maximum in the summer months. Here it is important to note that this estimate is mainly representative for the near-surface layers.

For the comparison with the satellite data sets, the vertically integrated quantities are used. For MAX-DOAS observations, these quantities are mainly constrained by the high elevation angles (≥ 30° degree). For such high elevation angles, the sensitivity range is much closer to the instrument (at distances up to 1 and 2 km for layer height of 0.5 and 1km, respectively (see Fig 6).

Accordingly, we have modified lines 562-563 of the original manuscript as follows (lines 545-554 of the revised manuscript):

"MAX-DOAS measurements are spatially representative of a few kilometres in the field of view, depending on the ambient aerosol load and elevation angle, whereas the ground footprints of individual OMI pixels are 13×24 km$^2$ in the best case. We have calculated the horizontal sensitivity distance (HSD) of MAX-DOAS for low elevation angles as the e-folding distance of $O_4$ dAMF from the instrument location (Wagner and Beirle, 2016). Fig. F14 shows that the mean afternoon time (between 12:00 and 15:00 local time) HSD ranges between 5 and 7 km for clear sky condition with low aerosol load and between 3 and 6 km for high aerosol conditions. Here it is important to note that this estimate is mainly representative for the near-surface layers. For the comparison with the satellite data sets, the vertically integrated quantities are used. For MAX-DOAS observations, these quantities are mainly constrained by the high elevation angles (≥ 30°). For such high elevation angles, the sensitivity range is much closer to the instrument (at distances up to 1 and 2 km for layer height of 0.5 and 1 km, respectively) (see Fig 6A)."

And added Figure F14, as shown above in response:

In the original manuscript, we have retained MODIS AOD measurements within 2 km of Mohali for comparison with MAX-DOAS measurements. Incorporating the reviewer's suggestion, we performed

a sensitivity study by retaining MODIS measurements within 5 km of Mohali for comparison with MAX-DOAS AOD measurements (see figure below). However, this did not bring a noticeable change in the agreement as shown in the figure below (compared to Fig 7 of the original manuscript).

[Figure]

Page 21, lines 646-648: Since satellite total HCHO AMFs and averaging kernels are missing in the files, you could proceed the other way round for eliminating the difference caused by the non-representative satellite a priori HCHO profiles, i.e. recalculating satellite AMFs using MAX-DOAS vertical profiles and dividing the satellite slant column
densities by those new AMFs (see e.g. De Smedt et al., Atmos. Chem. Phys., 15, 12519–12545, 2015). Maybe something worth to try.

We thank the reviewer for her/his suggestion for recalculation of satellite AMFs using MAX-DOAS vertical profiles. This could easily be done if averaging kernels of MAX-DOAS measurements were known (e.g. from profile inversion using optimal estimation method). However, for parametrised profile inversions (e.g. MAPA), as used in this study, averaging kernels are not provided. We have indicated the limitation of our approach in lines 613-616 of the original manuscript.

*"A different approach for improved agreement between MAX-DOAS and satellite VCDs is by using the MAX-DOAS $NO_2$ profiles as a priori profiles for the calculation of airmass factors for the satellite retrieval (Chan et al., 2019). However, such an approach was not possible in our study because for parameter-based profiles inversion (like MAPA), no averaging kernels are provided"*

Page 50, Figure D4: How MODIS would compare to the AERONET sun photometer measurements at these two sites? Angstrom exponent could be used to convert MODIS AOD from 470 nm to 360 nm.

We observe very good agreement between AERONET and MODIS AOD measurements at Lahore for all the months of the year. For New Delhi, though the agreement is good for most of the months, a larger scatter in the data is observed due to a smaller sample size of AERONET measurements. AERONET measurements are available at 440nm, and we have used the Ångström exponent to convert MODIS AOD from 470nm to 440nm.

In the revised manuscript, we now also show the seasonal variation of MODIS AOD measurement at these two sites together with AERONET measurements in Figure D4. Additionally, we also show the scatter plot for agreement between AERONET and MODIS AOD observation. Please note that, in the revised Fig F5, now we only use the AERONET measurement between 9:30 and 11:30 and between 12:30 and 14:30 local time to ensure consistency with the MODIS overpass times.

[Figure]

Figure F5: Monthly variation of the AOD (at 440nm) as observed by AERONET sun photometers (red boxes) and MODIS (black boxes) at two sites (A. Lahore and B. New Delhi), which are the nearest stations to Mohali in the Indo-Gangetic Plain. The bottom panel shows the corresponding scatter plots indicating the agreement in the daily MODIS and AERONET measurements.

Technical corrections:

Page 1, line 11: 'We investigate the temporal variation and the vertical profiles. . .'. I find the sentence a bit misleading since you also investigate the temporal variation of the vertical profiles (-> see Figure 5). Maybe some rephrasing is needed here.

Many thanks for this hint. We have rephrased line 11 of the original manuscript to the following (lines 11-13 of the revised manuscript):

"We investigate the temporal variation of tropospheric columns, surface volume mixing ratio (VMR) and vertical profiles of aerosols, $NO_2$ and HCHO and identify factors driving their ambient levels and distributions for the period from January 2013 to June 2017."

Page 2, line 59: 'ground based' -> 'ground-based'; There are some other places where this should be corrected too.

We thank the reviewer for this suggestion for consistency. We have corrected it at lines 59 and 306. At other places in the manuscript, it was written as "ground-based" in the original manuscript.

Page 5, line 138: Multi Axis Differential Optical Absorption Spectroscopy (MAX-DOAS).
Done.

Page 14, line 444: 'diurnal trends' -> 'diurnal variations' ?
Done.

Page 28, Figure 1: The names of the cities and the x and y axes labels are difficult to read. Maybe you could use a larger font size.
Done. We have increased the font size and axis label sizes from 10 to 14.

Page 37, Figure 13: in the legend, it should be 'Daily mean HCHO mixing ratios. . .' and
Thanks for spotting this. We have corrected it in the revised manuscript.

References:

Donner, S., Kuhn, J., Van Roozendael, M., Bais, A., Beirle, S., Bösch, T., Bognar, K., Bruchkouski, I., Chan, K. L., Dörner, S., Drosoglou, T., Fayt, C., Frieß, U., Hendrick, F., Hermans, C., Jin, J., Li, A., Ma, J., Peters, E., Pinardi, G., Richter, A., Schreier, S. F., Seyler, A., Strong, K., Tirpitz, J. L., Wang, Y., Xie, P., Xu, J., Zhao, X., and Wagner, T.: Evaluating different methods for elevation calibration of MAX-DOAS (Multi AXis Differential Optical Absorption Spectroscopy) instruments during the CINDI-2 campaign, Atmos. Meas. Tech., 13, 685-712, 10.5194/amt-13-685-2020, 2020.

Wagner, T. and Beirle, S.: Estimation of the horizontal sensitivity range from MAX-DOAS O4 observations, Tech. rep., QA4ECV, 2016

---

## Author Comment (AC2) · 3 Sep 2020

We would like to thank the reviewer for her/his encouraging comments regarding the quality of the submitted work and suggestions. We respond to the specific comments below, where the reviewers' comments are marked in blue, our responses are shown in black, and the modification in the manuscript is shown in red.

The paper titled 'Long term MAX-DOAS measurements of NO2, HCHO and aerosols and evaluation of corresponding satellite data products over Mohali in the Indo-Gangetic plain' reports long term MAX-DOAS observations of AOD, NO2 and HCHO from Mohali, a suburban site in the Indo-Gangetic plain covering the period of January 2013 – June 2017. The MAPA algorithm is used to retrieve vertical profile and vertical column densities (VCD) and the results are discussed in detail. Seasonal and annual trends along with diurnal variation of vertical profiles are discussed and inter-comparison of the MAX-DOAS measured AOD, NO2 and HCHO with satellite observations are reported. Finally, they have compared surface volume mixing ratios of NO2 and HCHO with in situ observations. Whilst the study is strong on analysis, with the methodology explained in detail, the paper is weak on new results. At present, the manuscript as it stands is more appropriate for a methods journal, for example Atmospheric Measurement Techniques. No novel results are presented in terms of improving our understanding of the chemistry or physics of the region, or of the two chemical compounds and their impacts.

We thank the reviewer for her/his assessment and suggestion. With respect to the suggestion to submit the paper to another journal, we are sorry if the novelty of the results and the new findings got diluted by the emphasis on the technical description pertaining to methodology. To address this concern of the reviewer and strengthen the results of the paper, we have now shifted several technical aspects to the appendix and improved the discussion of the novel results (for details, see below). We would like to add that like many other esteemed papers published in ACP, our paper contains methodological aspects (which are important for a thorough understanding, especially as such studies are lacking from the region) and also many results with relevance for atmospheric chemistry and physical processes.

In our opinion, the most important novel aspects of our study are the following:

a)  In addition to the first MAX-DOAS measurements of $NO_2$ and HCHO vertical column densities from this under-represented yet crucial part of the world, we report in particular the vertical profiles of aerosol, $NO_2$ and HCHO for the first time from the Indo Gangetic plain (and India). This is important information for several aspects of atmospheric chemistry and physics including atmospheric modelling, satellite retrieval and understanding atmospheric dynamics. The application of vertical profile includes calculation of airmass factors required for conversion of slant column densities (SCD) to vertical column densities (VCD), understanding the atmospheric chemistry at higher altitudes, understanding atmospheric dynamics, medium and long-range transport, evaluation of the vertical distribution of chemical tracers in atmospheric models.

A major fraction of the $NO_2$ column was found to be located in the bottom-most layer extending from surface until 200 meters in all the seasons. We show that during summer and monsoon seasons, there is a significant fraction of formaldehyde present at intermediate layers (between 200 and 600m altitudes) and sometimes even higher than the surface indicating an active photochemistry at these layers. Following the reviewer's feedback, we now also discuss the vertical distributions with respect to the ERA5 boundary layer height (BLH) and found that the seasonal trends in the derived vertical distributions are not strongly influenced by the BLH. Monsoon season is particularly interesting as the pollutants from the surface are lifted up to higher altitudes due to deep convection even though the ERA5 BLH are shallow. $NO_x$ and VOCs transported to high altitudes can be transported to a larger area and can also participate in secondary chemistry producing reservoir species such as PAN.

b)  We analyse the annual, seasonal and diurnal profiles of AOD, $NO_2$ and HCHO. We acknowledge that several AOD measurements and in situ measurements of $NO_2$ have been reported from this region. However, HCHO (which is primarily a secondary photo-oxidation product and serves as an indicator of photochemical activity and VOCs) has rarely been reported. By performing analyses of seasonal and diurnal trends of AOD, $NO_2$ and HCHO,

we identify sources and chemical processes that drive their ambient levels. The sources of HCHO are indentified to be quite contrasting from majorly photochemical from biogenic and anthropogenic sources in summer, monsoon and early post monsoon to primary anthropogenic in winter and late post monsoon. We show that even though the region around the measurement location has undergone urbanisation, an obvious trend was not observed in AOD, $NO_2$ and HCHO for more than four years of measurements.

c) Using the measured HCHO and $NO_2$ VCDs, we show that ozone production is sensitive to $NO_x$ and VOCs in winter, but shifts towards $NO_x$ in summer. This analysis was originally performed for the peak daytime hours, which overlaps with OMI overpass and generally when the maximum in the diurnal profiles of ozone is observed. Following the reviewer's suggestion, we have extended this analysis for the morning and late afternoon hours, which strengthens the observation part of the manuscript.

The main conclusions of the paper are twofold – presenting an inter-comparison with satellite products, which is a methodology based conclusion and second that ozone production is sensitive to NOx and VOCs in winter, but more to NOx during the other seasons – not a novel result considering the past publications globally and in this region. Even the methods section is not new considering it has been developed in the past by some of the co-authors and has already been used in different parts of the world.
Hence, I would reject the current paper and encourage submission in a methods journal or ask the authors to focus more on the observations rather than the observation methodology to decipher novel results. At present the manuscript, although replete with instrument and retrieval details, does not include significant results on atmospheric chemistry.

a) We understand the reviewers' concern that intercomparison with satellite observation is primarily method based. In our view, putting the technical rigour for the measurements and intercomparison, which are first of its kind from this region is, however, crucial for the manuscript. Keeping in view that several studies, especially over India, use only satellite observations to draw important conclusions regarding $NO_x$ and VOC emissions, ozone production control, VOC source identification, long term trend analyses (e.g. Ghude, et al. 2008, Surl, et al. 2018, Chaliyakunnel, et al. 2019, Hilboll, A., et al. 2013), the evaluation of various satellite data products is crucial for improving the understanding of atmospheric chemistry and physics over the IGP. This has not so far been done for the IGP (lines 76 -92 of the original manuscript). Our study provides the first evaluation of three OMI $NO_2$ data products and two OMI HCHO data products over the IGP.

b) Quantitative evaluation of the sensitivity of ozone production in different seasons over India for a period longer than one year has been reported by Mahajan et.al., 2013 using SCIAMACHY observations for the mean of the years 2002-2013, and by Kumar et al. 2010 and Sharma et al. 2016 using WRF model for the year 2008 and 2010, respectively. Recognising the rapid urbanisation and industrialisation in the IGP, the sensitivity might change, and our observations provide a crucial update for the same. Moreover, the unique feature of our study is that we calculate the sensitivity using ground-based observations as opposed to the previous studies using SCHIMACY observation (coarse resolution, limited sensitivity close to the ground) and model simulations (which rely on coarse resolution and uncertain emission inventories in the region).

In order to focus more on the observations and their interpretation, we have addressed the specific comments of both the reviewers and restructured the manuscript in the following way:

1. We have made section 2.5 more concise and moved technical details about the various satellite data products in the appendix.

2. We have modified Figure 6 to also include boundary layer height from ERA5 and included its discussion with respect to characteristic profiles heights and vertical distribution of aerosol in section 3.2.
3. We have included a comparison of MAX-DOAS surface VMRs of $NO_2$ and HCHO to available previous works from India.
4. We have included the $HCHO/NO_2$ ratio for the morning and late afternoon hours in Figure 11 and the relevant discussion in section 3.6.
5. We have restructured section 3.7 to focus more on the interpretation of the retrieved surface concentrations of HCHO and $NO_2$ and moved technical details about the intercomparison with in situ observations to the appendix. We have also shown the seasonal variation of the surface VMR of $NO_2$ and HCHO in the insets of Figures 12 and 13.
6. We have modified the abstract to focus more on the novel findings.

Specific comments:
Measurement technique:
1. Why did the detector temperature have to be adjusted for different seasons? If the temperature was not stabilised for different seasons, would the diurnal temperature change not lead to the same issue? Was the DC and offset measured for different temperatures and removed from the spectra according to the ambient temperature? (line 151-152)

The detector temperature **was stabilised** using a Peltier cooler and set to values such that the following two conditions (lines 149-151 of the original manuscript) are met:
1. The detector temperature is lower than the ambient temperature.
2. The difference between the ambient temperature and detector temperature is not more than 20 ºC.

The complete mini-MAX DOAS instrument was installed in the open, and we had to consider large variation in ambient temperature to ensure a manageable workload on the Peltier cooler. The ambient temperature in Mohali ranges from less than 5 °C in winter to up to higher than 40 °C in summer, but the amplitude of diurnal temperature variation is typically less than 20 °C. Hence, **we did not need to adjust the detector temperature to account for the diurnal temperature change**. The dark current and offset measured for the different temperatures were removed from the spectra according to the respective detector set temperatures. This information was provided in the original manuscript in lines 151-155. However, to make it clearer for the readers, we have changed to the following in the revised manuscript (lines 157-158):

"The dark current and offset spectra were recorded every night, and while performing the spectral analysis, these were subtracted from measured spectra recorded at similar detector temperature."

2. 'During the period of measurement, the horizon in the viewing direction was determined by a residential building with a height of about 40m at a distance of 3 km.' - This part is not clear. Does it mean the line of sight has an obstacle within 3 km? Is this why a 1 degree elevation angle was not used? (line 164-165)

One of the crucial steps for setting up a MAX-DOAS measurement is the elevation calibration. We performed the elevation calibration using the horizon scan method as described by Donner et.al., 2020. One of the pre-requisites of this method is the knowledge of the approximate horizon. Several residential buildings of the city of Mohali and Chandigarh lie in the viewing direction of the MAX-DOAS instrument, and hence the first estimate of the horizon was calculated using a tall building in the field of view. The MAX-DOAS instrument is installed at an altitude of 20 m above ground level. Hence, a 40m high building at 3 km distance would correspond to an angle of 0.38°.

We realise that by mistake, in the manuscript, we write the angle of the visible horizon to be about 0.2°. We apologise for it and correct it in the revised manuscript in line 168. Please note that, even after this correction, the visible horizon and that determined from using the horizon scan are close to each other, and further correction is not required.

The reviewer is right that this is also the reason why the measurements at 1° elevation angle were not used. As can be seen from Fig. F3, the field of view (FOV) of the instrument is rather large, and typically the RMS of the spectral analysis for the measurements at 1° elevation is substantially larger

than those for the higher elevation angles. This indicates that these measurements are still affected by the reflected light from the surface. Therefore, we excluded measurements at 1° elevation angle from further processing.

We have modified lines 145-146 of the original manuscript and added lines 170-172 in the revised manuscript to include this information:

"The scattered sunlight spectra were recorded for elevation viewing angles 1º, 2º, 4º, 6º, 8º, 10º, 15º, 30º and 90º at a total integration time (number of scans × acquisition time for one scan) of 60 seconds each."

"We also see from Fig. F3 that the field of view (FOV) of the instrument is rather large (> 0.7°), and typically the RMS of the spectral analysis for the measurements at 1° elevation is substantially larger than those for the higher elevation angles. Hence, we excluded the measurements at 1° elevation angle from further analyses."

3. What was the calibration process for the instrument? Did the authors perform any spectral calibration?

The spectral calibration is performed with respect to a high resolved Fraunhofer spectrum. This information is provided in lines 157-160 of the original manuscript.

*"Wavelength to pixel calibrations were performed in QDOAS software (http://uvvis.aeronomie.be/software/QDOAS/: last access 05.03.2020) (Danckaert et al., 2012) every time the detector temperature was changed, by matching the structures in a measured spectrum in the zenith direction at around noontime with those in a highly resolved solar spectrum"*

4. 'Based on the measured radiances at 360 nm, colour index (ratio of measured radiances at 330 and 390 nm) and measured O4 airmass factors (O4 SCD/ O4 VCD), we can classify the sky conditions into the following seven categories:' The difference between the upper and lower wavelengths for the classification is small. Why have the authors have not used radiances from ends of the measured window? (Line 204-205)

In principle, a wavelength pair with a larger difference between the lower and upper wavelength could be used for the cloud classification (e.g. 320 nm/440nm) which is close to the ends of the measured window. However, the chosen wavelength pair has two advantages (Wagner et. al., 2016):

1. The absorption effect of atmospheric ozone is smaller at a longer wavelength, and hence a longer wavelength (e.g. 330 nm) is more robust to variability in ozone.
2. The variability of the surface reflectance is smaller for shorter wavelength (e.g. 390 nm) as compared to longer wavelengths (e.g. 440 nm). Hence a global threshold is more robust for 330nm /390 nm wavelength pair.
3. The signal to noise ratio of the measured spectra is rather high, while the changes of the CI caused by clouds and aerosols are rather strong. Thus, the limitation of the spectral range is not critical.

5. The sky conditions defined by colour indexes (CI) were not supported by any other supporting information. Were there any other methods like visual inspection, or comparison with sky images done for validation? Without validation how can the CI be used for cloud classification – especially since it is based only on radiances? What were the thresholds used for the classification – what determines a cloud hole as compared to broken clouds? Although the authors have cited some past work, that is from different parts of the world, with very different aerosol loading and SZAs – this information is missing from appendix B. (Line 206-207)

Comprehensive validation of the cloud classification scheme was performed in earlier studies (Wagner et al., 2014, 2016; Wang et al., 2015). In these publications, the detailed description for the calibration of the thresholds of the CI and their dependencies on elevation angles and time are also given, which are applied in this study.

We think that the detail of the method need not be repeated in our study, because they are well documented in Wagner et. al., (2016). But we added some more information about the general idea of the algorithm (see below).

The cloud classification scheme is based on the measured radiances at 360 nm, colour index (ratio of measured radiances at 330 and 390 nm) and the measured $O_4$ airmass factors ($O_4$ SCD/ $O_4$ VCD) (for details see Wagner et al. (2016)). Besides the absolute values of these quantities, also their temporal variation and their elevation dependencies are considered. For the analyses presented in this manuscript, identification of thick clouds and fog was most important, as we retain DOAS measurements corresponding to sky condition without thick clouds and fog. The identification of thick clouds and fog are performed according to measured radiance and $O_4$ AMF, respectively. The thresholds for normalised measured radiances are calculated specifically for our site (Appendix B). The thresholds for the spread of $O_4$, normalised CI, the spread of the CI and the temporal variation of the CI are calculated using SZA dependent polynomials provided in Wagner et al. (2016) (Table 1). We agree with the reviewer that absolute values of the radiances and CI would vary for different parts of the world depending on several factors which include aerosol conditions and spectrometer characteristics. In order to account for the spectrometer characteristics, calibration of CI is performed to get a proportionality constant $\beta$, which relates measured CI ($CI_{meas}$) to the calculated CI ($CI_{cal}$).

$$CI_{cal} = \beta . CI_{meas}$$

Only after taking β into account, measured CI is compared to the threshold for cloud classification.

Furthermore, Wagner et al., 2016 have shown that the minimum CI varies only slightly with the atmospheric properties (e.g. AOD). Hence, we first normalise the measured radiance with respect to the corresponding simulated SZA dependent minimum CI. This generally also removes the SZA dependence of CI for SZA < 60°. In the next step, the frequency distribution of the normalised CI for SZA < 60° is plotted. Occurrence of a clear accumulation point (in our case shown in the figure below) similar to Wagner et. al., 2016 shows that CI can be used for cloud classification for our location also.

[Figure]

The maxima of the frequency distribution represent the inverse of $\beta$, which can be used to derive the calibrated CI. The calibrated CI can be directly compared to the thresholds. Please note that we use SZA dependent threshold values which also accounts for the variability in the SZA.

The procedure of CI calibration is already depicted in detail by Wagner et. al., 2016 in section 2 and we think that describing the calibration procedure again into our manuscript is beyond the scope of the article considering the reviewer's recommendation to focus more on the interpretation of results rather than technical details.

The reviewer is, of course, right, in that the aerosol type is probably different for the measurement of our study compared to other places. As a consequence, the classification of aerosols by the cloud

classification might be slightly different compared to other places. But this is not critical here, because the main aim – the cloud classification – is hardly affected by these differences.

Following the reviewer's concerns, we modify lines 211-213 of the original manuscript to provide further details about the cloud classification and add lines 228-230 in the revised manuscript to mention the effect of aerosol properties:

"The cloud classification scheme is based on the measured radiances at 360 nm, colour index (ratio of measured radiances at 330 and 390 nm) and the measured $O_4$ airmass factors ($O_4$ SCD/ $O_4$ VCD) (for details see Wagner et al. (2016)). Besides the absolute values of these quantities, also their temporal variation and their elevation dependencies are considered. The thresholds for these quantities (the spread of $O_4$, the normalised CI, the spread of the CI and the temporal variation of the CI) are parametrized as polynomials of the SZA as provided in Wagner et al. (2016)."

"While the classification of aerosols might be slightly affected by the specific properties of the local aerosol, the cloud classification is robust to the variability of aerosol properties. However, this is not critical here, because the main aim – the cloud classification – is hardly affected by these specific aerosol properties."

Broken clouds refer to few cloudy patches in the clear sky, while cloud holes refer to clear sky between clouds. Both cloud holes and broken clouds are detected by a rapid temporal variation of the observed for normalised CI. In the cloud classification algorithm, the normalised measured CI is smaller than the threshold CI for broken clouds, and the inverse is true for cloud holes.

6. What was the threshold for RMS used for filtering the QDOAS analysis? Was there a reason for using this filter?
We thank the reviewer for raising this important question. We have filtered out the $O_4$, $NO_2$ (UV and VIS) and HCHO dSCDs corresponding to a DOAS fit RMS greater than 0.002. Additionally, we filter out all the measurements at solar zenith angles greater than 85°. The RMS threshold was considered according to the recommendation from Wang et.al., 2019 and removed most of the obvious outliers. More precisely, this threshold removes 1.1%, 1.4%, 0.7% and 1.3% of the $O_4$, $NO_2$(UV), $NO_2$(VIS) and HCHO dSCDS respectively, for the elevation angles considered for our analyses.
This information was not present in the original manuscript, and we have added the following in the revised manuscript (lines 175-180 of the revised manuscript):

"The typical values (peak of the frequency distribution) of the root mean square (RMS) of the DOAS fit residuals are around $5\times10^{-4}$, $7\times10^{-4}$, $6\times10^{-4}$, $6\times10^{-4}$, for $O_4$, $NO_2$ (UV), $NO_2$ (VIS) and HCHO, respectively. In order to retain analyses results corresponding to good quality fits, we have excluded the $O_4$, $NO_2$ and HCHO dSCDs corresponding to a RMS greater than $2\times10^{-3}$ and solar zenith angles higher than 85° (Wang et al., 2019). The RMS threshold removes 1.1%, 1.4%, 0.7% and 1.3% of the $O_4$, $NO_2$ (UV), $NO_2$ (VIS) and HCHO dSCDs, respectively, of all the measured dSCDs at solar zenith angles less than 85°."

7. 'This value was derived as the mean of the Ångström exponent (AE) between 470- 550 nm measured by MODIS for the measurement period, where we do not observe a strong intra-annual variation (Fig. D3)' The calculation of the AE value from MODIS poses two concerns: The wavelength range is not same and satellite instrument viewing geometry is different than ground based observation. The overpass times of the satellite will also determine what AE is measured, which will change drastically in places with high aerosol loading, or in seasons of biomass burning etc. Why not to use any ground based AOD observations if available – if not, what is the sensitivity of MAPA to the AE values used? (Line 246-247)
Unfortunately, ground-based AOD measurements are not available around the measurement site. These measurements are only available at ~ 250 km from the measurement site (Lahore and New Delhi) (lines 378-381 of the original manuscript). We had foreseen this limitation and checked the

sensitivity of MAPA to the AE values for a smaller subset of our data. We mention the inference of this sensitivity study in lines 249-253 of the original manuscript:

*"We also investigated the effect of the choice of Ångström exponent on the profile inversion for a smaller subset of our data spanning 15 days. We found that AE values of 1.25 and 1.75 (minimum 5th percentile and maximum 95th percentile in Fig. D3) resulted in same number of valid retrievals and the difference in the mean NO2 VCD was less than 0.1%. The surface NO2 concentration were slightly higher (4%) for AE value of 1.25 and were 3% lower for AE value of 1.75 as compared to those for an AE value of 1.54."*

8. Were there any radiosonde or BL height measurements available? This would add to the discussion on the differences in the in situ and MAX-DOAS profiles and also to the seasonal variation.

Unfortunately, radiosonde or boundary layer height measurements are not available at Mohali. We have added discussion about boundary layer height from ERA5 data. Please see the response to reviewer #1 corresponding to the question regarding Page 14, lines 447-44 for a detailed discussion.

Chemistry:

9. 'Fig. 11 shows the afternoon time (12:30-14:30) monthly mean HCHO/NO2 ratio calculated using the MAX-DOAS observations. We observe that in winter months, mean daytime HCHO/NO2 ratios between 1 and 2 are observed, which represents sensitivity towards both NOx and VOCs.'- The HCHO/NO2 ratio was calculated only for 12:30-12:30 hrs. What was the ratio during rest of the day – emissions would show a diurnal profile, affecting this ratio. (line 650-655)

The HCHO/NO2 ratio provides a metric which discerns the sensitivity of ozone production towards $NO_x$ or VOC. This ratio was calculated for 12:30-14:30 hours, which is crucial for two reasons:

1. The daytime maximum of ozone is usually observed during this time window (Kumar et al., 2016).
2. OMI overpass (and also that of the recent TROPOMI instrument) usually happens in this time window. The $HCHO/NO_2$ ratio can also be calculated from the OMI data product and hence can be evaluated against similar metric calculated using ground-based observation.

We thank the reviewer for highlighting that the ratio might be affected because of emissions which vary on a diurnal scale. We now also calculated this indicator for 09:30-11:30 and 15:30-17:30 hours local time representing morning and late afternoon condition, respectively and revised Fig. 11 and the relevant discussion accordingly.

[Figure]

Figure 11: Monthly mean HCHO VCD/NO2 VCD ratios (triangles) calculated from MAX-DOAS measurements for the morning (09:30-11:30 L.T., red), noon around the OMI overpass time (12:30-14:30 L.T., black) and late afternoon (15:30-17:30 L.T., blue) over Mohali. The lines at the centres of the boxes represent the median; the boxes show the interquartile ranges whereas the whiskers show the 5th and 95th percentile values.

We have modified lines 650-653 and 661-666 of the original manuscript to the following:

"Martin et al. (2004) recommended the use of the ratio of the formaldehyde and $NO_2$ columns from satellite observations as an indicator for the ozone production regime. $HCHO/NO_2$ ratios less than 1 represent a VOC sensitive regime, whereas values greater than 2 indicate a $NO_x$ sensitive regime. Intermediate values of the $HCHO/NO_2$ ratio indicate a strong sensitivity towards both $NO_x$ and VOCs. The threshold for this indicator was initially calculated for afternoon time (between. 13:00 – 17:00 L.T.), but was later extended to also include morning period by Schofield et al. (2006). However, Schofield et al. (2006) also indicated that the upper limit of the intermediate regime might vary spatio-temporally. Nonetheless, higher $HCHO/NO_2$ indicate that reduction in $NO_x$ emissions would be more effective for ozone reduction."

"Fig. 11 shows the monthly mean $HCHO/NO_2$ ratio calculated using the MAX-DOAS measurements for the morning (09:30-11:30 L.T.), noontime around the OMI overpass (12:30-14:30 L.T.) and late afternoon (15:30-17:30 L.T.). We observe a stronger (smaller) sensitivity towards $NO_x$ during the late afternoon (morning) as compared to noontime similar to other urban locations in the USA (Schroeder et al., 2017). VOCs contribute to ozone production via their oxidation by OH radicals and subsequent formation of peroxy radicals. During the build-up hours of ozone (between sunrise until noontime) at Mohali, radicals' abundance is also expected to be limited. Hence, the ozone production is more sensitive to VOC (or "radicals") during morning which shifts towards $NO_x$ later during the day. In winter months, mean daytime $HCHO/NO_2$ ratios between 1 and 2 are observed, which represent sensitivity towards both $NO_x$ and VOCs. The sensitivity of the ozone production regime changes towards $NO_x$ with the onset of summer and stays like that until the end of the post-monsoon season. Over the Indo-Gangetic plain, the strongest ozone pollution episodes are observed in the summer and post monsoon months during the afternoon hours between 12:00 and 16:00 L.T.(Kumar et al., 2016;Sinha et al., 2015). Surface ozone measurements from Mohali have shown enhancement in its ambient concentrations during the late post monsoon as compared to the early post monsoon even though the daytime temperature drops by 6 °C. During summer, enhanced precursor emission from fires lead to an increase in ~19 ppb ozone under similar meteorological conditions. Considering the stronger sensitivity of daytime ozone production towards $NO_x$, the ozone mitigation strategies should focus on $NO_x$ emission reductions."

Additionally, following the general comments by the reviewer, we have modified lines 655-658 in the original manuscript to highlight the importance of this analysis.

"Mahajan et al. (2015) evaluated the ozone production regime over India using the ratio of HCHO and $NO_2$ VCDs observed from SCIAMACHY for the mean of years 2002-2012. Over the north-west IGP, the $HCHO/NO_2$ was observed to be less than 1 in the winter months and between 1 and 2 in all other months. From our intercomparisons in the previous sections, we note that while the OMI $NO_2$ VCDs are generally underestimated, the HCHO VCDs are generally well accounted for. Hence the true $HCHO/NO_2$ will be smaller than those indicated by satellite observations, which indicates that the estimated sensitivity of the ozone production regime towards $NO_x$ should be smaller and shifted towards VOCs. Using WRF-CMAQ model simulation at 36×36 $km^2$ resolution model over India for 2010, Sharma et al. (2016) have evaluated the ozone production to be strongly sensitive to $NO_x$ emissions throughout the year and recommended reduction in transport emissions which account for 42% of the total $NO_x$ emissions. However, with an increase in transport and powerplant emissions (strong $NO_x$ sources) over India, the regimes are susceptible to shift away from $NO_x$ limited and need to be re-evaluated."

10. 'However, these are comparable to previous in situ NO2 VMR measured for a period of more than one year at urban and suburban locations of India (e.g. Mohali 8.9 ppb, Pune ~9.5ppb and Kanpur 5.7ppb)(Gaur et al., 2014;Kumar et al., 2016).'What about other locations from India such as Debaje and Kakade (2009) and Beig et al. 2007. There are many other observations of NO2 and HCHO from urban and suburban regions in India. Please update the ciations.

We thank the reviewer for indicating the additional works. We have included several additional $NO_2$ measurements from India for comparison. The two references suggested by the reviewer, however, report the total $NO_x$ and not $NO_2$. In order to compare these to our measurements, we have used $NO_2/NO_x$ ratio of 0.9 (Kunhikrishnan et.al., 2006) to estimate mean $NO_2$ VMR.

Accordingly, lines 685-687 of the original manuscript has been modified as follows:

"However, these are comparable to previous *in situ* $NO_2$ VMR measured for a period of more than one year at urban and suburban locations (distant from traffic) of India (e.g. Mohali: 8.9 ppb, Pune: ~9.5ppb/8.7ppb and Kanpur 5.7ppb) (Gaur et al., 2014;Kumar et al., 2016;Beig et al., 2007;Debaje and Kakade, 2009), but smaller than near traffic urban measurement (e.g. New Delhi: 12.5ppb/18.6 ppb, Agra: 15-35 ppb) (Saraswati et al., 2018;Tiwari et al., 2015;Singla et al., 2011). Please note that we have used a $NO_2/NO_x$ ratio of 0.9 to estimate $NO_2$ VMR for comparison with the previous measurements which reported $NO_x$ VMR and hence have a larger uncertainty (Kunhikrishnan et al., 2006)."

Concerning formaldehyde, we have found three previous ambient measurements from India, one using MAX-DOAS and other two employing offline techniques. Out of these two offline measurements, Ghosh et.al. 2015 reported mean ambient HCHO VMR of 217ppb, which is very high and does not represent ambient concentration in our opinion. In the revised manuscript, we added the following line:

"The measured HCHO VMRs are comparable to previous MAX-DOAS measurements from India (Pantnagar: 2-6 ppb), but much lower than those measured previously in India using offline techniques (e.g. North Kolkata:16 ppb, South Kolkata: 11.5ppb) (Dutta et al., 2010;Hoque et al., 2018)"

11. The explanation for higher mixing rations in the MAX-DOAS as compared to in situ observations is not satisfactory. The fact that both the species how higher values for the MAX-DOAS are indicative of an instrumental bias, rather than source regions as speculated in the paper. If the authors are convinced that the power plant, or VOC degradation at higher altitudes contributes to this effect, it can be checked using air mass back trajectories.

The two important factors contributing to the higher $NO_2$ mixing ratios for MAX-DOAS as compared to in situ observations are:

1. The measurement location is relatively cleaner than the surroundings. In contrast to the *in situ* measurements, MAX-DOAS measurements are not only sensitive to the trace gas mixing ratios at the measurement location, but also to the trace gas mixing ratios in the viewing direction upto a distance of several km. The MAX-DOAS instrument is pointing towards the city of Chandigarh which also shows higher $NO_2$ VCD (Fig. 1.)
2. MAX-DOAS surface VMR are influenced by higher altitudes (which are more representative of the larger area with higher $NO_2$ mixing ratios). Due to the coarse vertical resolution of MAX-DOAS profiles, the MAX-DOAS surface VMRs are also influenced by the $NO_2$ at higher altitudes (e.g. that from powerplant plumes)

The Rupnagar power plant (PP1) powerplant is located ~45 km, 340 °N from the measurement site and was operational until the end of 2014.

To further confirm the possible role of PPI towards high surface VMR observed by MAX-DOAS, here we show the hexbin plot showing the frequency of the ratios of MAX-DOAS and in situ $NO_2$ vs *in situ* $NO_2$ VMR separately for the years 2013-2014 (left) and 2015-2017 (right). We observe that for the year 2013-2014, the ratio was > 1 for a large fraction of data.

[Figure]

Fig: Hexbin plot showing the frequency of the ratio of MAX-DOAS and NO₂ surface VMRs against in situ NO₂ VMRs for 2013-2014 (left panel) and 2015-2017(right panel).

We acknowledge the reviewer's recommendation to also check the airmass back trajectories to confirm if plumes from PP1 approached the field of view of the MAX-DOAS. Figure 4 of Pawar et.al., (2015) have previously shown back trajectories of air masses arriving at Mohali for a period of 2 years (2011-2013). Except for monsoon, more than 80% of the back trajectories were among the clusters 'westerlies', 'local' or 'calm', all of which include the location of PP1. In monsoon, these clusters accounted for more than 50% of the total. The GDAS (global data assimilation system) meteorological inputs used for calculating the back trajectories using HYSPLIT are available at 0.5° (~56 km along latitudes) resolution in the best case. Hence, we feel that local wind vectors, measured at Mohali, would provide a more robust validation of our hypothesis than the air mass back trajectories for distances of this scale.

Hence, we show the wind rose plot for the four major seasons overlaid on mean TROPOMI NO₂ maps around Mohali similar to that in Figure 1 of the manuscript. We observe that in all the seasons except monsoon, the major fetch region includes PP1 (which also lies in the viewing direction of MAX-DOAS). It should be noted that the NO₂ maps were generated using TROPOMI measurements for the period Dec 2017-Oct 2018 and hence PP1 does not stand out as a strong NO₂ source

[Figure]

Fig: Wind rose plots overlaid on the mean TROPOMI NO₂ map around Mohali showing the prevalent wind speed and direction during the four major seasons of the year.

In the revised manuscript, we show the wind rose plots (not overlaid on the TROPOMI map shown in Figure 1) as Fig F1, modify lines 130-132 of the original manuscript and add the following text:

Lines 138-140
Fig. F1 shows the wind rose plots indicating the wind speed and wind direction frequencies around Mohali in the four major seasons over the measurement period.

Lines 1168-1170
Pawar et al. (2015) have previously shown back trajectories of air mass arriving at Mohali for a period of 2 years (2011-2013). Except for monsoon, more than 80% of the back trajectories were among the clusters 'westerlies', 'local' or 'calm', all of which include the location of PP1. In monsoon, these clusters accounted for more than 50% of the total. From the wind rose plot of Fig F1, we also observe that in all the seasons except monsoon, the major fetch region includes PP1.

[Figure]

Figure F1: Wind rose plot showing the major fetch region of air mass arriving at Mohali for the four major seasons of the year

For HCHO, however, we do not speculate about any particular source region to be responsible for the observed higher MAX-DOAS surface VMRs. We propose, one plausible explanation based on the vertical profiles of HCHO (as shown in Fig 5 and also previous works e.g. Fig 8 of Kaiser et.al., 2015). HCHO is formed from its precursors (e.g. alkenes, isoprene) during the course of their vertical mixing in the boundary layer (e.g. lines 757-760 of the original manuscript). In situ measurements, which are more sensitive close to the inlet location, do not sample the HCHO which is formed at higher altitudes. As the surface VMR from MAX-DOAS are influenced by the values from higher altitudes, these are accounted for in the mean MAX-DOAS surface VMR in the lowest 200m layer.

12. Figures D6 and D7 need to be discussed in more details in terms of the chemistry leading to large diurnal differences between the in situ and MAX-DOAS observations.

We observe similar diurnal profiles of $NO_2$ surface VMR from in situ and MAX-DOAS measurements. The slightly higher absolute values in MAX-DOAS $NO_2$ VMR are related to the fact that the diurnal profiles are calculated by binning the raw time series data according to the hour of the day and then calculating the statistics. Hence, any bias in the raw time series data will also be propagated to the diurnal profiles. This difference is more pronounced in winter, because of shallower layer heights and the presence of $NO_2$ at altitudes higher than the inlet of in situ analyser (Fig 13c and lines (738-740 of the original manuscript). Additionally, there was a noticeable difference in the occurrence of the morning peak between the two measurements, which we have explained in lines 726-740 of the original manuscript.

For HCHO, indeed, we observe a larger difference in the diurnal patterns. However, these differences are also seen in the raw time series data (e.g. Fig 13 of the original manuscript). As the reviewer indicated, the chemistry might lead to these differences, we have mentioned the following in lines 1206-1214 of the revised manuscript:

"Secondary photochemical production is the major source of atmospheric formaldehyde. The photo-oxidation of primarily emitted VOCs occurs during the course of their mixing up in the boundary layer, and hence, a significant amount of formaldehyde is observed at altitudes up to 600 m or even higher in some cases. The surface VMRs from MAX-DOAS shown in Fig. 13 represent the mean in the lowest 200 m layer of the MAPA output, which might also be influenced by higher altitudes due to limited vertical resolution of MAX-DOAS. Surface VMRs from the PTR-MS measurements are sensitive to the inlet height (~15m). Hence, a higher VMR from MAX-DOAS measurement was expected. This is further supported by our observations in Fig. F8, where we observe that for the periods when the emissions of precursors of HCHO are higher (e.g., from crop residue fires in May, June, October and November and from burning for domestic heating in Dec. and Jan.), the bias between the MAX-DOAS and in situ VMRs is also higher."

**References:**
Donner, S., Kuhn, J., Van Roozendael, M., Bais, A., Beirle, S., Bösch, T., Bognar, K., Bruchkouski, I., Chan, K. L., Dörner, S., Drosoglou, T., Fayt, C., Frieß, U., Hendrick, F., Hermans, C., Jin, J., Li, A., Ma, J., Peters, E., Pinardi, G., Richter, A., Schreier, S. F., Seyler, A., Strong, K., Tirpitz, J. L., Wang, Y., Xie, P., Xu, J., Zhao, X., and Wagner, T.: Evaluating different methods for elevation calibration of MAX-DOAS (Multi AXis Differential Optical Absorption Spectroscopy) instruments during the CINDI-2 campaign, Atmos. Meas. Tech., 13, 685-712, 10.5194/amt-13-685-2020, 2020.

Ghosh, D., Sarkar, U. & De, S. Analysis of ambient formaldehyde in the eastern region of India along Indo-Gangetic Plain. Environ Sci Pollut Res 22, 18718–18730 (2015), 10.1007/s11356-015-5029-y

Kaiser, J., Wolfe, G. M., Min, K. E., Brown, S. S., Miller, C. C., Jacob, D. J., deGouw, J. A., Graus, M., Hanisco, T. F., Holloway, J., Peischl, J., Pollack, I. B., Ryerson, T. B., Warneke, C., Washenfelder, R. A., and Keutsch, F. N.: Reassessing the ratio of glyoxal to formaldehyde as an indicator of hydrocarbon precursor speciation, Atmos. Chem. Phys., 15, 7571-7583, 10.5194/acp-15-7571-2015, 2015.

Kunhikrishnan, T., Lawrence, M. G., von Kuhlmann, R., Wenig, M. O., Asman, W. A. H., Richter, A., and Burrows, J. P.: Regional NOx emission strength for the Indian subcontinent and the impact of emissions from India and neighboring countries on regional O3 chemistry, Journal of Geophysical Research: Atmospheres, 111, 10.1029/2005jd006036, 2006.

Pawar, H., Garg, S., Kumar, V., Sachan, H., Arya, R., Sarkar, C., Chandra, B. P., and Sinha, B.: Quantifying the contribution of long-range transport to particulate matter (PM) mass loadings at a suburban site in the north-western Indo-Gangetic Plain (NW-IGP), Atmos. Chem. Phys., 15, 9501-9520, 10.5194/acp-15-9501-2015, 2015.

Seidel, D. J., Zhang, Y., Beljaars, A., Golaz, J.-C., Jacobson, A. R., and Medeiros, B.: Climatology of the planetary boundary layer over the continental United States and Europe, Journal of Geophysical Research: Atmospheres, 117, 10.1029/2012jd018143, 2012.

Wagner, T., Apituley, A., Beirle, S., Dörner, S., Friess, U., Remmers, J., and Shaiganfar, R.: Cloud detection and classification based on MAX-DOAS observations, Atmos. Meas. Tech., 7, 1289-1320, 10.5194/amt-7-1289-2014, 2014.

Wagner, T., Beirle, S., Remmers, J., Shaiganfar, R., and Wang, Y.: Absolute calibration of the colour index and O4 absorption derived from Multi AXis (MAX-)DOAS measurements and their application to a standardised cloud classification algorithm, Atmos. Meas. Tech., 9, 4803-4823, 10.5194/amt-9-4803-2016, 2016.

Wang, Y., Penning de Vries, M., Xie, P. H., Beirle, S., Dörner, S., Remmers, J., Li, A., and Wagner, T.: Cloud and aerosol classification for 2.5 years of MAX-DOAS observations in Wuxi (China)

and comparison to independent data sets, Atmos. Meas. Tech., 8, 5133-5156, 10.5194/amt-8-5133-2015, 2015.

Wang, Y., Dörner, S., Donner, S., Böhnke, S., De Smedt, I., Dickerson, R. R., Dong, Z., He, H., Li, Z., Li, Z., Li, D., Liu, D., Ren, X., Theys, N., Wang, Y., Wang, Y., Wang, Z., Xu, H., Xu, J., and Wagner, T.: Vertical profiles of NO2, SO2, HONO, HCHO, CHOCHO and aerosols derived from MAX-DOAS measurements at a rural site in the central western North China Plain and their relation to emission sources and effects of regional transport, Atmos. Chem. Phys., 19, 5417-5449, 10.5194/acp-19-5417-2019, 2019.

---

## Author Response (AR2)

Dear Editor,

Thanks a lot for your positive feedback on the revised manuscript. We acknowledge your suggestion for minor revisions. Our response to your comments and questions are listed below. For a convenient perusal, we have also appended the currently revised manuscript with track changes enabled after the answers to the individual points. We hope that the changes in the manuscript address yours and the reviewers remaining concerns and you find the manuscript suitable for the readership of Atmospheric Chemistry and Physics. Your comments are shown in blue, our response is coloured black, and the modifications in the manuscript are marked red.

Best regards,
Vinod Kumar
(on behalf of all the co-authors)

I found that the manuscript was revised successfully, by taking the comments from the two reviewers into account. I would request minor revision/discussion on the following points.

Thanks a lot. We have undertaken the minor revisions recommended by you.

1. The original comment made by the reviewer #1 as follows aims to integrate "satellite AMFs" into account, not "MAX-DOAS averaging kernels". However, the authors' rejected this comment based on the unavailability of the "MAX-DOAS averaging kernels". The reply needs to be revised.

Page 21, lines 646-648: Since satellite total HCHO AMFs and averaging kernels are missing in the files, you could proceed the other way round for eliminating the difference caused by the non-representative satellite a priori HCHO profiles, i.e. recalculating satellite AMFs using MAX-DOAS vertical profiles and dividing the satellite slant column densities by those new AMFs (see e.g. De Smedt et al., Atmos. Chem. Phys., 15, 12519–12545, 2015). Maybe something worth to try.

In the original manuscript, we indicate that the satellite air mass factors (AMFs) might not be accurate because of the non-representative vertical profiles. Accurate satellite AMFs can be recalculated using MAX-DOAS vertical profiles as a priori, which could improve the agreement with MAX-DOAS observations as also shown by De Smedt et al. (2015). We now investigate this method. We have changed lines 602-605 of the previously revised manuscript to the following in current revision and extended the appendix D.

"A different approach for improved agreement between MAX-DOAS and satellite VCDs is by using the MAX-DOAS $NO_2$ profiles as a priori profiles for the calculation of airmass factors for the satellite retrieval (Chan et al., 2019;De Smedt et al., 2015). We discuss this approach and its limitations in appendix D"

The following discussion is added to the Appendix D:

Accurate satellite AMFs can be recalculated using MAX-DOAS vertical profiles as a priori, which could improve the agreement with MAX-DOAS observations as also shown by De Smedt et al. (2015). The satellite AMFs corresponding to the MAX-DOAS a priori profiles ($x_m$) can be recalculated according to the following equation:

$$AMF_{trop}(x_m) = AMF_{tot}(x_a) \frac{\sum_{l=1}^{L} A_l x_{m,l}}{\sum_{l=1}^{L} x_{m,l}} \qquad \text{D2}$$

Here,

$x_a$ : Original satellite (TM5) a priori trace gas profile
$L$ : Tropopause level index
$A$ : Satellite averaging kernels
$AMF_{tot}$ : Total airmass factor

Total AMF ($AMF_{tot}$) and satellite averaging kernels used for OMI retrievals are crucial information required to recalculate the satellite (OMI) AMF. For $NO_2$, only the DOMINO product provides both $AMF_{tot}$ and satellite averaging kernels in the data product. We attempted to recalculate the $AMF_{trop}$ using the MAX-DOAS profiles, but this resulted in very small airmass factors (and very large recalculated OMI VCDs). The small $AMF_{trop}$ is due to the fact that the MAX-DOAS profiles do not account for the background $NO_2$ in the free troposphere, where the satellite averaging kernels are large. In the next step, we used hybrid profiles such that we only replaced the profiles in the lowest 5 km and 2.5 km of the TM5 profile with those retrieved from MAX-DOAS measurements. The observations are summarized in Fig D3. We note that even with the hybrid approach, there is an

overestimation of VCDs for many months. This is probably caused by the incorrect aerosol profiles used for the calculation of the averaging kernels in the satellite analyses.

[Figure]

**Figure D3: Time series of the MAX-DOAS and OMI DOMINO NO₂ tropospheric VCDs for A) when no modifications in the a priori profiles are applied, B) when the a priori profiles of DOMINO are replaced by the MAX-DOAS profiles, C) when the lowest 5 km of the profiles of the a priori are replaced by the corresponding MAX-DOAS profiles and D) when the lowest 2.5 km of the profiles of the a priori are replaced by the corresponding MAX-DOAS profiles.**

For formaldehyde, $AMF_{tot}$ is not provided in the QA4ECV products, while averaging kernels are not available for OMHCHO products. However, we approximated $AMF_{tot}$ to be close to $AMF_{trop}$ because of the negligible amount of HCHO present in the stratosphere. Using this approximation, we have recalculated the modified $AMF_{trop}$ using MAX-DOAS profiles as a priori. Similar to NO₂ we have replaced the profile in the lowest 2.5 km of the TM5 profile with that retrieved from MAX-DOAS measurements. For HCHO, the modified VCDs are largely positive biased. Like for NO₂, this overestimation might be caused by incorrect aerosol profiles used for the calculation of the averaging kernels in the satellite analyses.

[Figure]

**Figure D4: Time series of the MAX-DOAS and OMI QA4ECV HCHO tropospheric VCDs for A) when no modifications in the a priori profiles are applied, B) when the a priori profiles of QA4ECV are replaced by corresponding MAX-DOAS profiles and C) when the lowest 2.5 km of the a priori profiles are replaced by corresponding MAX-DOAS profiles.**

2. Another original comment made by the reviewer #1 as follows implicitly aimed comparison between the converted MODIS AOD at 360 nm with that of MAX-DOAS. Some comments are needed on this point.

Page 50, Figure D4: How MODIS would compare to the AERONET sun photometer measurements at these two sites? Angstrom exponent could be used to convert MODIS AOD from 470 nm to 360 nm.

In the original manuscript, we have used the Ångström exponent to convert MODIS AOD at 470 nm to that at 360 nm (equation 3) and compared the converted AODs with MAX-DOAS (Fig 7A and 7B and section 3.3 of the original and previously revised manuscripts).

We followed from the above-mentioned comment of reviewer #1 that she/he wanted to see, how does MODIS AOD (at 360 nm) compare with **AERONET** measurements at Lahore and New Delhi. Please correct us if we missed something here. In the previously revised manuscript, though, we modified Fig F5 to show the comparison of AOD measurements from MODIS and AERONET for these two sites, yet we did not add the corresponding discussion in the manuscript. We now add the following lines (345-348) in the revised manuscript to include the relevant test. Please note that, for this comparison, we converted the MODIS AOD from 470 nm to 440 nm for comparison with similar AERONET measurements.

"We have also compared the MODIS AOD (converted to 440 nm using the AE derived from equation 3) to the AERONET AOD at these two stations and found very good agreement in the daily measured values with Pearson correlation coefficients (r) > 0.84 (Fig. F5) and an overall bias < 10% for both sites."

3. Replies to the reviewer #2 and the relevant revisions are mostly reasonable, particularly for making focused scientific points, regarding 1) source/process analysis on the seasonal and diurnal trends of AOD, $NO_2$, and HCHO, 2) analysis of BLH and deep convection, and also regarding 3) $O_3$ production regimes in the morning and late afternoon hours. Also, I believe that the point that this study provides the first evaluation of three OMI $NO_2$ data products and two OMI HCHO data products over the IGP is strong.

Thanks for the encouraging remarks and indicating this strong point of our study. We have added this to the abstract and the conclusion of the revised manuscript at lines 29-30 and 765-767, respectively.

"The ground-based MAX-DOAS measurements were used to evaluate three $NO_2$ data products and two HCHO data products of the ozone monitoring instrument (OMI) for the first time over India and the IGP."

"We use the MAX-DOAS measurements of $NO_2$ and HCHO to evaluate the three widely used $NO_2$ data products (DOMINO v2, QA4ECV and OMNO2) and two HCHO data products (QA4ECV and OMHCHO) of OMI for the first time over India and the Indo-Gangetic plain."

4. Lines 1395-1396 of the change-track version. I would suspect that information from lower elevation angles than 30 degrees are equally important when deriving VCD. The aerosol information from the O4 analysis would be most worth when systematic difference between NO2 VCDs between satellites and MAX-DOAS is analyzed. I believe this was the main point that the reviewer #1 made with the relevant comment.

Thanks for the suggestion. We agree with you that lower elevation angles are also important for deriving trace gas VCDs. We did sensitivity studies to check if the true VCD (AOD) is retrieved from the profile inversion if high or low elevation angles are skipped. These studies were performed using synthetic trace gas dSCDs corresponding to a trace gas VCD of $1.0 \times 10^{16}$ molecules $cm^{-2}$ (AOD of 0.5) and noise of $2.0 \times 10^{41}$ molecules$^2$ $cm^{-5}$ and $5.0 \times 10^{14}$ molecules $cm^{-2}$ for $O_4$ and $NO_2$, respectively (Frieß et al., 2019). For trace gases, there was no systematic (or strong) effect if individual elevation angles were skipped. Even if all the elevation angles > 10° or < 5° were skipped, the retrieved VCDs were within 10% of the true VCD. However, for aerosol, skipping high elevation angles resulted in larger biases.

[Figure]

**Figure 1: Left trace gas VCD retrieved for synthetic spectra corresponding to $1.0 \times 10^{16}$ molecules cm$^{-2}$ using MAPA if specific elevation angles are skipped from the elevation sequence. Right: Same as left but for aerosol inversions using synthetic $O_4$ dSCDs (AOD = 0.5).**

We agree that the sensitivity of the MAX-DOAS instrument depends on the aerosol load, and this is a crucial information for comparing MAX-DOAS and satellite observations of trace gases. Nonetheless, the sensitivity distance is less than 10 km (e.g. Fig F13 of the revised manuscript), while we extracted OMI $NO_2$ and HCHO VCDs within ~25 km of Mohali for the comparison with MAX-DOAS. The aerosol information and the agreement of satellite and MAX-DOAS HCHO VCDs for various aerosol scenario would be interesting to study for the new satellite instrument, e.g. TROPOMI which have a much finer spatial resolution. For the AOD comparison in our study, this is crucial as the MAIAC data is available at a spatial resolution of $1 \times 1$ km$^2$. However, as demonstrated above, for the aerosol inversion, the AOD is mainly constrained using the high elevation angles, for which the sensitivity range is very close to the MAX-DOAS instrument.

As recommended by reviewer #1, we investigated the effect of expanding the area of consideration for the AOD comparison from 2 km to 5 km around Mohali. However, this did not bring noticeable change in the agreement between MAX-DOAS and MAIAC AOD observations, indicating a rather smooth horizontal distribution of aerosol within 5 km around Mohali. Though this was indicated the response to reviewer #1, it was not included in the previously revised manuscript. In the revised manuscript now, we have moved the discussion of horizontal sensitivity distance from section 3.5 to 3.4 with some changes as shown below (lines 517-528) of the revised manuscript:

"MAX-DOAS measurements are spatially representative of a few kilometres in the field of view, depending on the ambient aerosol load and elevation angle, whereas the ground footprints of individual OMI pixels are $13 \times 24$ km$^2$ in the best case. We have calculated the horizontal sensitivity distance (HSD) of MAX-DOAS for low elevation angles as the e-folding distance of $O_4$ dAMF from the instrument location (Wagner and Beirle, 2016). Figure F13 shows that the mean afternoon time (between 12:00 and 15:00 local time) HSD ranges between 5 and 7 km for clear sky condition with low aerosol load and between 3 and 6 km for high aerosol conditions. Here it is important to note that this estimate is mainly representative for the near-surface layers. While for the trace gas inversions, the VCD is constrained by all elevation angles, the determination of the AOD is mostly constrained by the high elevation angles. For high elevation angles, the sensitivity range is much closer to the instrument (at distances up to 1 and 2 km for layer height of 0.5 and 1 km, respectively) (see Fig 6A). Comparing the spatially degraded time series with MAX-DOAS AOD resulted in a worse agreement (r=0.75 and 0.79 for the daily and monthly means, respectively) for 25 km but did not change significantly for 5 km (Fig. F14) as compared to the original comparison when only a 2 km area around Mohali was considered for spatial averaging."

[Figure]

Figure F14: Intercomparison of daily (dots) and monthly mean (lines and markers) AOD at 360 nm retrieved from ground-based MAX-DOAS $O_4$ measurements and from the MODIS MAIAC data product when spatially averaged over 5 km (top panel) and 25 km (bottom panel) around Mohali. $O_4$ dSCD were scaled by a factor of 0.8, as discussed in the main text.

Figure F13 or the previously revised manuscript is updated accordingly (as shown above) (F14 of the revised manuscript))

5. Lines 1488 of the change-track version: Schofield et al. (2006) must be Schroeder et al. (2017).

Thanks for the correction. Done.

**References:**

Frieß, U., Beirle, S., Alvarado Bonilla, L., Bösch, T., Friedrich, M. M., Hendrick, F., Piters, A., Richter, A., van Roozendael, M., Rozanov, V. V., Spinei, E., Tirpitz, J. L., Vlemmix, T., Wagner, T., and Wang, Y.: Intercomparison of MAX-DOAS vertical profile retrieval algorithms: studies using synthetic data,

[revised manuscript text omitted]